# RESEARCHTOWN: Simulator of Human Research Community

**Haofei Yu** [1] [*]   **Zhaochen Hong** [1] [*]   **Zirui Cheng** [1] [*]   **Kunlun Zhu** [1] [*]
**Keyang Xuan** [1]   **Jinwei Yao** [1]   **Tao Feng** [1]   **Jiaxuan You** [1]

 Code: `https://github.com/ulab-uiuc/research-town`
 Data: `https://huggingface.co/datasets/ulab-ai/research-bench`

## Abstract

Large Language Models (LLMs) have demonstrated remarkable potential in scientific domains, yet a fundamental question remains unanswered: *Can we simulate human research communities with LLMs?* Addressing this question can deepen our understanding of the processes behind idea brainstorming and inspire the automatic discovery of novel scientific insights. In this work, we propose RESEARCHTOWN, a multi-agent framework for research community simulation. Within this framework, the human research community is simplified as an *agent-data graph*, where researchers and papers are represented as agent-type and data-type nodes, respectively, and connected based on their collaboration relationships. We also introduce *TextGNN*, a text-based inference framework that models various research activities (*e.g.*, paper reading, paper writing, and review writing) as special forms of a unified message-passing process on the agent-data graph. To evaluate the quality of the research community simulation, we present RESEARCHBENCH, a benchmark that uses a node-masking prediction task for scalable and objective assessment based on similarity. Our experiments reveal three key findings: (1) RESEARCHTOWN can provide a realistic simulation of collaborative research activities, including paper writing and review writing; (2) RESEARCHTOWN can maintain robust simulation with multiple researchers and diverse papers; (3) RESEARCHTOWN can generate interdisciplinary research ideas that potentially inspire pioneering research directions.

---

[*]Leading authors. Details of individual contributions are available at Appendix §A. [1]University of Illinois Urbana-Champaign. Correspondence to: Haofei Yu <haofeiy2@illinois.edu>, Jiaxuan You <jiaxuan@illinois.edu>.

*Proceedings of the $42^{nd}$ International Conference on Machine Learning*, Vancouver, Canada. PMLR 267, 2025. Copyright 2025 by the author(s).

## 1. Introduction

LLMs have proved to be powerful copilots in scientific research (AI4Science & Quantum, 2023), demonstrating their great potential for accelerating scientific discovery. Despite the promising finding, a more ambitious question remains: *Can we simulate the human research community with LLMs*? Answering such a question has multiple benefits: (1) simulating the human research community helps understand the underlying process behind the discovery of existing research ideas; (2) it can further help democratize and accelerate the discovery process of new research ideas.

However, simulating the human research community is challenging, as it involves leveraging multiple LLM agents to interact with complex research data. While existing multi-agent LLM frameworks have been successfully applied to areas like social simulation (Zhou et al., 2023; Gao et al., 2023) and game simulation (Hua et al., 2023; Xu et al., 2023), they are not well-suited for simulating research communities due to the complexity of collaborative research activities like paper writing and review writing. Although recent efforts have explored research automation using LLMs, these frameworks are typically limited to specific research tasks, such as idea generation (Girotra et al., 2023; Baek et al., 2024) or code experimentation (Huang et al., 2024), or focus on simulating single-agent workflows (Lu et al., 2024). These frameworks cannot simulate collaborative research activities where researchers with diverse backgrounds work together to brainstorm ideas, review papers, etc—processes that are fundamental to modern human research.

**Research community as graph**. Our key observation is that the deeply interconnected research community can be naturally represented as graphs. Indeed, similar graph structures like citation networks (Newman, 2001) and academic social networks (Tang et al., 2008) have been extensively studied within data mining research, with proven values in applications such as citation prediction (Holm et al., 2020), recommendation (West et al., 2016), and community detection (Yang & Leskovec, 2012). However, introducing LLMs to a graph-structured research community can extend these previous works from prediction and analysis with existing

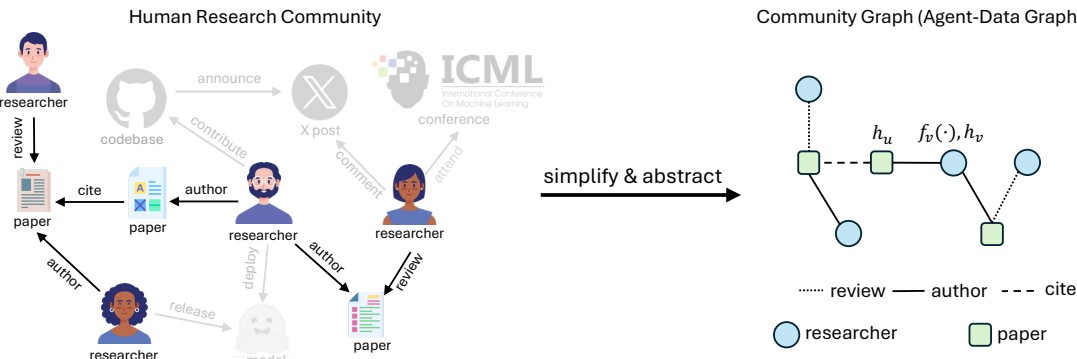

Figure 1: **Abstracting and simplifying human research community as an agent-data graph, *i.e.*, community graph**. An agent-data graph has researchers as agent nodes and blogs, codebases, and papers as data nodes. Without losing generality, we abstract it into a simplified version with only researcher and paper nodes and focus on critical research tasks, including paper reading, paper writing, and review writing. Each data node has a hidden state $h_u$, and each agent node is paired with an agent function $f_v(\cdot)$ and a hidden state $h_v$.

data to dynamic simulation and real-time forecasting.

**Novel framework for research simulation**. In this work, we propose RESEARCHTOWN, a simulator of the human research community. To bridge the gap between existing multi-agent simulation frameworks and the complexity of research activities, we propose a graph-based framework, inspired by the message-passing mechanism in Graph Neural Networks (GNNs), for multi-agent simulation. Concretely, as shown in Figure 1, we propose a new concept of *agent-data graph* with 2 generic types of nodes: (1) *agent* nodes, suitable for entities like agents; (2) *data* nodes, suitable for entities such as papers, reviews, and blogs. Agent-data graphs are unique from standard heterogeneous graphs; here, the key conceptual difference between agent and data nodes is that an agent node can be considered a function over data nodes. To inference on agent-data graphs, we propose a *TextGNN* framework where message-passing processes are defined based on text-form information processing with LLMs, thanks to their strong in-context learning (Wei et al., 2023) and reasoning (Lee et al., 2024) ability. We apply the proposed agent-data graph and TextGNN to the research simulation. Here, a research community can be regarded as a special form of agent-data graph, called *community graph*, with research agents and research papers as two types of nodes, and we consider three types of edges (review, author, and cite) in the graph. Different community activities, such as paper writing and review writing, can be modeled as special message-passing processes on the community graph.

**Novel evaluation for research simulation**. With RESEARCHTOWN for research simulation, a further research question is to evaluate the quality of that. Prior works primarily use human evaluation with breakdown metrics such as novelty, excitement, feasibility, and expected effectiveness (Si et al., 2024; Hu et al., 2024b). These approaches

inevitably suffer from subjectiveness and high costs. In our work, since RESEARCHTOWN functions as a simulator, our primary focus is on measuring how closely its outputs align with those of the real-world research community. Community graphs naturally provide a similarity-based evaluation method by masking a given paper node in the community graph and evaluating whether a simulator can reconstruct the masked nodes. This definition focuses on simulation similarity, making it scalable and objective. Based on such a node masking prediction task, we build a benchmark called RESEARCHBENCH with 1,000 paper writing tasks and 200 review writing tasks requiring multi-agent collaboration.

**Main discoveries**. Based on the evaluation results from RESEARCHBENCH, we highlight three key findings: (1) RESEARCHTOWN effectively simulates collaborative research activities, achieving an average similarity score of 0.68 for paper writing and 0.49 for review writing, as measured by the state-of-the-art text embedding model; (2) RESEARCHTOWN demonstrates robustness and effectiveness in research simulation, showing improvement when more agents are added and maintaining performance when including unrelated papers; (3) RESEARCHTOWN inspires interdisciplinary research, generating innovative ideas that combine insights from NLP, criminology, and astronomy and does not exist in the real-world research.

**Stressing ethical concerns**. As our work targets simulating the human research community, multiple ethical concerns, including facilitating research plagiarism and producing low-quality or misleading claims, appear. These ethical concerns are addressed in detail in Appendix §B.

## 2. Additional Related Work

**Graphs with text attributes**. In real-world graph tasks, nodes often have textual attributes to carry richer informa-

tion, forming text-attributed graphs (TAGs) (Yang et al., 2021; He et al., 2023). Previous work on TAGs mainly treats LLMs as tools for understanding text attributes and aims at achieving co-training LLMs and GNNs (Zhao et al., 2023; Chen et al., 2024). In contrast, our approach incorporates agent nodes into the graph, enabling text-based message passing between agent nodes and data nodes. Furthermore, while previous TAG research mainly focuses on node prediction and link prediction tasks (Yan et al., 2023), RESEARCHTOWN extends it to both the reconstruction of existing nodes and the prediction of new, non-existent nodes.

**Graphs for multi-agent modeling**. Recent works model multi-agent communication using graphs and develop learnable methods to optimize the communication process (Zhuge et al., 2024; Martinkus et al., 2023; Hu et al., 2024a). However, these works often neglect the interactive nature of data, where agents can read, write, and update shared data iteratively. Currently, few works include a well-defined framework to represent graphs that integrate both agents and their associated data.

## 3. Agent-Data Graph for Multi-agent LLMs

**Definition of agent-data graphs**. To initiate our discussion, we formally define the proposed agent-data graph. An agent-data graph is a special type of heterogeneous graph $\mathcal{G} = (\mathcal{V}, \mathcal{E})$, where $\mathcal{V} = \mathcal{V}_a \cup \mathcal{V}_d$ is the node set consisting of two types of nodes, agent nodes and data nodes, and $\mathcal{E} = \mathcal{E}_{aa} \cup \mathcal{E}_{ad} \cup \mathcal{E}_{dd}$ is the edge set consisting of three types of relations, agent-agent, data-data, and agent-data interactions. Here, each data node $v \in \mathcal{V}_d$ comes with attributes, e.g., a piece of text, $\mathbf{x}_v$; each agent node $u$ is accompanied with an *agent function*, e.g., an LLM $f_u(\cdot)$ with its prompt template and the profile. Each agent function is responsible for two types of tasks: message generation and message aggregation. More details about agent functions are in Appendix §D.3. Without loss of generality, we assume that the data nodes have text attributes, and leave the multi-modal extension of our work, e.g., images, audio, and videos, to future works.

**Uniqueness of agent-data graphs**. Unlike standard heterogeneous graphs, the uniqueness of an agent-data graph is that the agent nodes take functions as their attributes, rather than embeddings. Concretely, each agent node could take a piece of text, e.g., $\mathbf{x}_v$ from one data node, as the input and output new data based on its profile prompt $\mathbf{x}_u$, e.g., $\mathbf{x}_{uv} = f_u([\mathbf{x}_u, \mathbf{x}_v])$ where $[\cdot]$ indicates filling the prompt template with $\mathbf{x}_u$ and $\mathbf{x}_v$. Such definition greatly facilitates the multi-agent scenarios where agents could communicate among themselves, with edge type $\mathcal{E}_{aa}$; interacting with the environment, with edge type $\mathcal{E}_{ad}$; representing the inherent data relationships within an environment $\mathcal{E}_{dd}$.

**Example of agent-data graphs**. Figure 1 shows an example of the agent-data graph. Its definition could be extended to more node types (e.g., codebase, blogs) and edge types (e.g., attend, post, commit). Typically, one blog post can be directly connected to multiple researchers, papers, and other blog posts if they are related to each other.

## 4. Building TextGNN on Agent-Data Graphs

**TextGNN motivations**. The agent-data graph $\mathcal{G}$ provides a platform for expressing a complex multi-agent scenario, e.g., a human research community. To further simulate based on a given real-world agent-data graph, we need agentic models, e.g., LLMs, to generate new data and interactions on the agent-data graph. To this end, motivated by the message-passing algorithm in GNNs, we proposed a text-based message-passing mechanism on an agent-data graph, called *TextGNN*, where all hidden states are defined in the text space instead of the embedding space.

**Recap: message passing in standard GNN**. In standard GNNs, input features $\mathbf{x}_v$ are used to initialize the initial states $\mathbf{x}_v = \mathbf{h}_v^{(0)}$. Afterward, the goal is to learn useful node embeddings $\mathbf{h}_v$ by iteratively aggregating information from local neighborhoods. Hidden states, message functions, and aggregation functions are the three main components in one GNN layer. The $k$-th iteration of message passing (or the $k$-th GNN layer) is typically defined as:

$$\mathbf{m}_u^{(k)} = \mathrm{MSG}^{(k)}(\mathbf{h}_u^{(k-1)}) \tag{1}$$

$$\mathbf{h}_v^{(k)} = \mathrm{AGG}^{(k)}\big(\mathbf{h}_v^{(k-1)}, \{\mathbf{m}_u^{(k)} \mid u \in \mathcal{N}(v)\}\big) \tag{2}$$

where $\mathbf{h}_v^{(k)}$ is the node embedding at the $k$-th layer, $\mathbf{h}_v^{(0)} = \mathbf{x}_v$ is the initial node feature, and $\mathcal{N}(v)$ is the set of neighbors of node $v$. $\mathrm{MSG}^{(k)}(\cdot)$ is a transformative function to convert the hidden states of one node into a message for aggregation. $\mathrm{AGG}^{(k)}(\cdot)$ is defined to update the hidden states of a node based on neighborhood messages. More generally, we can broadly consider the $k$-th layer of GNN to be an aggregation function that implicitly includes message functions inside:

$$\mathbf{h}_v^{(k)} = \mathrm{AGG}^{(k)}\big(\mathbf{h}_v^{(k-1)}, \{\mathbf{h}_u^{(k-1)} \mid u \in \mathcal{N}(v)\}\big) \tag{3}$$

**Message passing in TextGNN**. Following the message-passing process in the standard GNN, we now define a general form of the aggregation function to describe the text-based message-passing process on an agent-data graph $\mathcal{G}$. The key difference between a standard GNN and a TextGNN is that all hidden states in the standard GNN are defined in the embedding space ($\mathbf{h}_v \in \mathbb{R}^d$) while those in TextGNN are defined in the text space ($\mathbf{h}_v \in \Sigma^*$). In a TextGNN, we first set the initial hidden states for data nodes $\mathbf{h}_v^{(0)} = \mathbf{x}_v$, where $\mathbf{x}_v$ are text attributes, and the initial hidden states for agent nodes is empty $\mathbf{h}_u^{(0)} = \emptyset$. Next, we design a general form of message passing function that handles three

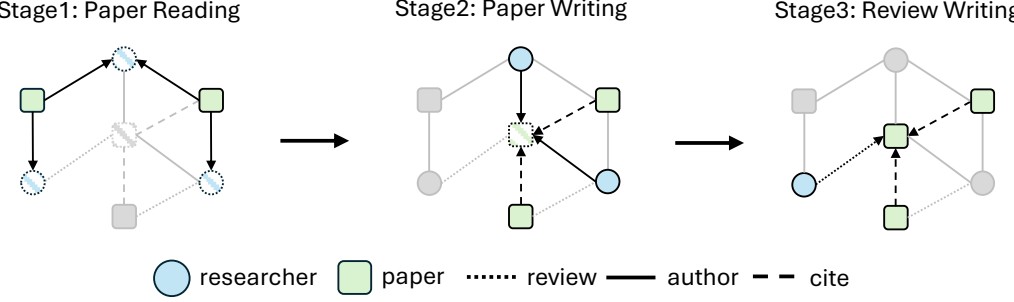

Figure 2: **RESEARCHTOWN simulation as TextGNN inference on the community graph**. The simulation proceeds in three stages: (1) paper reading, where new agent nodes are added based on existing data; (2) paper writing, where data nodes are created; (3) review writing, where the community evaluates and selectively removes (or retains) generated nodes.

distinctive types of interaction, including agent-agent $\mathcal{E}_{aa}$, agent-data $\mathcal{E}_{ad}$, and data-data $\mathcal{E}_{dd}$.

Specifically, the $k$-th TextGNN layer for an agent node $u \in \mathcal{V}_a$ can be written as:

$$\begin{aligned} \mathbf{h}_u^{(k)} &= \text{AGG}^{(k)}\big(f_u(\cdot), \mathbf{h}_u^{(k-1)}, \{\mathbf{h}_d^{(k-1)} \mid (u,d) \in \mathcal{E}_{ad}\}, \\ &\quad \{f_a(\cdot), \mathbf{h}_a^{(k-1)} \mid (u,a) \in \mathcal{E}_{aa}\}\big) \\ &= f_u\Big(\big[\mathbf{h}_u^{(k-1)}, \{f_a([\mathbf{h}_a^{(k-1)}, \mathbf{h}_u^{(k-1)}, \mathbf{h}_d^{(k-1)}])\} \mid \\ &\quad (u,a) \in \mathcal{E}_{aa}, (u,d) \in \mathcal{E}_{ad}\}\big]\Big) \end{aligned} \quad (4)$$

where $[\cdot]$ is the concatenation function between texts to fill in the prompt template, $\mathbf{h}_v^{(k)}$ represents the hidden states of the $k$-th layer of $v \in \mathcal{V}$, $f_a(\cdot)$ represents the agent function paired with the agent node in the neighborhood and $f_u(\cdot)$ represents the agent function paired with the agent node.

Similarly, the forwarding process of the $k$-th TextGNN layer for a data node $v \in \mathcal{V}_d$ can be written as:

$$\begin{aligned} \mathbf{h}_v^{(k)} &= \text{AGG}^{(k)}\Big(\mathbf{h}_v^{(k-1)}, \{\mathbf{h}_d^{(k-1)} \mid (v,d) \in \mathcal{E}_{dd}\}, \\ &\quad \{f_a(\cdot), \mathbf{h}_a^{(k-1)} \mid (v,a) \in \mathcal{E}_{ad}\}\Big) \\ &= f_g\Big(\big[\mathbf{h}_v^{(k-1)}, \{f_a([\mathbf{h}_a^{(k-1)}, \mathbf{h}_v^{(k-1)}, \mathbf{h}_d^{(k-1)}])\} \mid \\ &\quad (v,a) \in \mathcal{E}_{ad}, (v,d) \in \mathcal{E}_{dd}\}\big]\Big) \end{aligned} \quad (5)$$

where $f_g(\cdot)$ is defined as a global agent function without a specialized profile, and $f_a(\cdot)$ is the agent function paired with the agent node in the neighborhood.

## 5. RESEARCHTOWN: Applying TextGNN to Community Graph

**Inputs and outputs of RESEARCHTOWN**. Building on the definitions of TextGNN and the agent-data graph in Section §3 and Section §4, we simulate different research activities by modeling each as a specific instantiation of a TextGNN layer. RESEARCHTOWN processes diverse research materials and produces structured outputs. The input

varies by task: only paper abstracts are used for paper reading and writing, while full papers are provided for review writing. The output format is also task-specific: paper reading generates profile descriptions, paper writing generates bullet-point summaries, and review writing produces bullet-point critiques along with a numerical review score. These standardized output formats—described in more detail in Appendix §F—facilitate evaluation over long-context inputs and enable fine-grained, sub-component similarity scoring.

**Hidden states of RESEARCHTOWN**. In RESEARCHTOWN, the hidden state of each node represents a condensed version of research materials, such as papers or reviews. Initially, paper nodes are initialized with the full text of papers. Through iterative message passing, these nodes gradually evolve into a standardized bullet-point format, distilling key information for easier downstream evaluation. Similarly, review attributes associated with paper nodes are also represented using bullet points to make it in a compact form. Bullet-point compact form with limited length allows TextGNN to conduct message passing multiple times efficiently.

**Agent-data graph for research community modeling - community graph**. We adopt the agent-data graph $\mathcal{G} = (\mathcal{V}, \mathcal{E})$ to research community simulation, which we named as *community graph*. As is shown in Figure 2, each agent node $\mathcal{V}_a$ represents one researcher, and each data node $\mathcal{V}_d$ represents a paper. The edge set $\mathcal{E}_{dd}$ captures paper citations, the edge set $\mathcal{E}_{ad}$ captures authorship (a researcher writes a paper) and reviewing expertise (a researcher is qualified to review a paper). We omit the edge set $\mathcal{E}_{aa}$ to simplify the framework, as a collaboration between authors can typically be inferred through 2-hop paths via $\mathcal{E}_{ad}$ edges.

**TextGNN for research activity simulation**. Based on the constructed community graph, we further identify the key types of research activities where TextGNN can be used for simulation. Specifically, as shown in Figure 2, we split the research simulation process into three critical stages: (1) paper reading, (2) paper writing, and (3) review writing. We

believe these stages are crucial in the research community, and each stage relies on the output of the previous stage as input. We provide a detailed description for each stage and the corresponding TextGNN layer definition below.

▷ **Stage 1: Paper reading**. Reading papers to collect insights is a necessary process for initializing a research project. In the community graph, the paper reading process can be described as *inserting a new agent node* to the community graph and aggregating its neighborhood information based on Equation 4. Here, the new agent profile is non-existent before reading a collection of papers, and the profile is created after the paper reading process, making the TextGNN layer unique. Concretely, by adapting Equation 4, the TextGNN layer for paper reading can be written as:

$$\mathbf{h}_u = \text{AGG}\Big(f_u(\cdot), \{\mathbf{h}_d \mid (u,d) \in \mathcal{E}_{ad}\}\Big)$$
$$= f_u\Big(\Big[\{\mathbf{h}_d \mid (u,d) \in \mathcal{E}_{ad}\}\Big]\Big) \quad (6)$$

where $\mathbf{h}_u, \{f_a(\cdot), \mathbf{h}_a \mid (u,a) \in \mathcal{E}_{aa}\}$ in Equation 4 are empty since the agent node is initialized as empty and is not directly connected with any agents, and $\mathcal{E}_{ad}$ specifically refers to the authorship relation between agent and data nodes. Equation 4 degrades to an aggregation of papers based on the researcher agent without the profile, illustrated in Figure 2 "Stage 1".

▷ **Stage 2: Paper writing**. After paper reading, the next important research stage is paper writing. Different from paper reading, the paper writing process can be understood as inserting *a new data node* into the community graph. Here, the new data node is non-existent before writing the paper, and the data node is created after the paper writing process. Concretely, by adapting Equation 5, the TextGNN layer for paper writing can be written as:

$$\mathbf{h}_v = \text{AGG}\Big(\{f_a(\cdot), \{\mathbf{h}_d \mid (v,d) \in \mathcal{E}_{dd}\}, \mathbf{h}_a \mid (v,a) \in \mathcal{E}_{ad}\}\Big)$$
$$= f_g\Big(\Big[\{f_a([\mathbf{h}_a, \mathbf{h}_d]) \mid (v,a) \in \mathcal{E}_{ad}, (v,d) \in \mathcal{E}_{dd}\}\Big]\Big) \quad (7)$$

where $\mathbf{h}_v$ in Equation 5 is empty since paper node contents are non-existent before paper writing; $\mathcal{E}_{ad}$ specifically refers to authorship relations between agent and data nodes, and $\mathcal{E}_{dd}$ refers to citation relations within data nodes. A visualization of Equation 7 is shown in Figure 2 "Stage 2".

▷ **Stage 3: Review writing**. The review writing task is the final stage of the automatic research simulation, serving as a reflection stage in the multi-agent research simulator. The difference between the previous 2 stages is that, first, the researchers involved during review writing are not the authors but the reviewers of the paper. Additionally, review writing is based on a written paper where $\mathbf{h}_v$ is no longer empty. Concretely, by adapting Equation 5, the TextGNN layer for review writing can be written as:

$$\mathbf{r}_v = \text{AGG}\Big(\mathbf{h}_v, , \{\mathbf{h}_d \mid (v,d) \in \mathcal{E}_{dd}\}\{f_a(\cdot), \mathbf{h}_a \mid (v,a) \in \mathcal{E}_{ad}\}\Big)$$
$$= f_g\Big(\Big[\mathbf{h}_v, \{f_a([\mathbf{h}_a, \mathbf{h}_v, \mathbf{h}_d]) \mid (v,a) \in \mathcal{E}_{ad}, (v,d) \in \mathcal{E}_{dd}\}\Big]\Big) \quad (8)$$

▷ **Summary: RESEARCHTOWN simulation algorithm**. Utilizing the community graph $\mathcal{G}$, we propose a simulation algorithm named as RESEARCHTOWN. Overall, the simulation algorithm can be considered as a 2-layer GNN where the paper reading is the first layer of information aggregation. Both paper writing and review writing are the second layer of the GNN to generate the final simulation outputs. We formally summarize the research community simulation in Algorithm 1. To achieve better efficiency, the modified version for implementation is in Appendix §D.2.

---

**Algorithm 1** RESEARCHTOWN simulation algorithm

---

**Require:** community graph $\mathcal{G}(\mathcal{V}, \mathcal{E})$,
      paper contents $\mathbf{x}_v$ for all paper nodes,
      target paper node $v$
**Ensure:** paper content $\mathbf{h}_v$ and review content $\mathbf{r}_v$ for paper node $v$
 1: **for** each $u \in \mathcal{N}(v)$ **do**
 2:    **if** $u \in \mathcal{V}_d$ **then**
 3:        $\mathbf{h}_u \leftarrow \mathbf{x}_u$
 4:    **else**
 5:        $\mathbf{h}_u \leftarrow f_u\left([\{\mathbf{x}_d \mid (u,d) \in \mathcal{E}_{ad}\}]\right)$     ▷ Eq. (6)
 6:    **end if**
 7: **end for**
 8: $\mathbf{h}_v \leftarrow f_g\Big(\Big[\{f_a([\mathbf{h}_a, \mathbf{h}_d]) \mid$
 9:    $(v,a) \in \mathcal{E}_{ad}, (v,d) \in \mathcal{E}_{dd}\}\Big]\Big)$   ▷ Eq. (7)
10: $\mathbf{r}_v \leftarrow f_g\Big(\Big[\mathbf{h}_v, \{f_a([\mathbf{h}_a, \mathbf{h}_v, \mathbf{h}_d]) \mid$
11:    $(v,a) \in \mathcal{E}_{ad}, (v,d) \in \mathcal{E}_{dd}\}\Big]\Big)$   ▷ Eq. (8)
12: **return** $\mathbf{h}_v, \mathbf{r}_v$

---

## 6. Evaluating RESEARCHTOWN via Masked Node Prediction Task

Utilizing graph structures not only enables the design of the research simulation algorithm but also provides a natural way to evaluate it. As we show next, we propose to view research evaluation as a masked node prediction task, including evaluation for both paper writing and review writing.

**Evaluation by masked node prediction**. A masked node prediction task in the community graph $\mathcal{G}$ can be defined as first masking a specific node $v \in \mathcal{V}$ in the community graph by setting its hidden states $\mathbf{h}_v = \emptyset$, where the original hidden state is saved as $\mathbf{h}_v^*$; then an ideal model should be able to predict the hidden states $\mathbf{h}_v^*$ of the masked node from its neighborhood $\mathcal{N}(v)$. Concretely, in Equation 7, the output $\mathbf{h}_v$ can be regarded as the masked node prediction for evaluation of paper writing, suppose that the node $v$ is a masked version of a ground truth data node. Similarly, in Equation 8, the output $\mathbf{r}_v$ can be regarded

as the predicted node attributes for review writing, where the original review is represented as $\mathbf{r}_v^*$. In general, we have:

$$\mathbf{h}_v, \mathbf{r}_v = \text{RESEARCHTOWN}\Big(\mathcal{G}(\mathcal{V},\mathcal{E}); \{\mathbf{x}_u \mid u \in \mathcal{N}(v)\}; v\Big) \tag{9}$$

where $\mathbf{h}_v$ is the text-form hidden states of a masked node $v$ and $\mathbf{r}_v$ is the text-form prediction output of a masked node $v$. Since we have real-world results for both paper writing and review, we treat them as ground truth even though they are not perfect because the goal of RESEARCHTOWN is to simulate the human research community rather than to find optimal solutions for papers and reviews ($\mathbf{h}_v^*$ for paper ground-truth and $\mathbf{r}_v^*$ for review ground-truth) and we can systematically evaluate both processes to check the effectiveness of our simulation algorithm. More specifically, since we have access to ground-truth papers $\mathbf{h}_v^*$ when evaluating the review writing simulation, to avoid accumulated errors, we update Equation 8 during evaluation so that reviews $\mathbf{r}_v$ are generated based on $\mathbf{h}_v^*$, instead of $\mathbf{h}_v$:

$$\mathbf{r}_v = \text{AGG}\Big(\mathbf{h}_v^*, \{\mathbf{h}_d \mid (v,d) \in \mathcal{E}_{dd}\}, \{f_a(\cdot), \mathbf{h}_a \mid (v,a) \in \mathcal{E}_{ad}\}\Big) \tag{10}$$

**Evaluation metric**. We utilize state-of-the-art embedding models like text-embedding-large-3 [1] to build distance function for $d_p(\mathbf{h}_v, \mathbf{h}_v^*)$ and $d_r(\mathbf{r}_v, \mathbf{r}_v^*)$. More details related to formal embedding-based metric definitions for paper writing and review writing tasks are available in Appendix §F.

## 7. Experimental Settings

**RESEARCHTOWN setting**. We utilize GPT-4o-mini [2] as the LLM backbone for implementing the agent functions, with the decoding temperature set to 0 to ensure reproducibility. To evaluate different aggregation strategies, we conduct experiments using specific types of nodes connected to the target node: (1) *AGG-self*, where the aggregation relies solely on the target node; (2) *AGG-agent*, which includes the target node and its neighboring agent nodes; (3) *AGG-data*, which involves the target node and its neighboring data nodes; and (4) *AGG-global*, which incorporates the target node and all its neighboring nodes, including agent and data nodes. We specifically refer to *AGG-global* as our proposed RESEARCHTOWN method for simulation, while the others serve as baselines. This experimental design enables a systematic comparison of the effects of different neighborhood information on the aggregation process. More details about different settings are available in Appendix §D.1.

**RESEARCHBENCH setting**. To evaluate RESEARCHTOWN for research simulation, we introduce RESEARCHBENCH,

Table 1: **Evaluation results for paper writing simulation.** *Hard*, *Medium*, and *Easy* correspond to three subsets of the paper writing tasks with different difficulties, while *Overall* refers to the performance across all parts. Text-embedding-large-3 is used to build embedding-based similarity metrics. Comprehensive results are available in Appendix §J.

| AGG Type | Easy ↑ | Medium ↑ | Hard ↑ | Overall ↑ |
|---|---|---|---|---|
| AGG-self | 46.42 | 45.92 | 45.90 | 46.08 |
| AGG-agent | 56.90 | 55.55 | 53.26 | 55.24 |
| AGG-data | 74.36 | 66.42 | 56.02 | 65.30 |
| AGG-global | 73.79 | 67.85 | 60.89 | 67.51 |

Table 2: **Evaluation results for review writing simulation.** For *strength* and *weakness*, it shows embedding-based similarity results. We use text-embedding-large-3 as embedding models and select 5 reviewers for running *AGG-agent* and *AGG-global*. $\Delta\mathbf{S}$ refers to the average difference of review scores between real-world ones and generated ones. $\bar{\mathbf{S}}$ refers to the average scores of generated ones. Comprehensive results are available in Appendix §J.

| AGG Type | Strength ↑ | Weakness ↑ | $\Delta\mathbf{S}$ ↓ | $\bar{\mathbf{S}}$ |
|---|---|---|---|---|
| AGG-self | 51.23 | 47.16 | 1.27 | 5.33 |
| AGG-agent | 51.66 | 46.75 | 1.19 | 5.40 |
| AGG-data | 51.45 | 47.62 | 1.26 | 5.30 |
| AGG-global | 51.51 | 47.17 | 1.55 | 5.00 |

which consists of 1,000 paper writing tasks and 200 review writing tasks. All tasks are sourced from recent top-tier machine learning conferences such as NeurIPS 2024 [3] and ICLR 2024 [4]. Since most papers are released after the cutoff date of GPT-4o-mini, information leakage is not considered an issue. For paper writing tasks, we categorize them into three difficulty levels—*hard* (333 tasks), *medium* (334 tasks), and *easy* (333 tasks)—based on the similarity results of data-only aggregation. Specifically, for review writing tasks, the reviewers prepared for each paper are selected from the top 5 researchers most related to the paper, as reviewer information is not publicly available in the real world. More details about the data collection and prevention of information leakage during simulation are in Appendix §E.

## 8. Core Results: In-distribution Evaluation

In this section, we present the main results of our research simulation on RESEARCHBENCH, including 1,000 paper writing tasks and 200 review writing tasks. We evaluate existing paper nodes that have fully known their content and their neighborhoods within the community graph. We refer to these scenarios as *in-distribution* cases.

---

[1] https://openai.com/index/new-embedding-models-and-api-updates/

[2] We point to GPT-4o-mini-2024-07-18 for use.

[3] https://neurips.cc/Conferences/2024

[4] https://openreview.net/group?id=ICLR.cc/2024/Conference

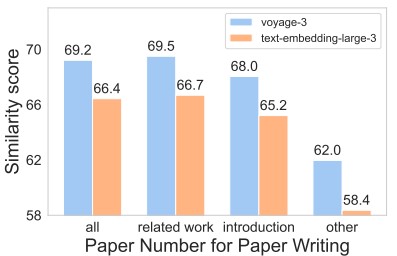
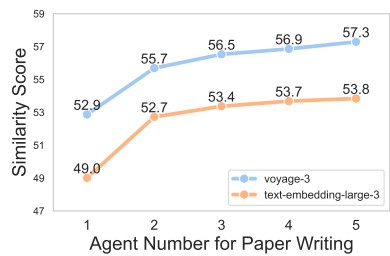
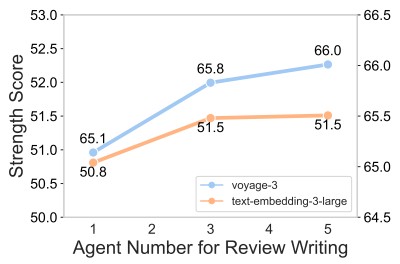

**Figure 3: Ablation study on paper number**. We select different sub-parts of cited papers in paper writing tasks.

**Figure 4: Ablation study on agent number**. We select different numbers of agents for paper writing tasks.

**Figure 5: Ablation study on agent number**. We select different numbers of agents for review writing tasks.

**Overall: RESEARCHTOWN can provide a realistic simulation of research activity**. To evaluate research simulation, we utilize state-of-the-art embedding models (text-embedding-3-large) to compare the semantic similarity between simulated results and real-world results. For paper writing, as shown in Table 1, the overall similarity score obtained using text-embedding-3-large across 1,000 papers is 67.51. Notably, the score increases to 73.79 for an easy subset of the benchmark. These results demonstrate that paper writing with RESEARCHTOWN can produce realistic outputs compared to real-world ones. Moreover, it indicates that some ideas in top-tier conference papers are not hard to think of and can be imagined by LLMs. For review writing, as shown in Table 2, the similarity scores are generally lower compared with paper writing, with strength-related scores averaging around 51 and weakness-related scores averaging around 47. This suggests that review writing is more challenging to generate with RESEARCHTOWN, particularly for weakness identification. A possible explanation is that real-world review data is often noisier and more diverse, making it harder to simulate accurately.

**Paper writing: participation of multi-researchers improves paper quality**. As shown in Table 1, cited papers contribute more effectively than authors in the paper writing simulation, with data-aggregation achieving a score of 65.30 compared to 55.24 for agent-aggregation. The best results are obtained by combining both, surpassing data aggregation by 2.21 points. Researchers are particularly beneficial under difficult scenarios, improving the text-embedding-large-3 score from 56.02 to 60.89, likely due to the inclusion of multi-hop paper information from researchers.

**Review writing: participation of multi-reviewers improves review quality**. Unlike paper writing, review writing mainly relies on the paper that needs to be reviewed, making reviewers and cited papers less impactful, with differences limited to within 1 point. However, as shown in Table 2, adding additional information consistently improves performance over the self-aggregation baseline. Agent aggregation performs best for writing strengths and assigning scores, while data aggregation achieves the best results for

writing weaknesses. This pattern likely reflects the role of related work comparisons in highlighting weaknesses, while multiple reviewers help provide a more balanced assessment of strengths. Interestingly, global aggregation leads to larger differences in scores. We consider it an exception since GPT-4o-mini tends to apply stricter novelty judgments under global aggregation—its average assigned score drops from 5.3 to 5.0. As shown in Table 3, this effect is not observed for Qwen-2.5-7B-Instruct or Deepseek-v3, which gain better results with global aggregation.

## 9. Ablation Study: RESEARCHTOWN is Robust

We conduct ablation studies on both hyperparameters and model selection. The results show that RESEARCHTOWN consistently produces high-quality simulations across a range of settings, demonstrating strong robustness. Detailed experimental configurations are provided in Appendix §I.

**Ablation on paper number**. In paper writing tasks, users can freely assign papers to simulate non-existent work, making robustness to the number of papers essential. As shown in Figure 3, papers cited in the related work section have the greatest positive impact, increasing the similarity score from 66.4 to 66.7 compared to using all papers. In contrast, using only papers cited in the introduction lowers the score to 65.2, while including papers from other sections reduces it further to 58.4. These results highlight the importance of selecting informative references when generating papers. In review writing, the number of papers is fixed, so no ablation study on the paper number is applicable.

**Ablation on agent number**. For RESEARCHTOWN simulation, users can assign different numbers of agents, making robustness to agent number critical for RESEARCHTOWN. In Figure 4, in the paper writing task, increasing the agent number improves simulation quality under the agent-aggregation setting. The most notable gain occurs when increasing from 1 to 2, boosting the similarity score from 49.0 to 52.7. Similar trends hold in review writing (Figure 5), where increasing the agent number consistently enhances output quality. The strength score improves from 50.8 to 51.5 when increasing the reviewer from 1 to 5.

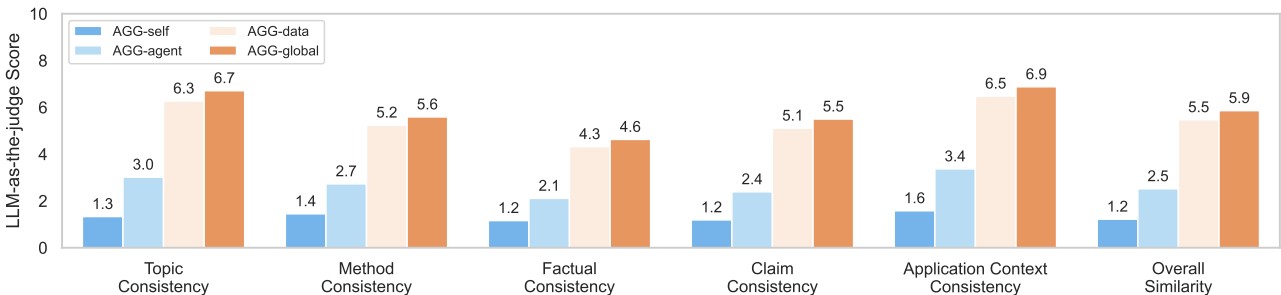

Figure 6: **Fine-grained similarity evaluation using LLM-as-a-judge for paper writing simulation.** We use GPT-4o as the evaluator, prompting it to score each dimension on a scale from 0 to 10. The first five dimensions assess specific aspects of similarity, while the final score (*overall similarity*) represents an overall score as judged by the LLM.

**Ablation on generation models**. The choice of LLMs significantly impacts simulation quality. In addition to GPT-4o-mini, we evaluate two models from different families: Qwen-2.5-7B-Instruct[1] and Deepseek-v3[2]. In Table 3, for both paper writing tasks, global aggregation (RESEARCHTOWN) consistently yields the highest similarity scores across all models. It also achieves the best review difference scores for Qwen-2.5-7B-Instruct and Deepseek-v3. The only exception is GPT-4o-mini, which shows an unexpected increase in review difference under AGG-global. Overall, Deepseek-v3 outperforms GPT-4o-mini, which in turn outperforms Qwen-2.5-7B-Instruct—consistent with their relative performance on other tasks.

**Ablation on embedding models**. Similarity scores can be computed using different models, and voyage-3[3] serves as an alternative to the text-embedding-3-large used in our main experiments. As shown in Figures 3, 4, and 5, voyage-3 produces consistent trends in ablation studies involving the number of papers and agents. This consistency suggests that RESEARCHTOWN is robust to the choice of embedding model, and different models lead to the same conclusions.

## 10. Discussion: RESEARCHTOWN is Effective

Besides computing embedding-based similarities, we provide more types of evaluations here. First, we prompt LLMs to calculate fine-grained similarity scores that assess consistency between real-world data and simulated ones across various dimensions. Next, we evaluate the intrinsic quality of the simulated outputs themselves and compare them with real-world data. Finally, we report results from human evaluations to validate the alignment between LLM-based evaluation and human judgments. More details about LLM-based evaluation are available in Appendix §G, and details about human evaluation are available in Appendix §H.

---

[1] https://huggingface.co/Qwen/Qwen2.5-7B-Instruct

[2] We point DeepSeek-V3-0324 for use.

[3] https://blog.voyageai.com/2024/09/18/voyage-3/

Table 3: **Comparison of simulation results with different generation models.** For *Qwen*, we refer to Qwen-2.5-7B-Instruct. For *GPT*, we refer to GPT-4o-mini. For *DS*, we refer to Deepseek-v3. For paper writing metrics, we utilize the overall similarity. For review writing metrics, we use $\Delta \mathbf{S}$ to represent its review alignment with the real world.

| AGG Type | Paper Writing | | | Review Writing | | |
|---|---|---|---|---|---|---|
| | Qwen | GPT | DS | Qwen | GPT | DS |
| AGG-self | 46.45 | 46.08 | 48.62 | 1.36 | 1.27 | 1.11 |
| AGG-agent | 53.91 | 55.24 | 56.19 | 1.41 | 1.19 | 1.05 |
| AGG-data | 65.03 | 65.30 | 65.05 | 1.28 | 1.26 | 1.07 |
| AGG-global | 65.30 | 67.51 | 65.33 | 0.79 | 1.51 | 0.81 |

**Automatic evaluation on fine-grained similarity**. A high cosine similarity score alone can mask important issues in simulated results. To capture a more complete picture of similarity, we move beyond a single score and instead evaluate across five fine-grained dimensions: *topic consistency*, *method consistency*, *factual consistency*, *claim consistency*, and *application context consistency*. These dimensions collectively reflect subcomponents of overall semantic similarity. For evaluation, we use GPT-4o to assign scores from 0 to 10 for each dimension for each paper. As shown in Figure 6, our proposed global aggregation method (RESEARCHTOWN) consistently outperforms all other aggregation baselines across these dimensions. This demonstrates that RESEARCHTOWN provides a more effective simulation of research activities compared to baselines.

**Automatic evaluation on intrinsic quality**. In addition to evaluating semantic similarity between simulated and real-world data, we also assess the intrinsic quality of the generated content. Specifically, we focus on two key dimensions: *novelty* and *feasibility*, which we consider the two most critical aspects of a research proposal. As shown in Table 4, the simulated outputs still do not match the novelty and feasibility levels of real-world articles but are close to those. This gap indicates that RESEARCHTOWN would benefit from a more coordinated agentic workflow to enhance the quality of the generated research outputs.

---

**NLP + Astronomy**

**What is the problem?**
How can kinematic modeling techniques from astrophysics be applied to analyze and understand the evolution of writing styles in large-scale linguistic datasets?

**What are the key components of my approach and results?**
…We will utilize *large-scale corpora from diverse language families*, focusing on historical texts to capture changes in syntax, vocabulary, and stylistic elements over time. Specifically, we will adapt *kinematic models* by defining linguistic "velocity" metrics that quantify shifts in writing style, analogous to how velocities are analyzed in astrophysics…

**NLP + Criminology**

**What is the problem?**
How can a multimodal LLM that integrates qualitative narrative analysis with real-time speech translation effectively address the communication needs of communities affected by mass incarceration?

**What are the key components of my approach and results?**
…I will employ a mixed-methods approach, combining qualitative data from interviews with *impacted individuals* and quantitative data from existing linguistic resources. The model will be trained on a diverse dataset that captures a wide range of narratives related to *mass incarceration*, ensuring representation of various socio-cultural contexts.…

Figure 7: **Examples of generated interdisciplinary research papers from RESEARCHTOWN.** For each example, we include RESEARCHTOWN's responses to two questions: "*What is the problem?*" and "*What are the key components of my approach and results?*" as these are the most critical among the five questions mentioned in Appendix §F. Appendix §K provides the full contents of the above two and more examples for interdisciplinary research.

Table 4: **Evaluation results on novelty and feasibility.** Each paper is assigned scores from 0 to 10 for novelty and feasibility. Both LLM-based evaluation and human evaluations are conducted to evaluate the quality of simulated papers. LLM-based evaluation includes results on 1,000 papers, and human evaluation includes results on 40 of them. *Simulation* represents the outputs of RESEARCHTOWN, *real-world* represents the existing papers.

| Evaluation | Simulation | | Real-world | |
|---|---|---|---|---|
| | **Novelty** | **Feasibility** | **Novelty** | **Feasibility** |
| LLM-based | 7.39 | 6.82 | 7.85 | 7.13 |
| Human-based | 5.50 | 7.98 | 5.90 | 7.85 |

**Human evaluation**. Evaluation based on LLMs may introduce bias into the results. To validate the reliability of LLM-based evaluations, we conduct additional human evaluations. For similarity-based assessments, human judgments correlate well with LLM scores, achieving a Pearson correlation of 0.61, indicating reasonable agreement. However, for intrinsic quality evaluations, the correlation between human and LLM scores is low. This is likely due to the inherent ambiguity of such tasks and the need for domain-specific expertise. Despite this, both human and LLM evaluations consistently indicate that simulated papers are slightly less novel than real-world ones—though the gap is relatively small (5.50 vs 5.90 for humans and 7.39 vs 7.85 for LLMs).

## 11. Case Study: Out-of-distribution Use

As discussed in Section §8, the node masking evaluation in RESEARCHTOWN targets *in-distribution* settings with predefined neighborhoods. In real-world use, however, RESEARCHTOWN must generate non-existing papers and reviews without such neighborhoods, requiring automatic construction via paper–researcher matching. This leads to *out-of-distribution* cases, such as interdisciplinary research, where unrelated papers and researchers form unconventional neighborhoods without prior related works.

**RESEARCHTOWN can inspire interdisciplinary research**. Interdisciplinary research is often challenging due to limited collaboration across fields. RESEARCHTOWN addresses this by enabling agents with diverse expertise to read, interact, and co-create novel ideas. For example, as shown in Figure 7, combining NLP and astronomy papers leads to using kinematic models to analyze language evolution, while linking NLP and criminology inspires the use of LLMs to support communities affected by mass incarceration. These domain pairings are rarely explored in existing literature, demonstrating RESEARCHTOWN 's ability to generate innovative, cross-disciplinary research directions.

**RESEARCHTOWN-written contents might have limited use in the real world**. RESEARCHTOWN exhibits failure modes when combining too many disparate domains, often producing incoherent or superficial outputs. For example, combining researchers and papers from LLM, biology, criminology, and astronomy, RESEARCHTOWN generates a research question of "*How does coded language in political discourse influence societal biases, and how can a Bayesian hierarchical model be employed to analyze this effect while simultaneously addressing observational biases in white dwarf population studies?*" It simply strings together terminology from different domains without presenting a clear research direction. Such vagueness might hinder the real use of the papers simulated from RESEARCHTOWN.

## 12. Conclusion

We introduce RESEARCHTOWN, a graph-based multi-agent framework that simulates research communities by modeling them as heterogeneous graphs. RESEARCHTOWN integrates key research activities—paper reading, writing, and reviewing—into a unified TextGNN-driven inference process. It enables realistic and robust simulations through agent collaboration and facilitates rare interdisciplinary interactions. RESEARCHTOWN offers a valuable platform for studying research dynamics and developing algorithms to support automated scientific discovery.

## Impact Statement

RESEARCHTOWN presents an LLM-based simulation framework that models human research communities as graph-based multi-agent systems, enabling the study of collaboration, knowledge diffusion, and institutional dynamics. By formalizing how agents create, refine, and evaluate academic papers, the simulator can inform the design of autonomous research systems that assist, rather than replace, human researchers. Potential applications include optimizing collaboration structures, identifying systemic bottlenecks in peer review or discovery, and stress-testing scientific workflows under various incentive and communication settings. While the framework is primarily a research tool, we acknowledge that future extensions involving autonomous agents could raise ethical considerations around authorship, influence, and epistemic trust. Our work highlights the imperative for adaptive ethical frameworks that keep pace with technological capabilities while protecting scholarly values.

## Acknowledgments

We sincerely appreciate the support from Amazon grant funding project #120359, "GRAG: Enhance RAG Applications with Graph-structured Knowledge", and Meta gift funding project "PERM: Toward Parameter Efficient Foundation Models for Recommenders".

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

## A. Individual Contribution

| | |
|---|---|
| **Haofei Yu** | Overall project leader |
| **Zhaochen Hong** | Co-lead, code writing, benchmark collection, review writing experiment |
| **Zirui Cheng** | Co-lead, paper writing, code writing, system design |
| **Kunlun Zhu** | Co-lead, benchmark collection, code writing, paper writing, experiment |
| **Keyang Xuan** | Participant, code writing, benchmark collection, case study |
| **Jinwei Yao** | Participant, code writing, evaluation experiment in early versions |
| **Tao Feng** | Co-lead in early versions, paper writing, code writing in early versions |
| **Jiaxuan You** | Overall project advisor |

## B. Ethical Concern

The development and deployment of RESEARCHTOWN raises several important ethical considerations that we have carefully addressed in our work. We first discuss how RESEARCHTOWN prevents dangerous use, including facilitating plagiarism, producing misleading or low-quality claims, and role-playing human researchers. Furthermore, we discuss the attribution and authorship issues for generated content and discuss the model and data license in our work.

### B.1. Potential to facilitate plagiarism

Generative AI's capabilities for image and text generation can potentially lead to plagiarism in research (Elali & Rachid, 2023). To address this, we have implemented safeguards to ensure responsible usage. RESEARCHTOWN is designed as an assistive tool to help researchers gather inspiration for papers and review writing, rather than generating complete, ready-to-use content. By design, RESEARCHTOWN ensures that its outputs serve as a starting point for further intellectual effort, rather than a replacement for human researchers.

For generated papers, RESEARCHTOWN provides only preliminary answers to five key research questions. These outputs are intentionally incomplete and generic, requiring significant refinement and further development by the user. Critical sections such as the introduction, background, methodology, discussion, and conclusion are not included, placing the responsibility for completing and validating the content on the researcher.

For generated reviews, RESEARCHTOWN provides general guidance on potential strengths and weaknesses, accompanied by an indicative score for reference. However, these reviews are intentionally non-definitive and generic, only as a supplementary aid to help reviewers organize their thoughts. Generated reviews do not replace human judgment in determining the acceptance or rejection of a paper. The final evaluation, including critical reasoning, detailed feedback, and the ultimate decision, remains the sole responsibility of the reviewer. Reviewers must ensure fairness, accuracy, and rigor, using AI outputs only as a starting point to enhance their assessment process.

### B.2. Potential to produce misleading or low-quality claims

The motivation of our paper is to simulate research activities and generate preliminary research progress (*e.g.*, papers and reviews that are in their condensed bullet-point summarized format) that can be scrutinized and validated by human researchers, ultimately contributing to the acceleration of the research process. We acknowledge that AI-generated ideas may vary in quality, and therefore, these outputs are not intended for direct dissemination. Instead, they serve as initial, unofficial suggestions that require further experimental validation by human researchers. This approach ensures that only rigorously tested and verified research is presented as final, high-quality work.

### B.3. Potential to role-play human researchers

The primary objective of our work is to leverage existing academic literature to simulate research activities. In this paper, our research agents are designed to act as research domain experts, generating informative and relevant content based on a given and limited research domain. Importantly, we do not aim to simulate human-like interactive dialogues between research agents, nor do we attempt to mimic the specific research styles of individual human researchers. Instead, we focus on using related academic papers as conditions for generating more related research content.

The research agents are built using publicly available, properly cited academic papers, which eliminates the need for additional consent. We utilize the LLM-based research agents, each with one or more specific research domains, modeling

the typical academic process, where researchers read, synthesize, and build upon available public academic data. By focusing on publicly available research papers, we align with the papers' intended purpose: contributing to the collective advancement of knowledge and fostering academic growth.

### B.4. Attribution and authorship

The AI-generated content, such as papers, reviews, or other research outputs, is meant for internal discussion and as a reference to assist human researchers. These outputs are not intended for direct publication. Our proposed methods serve as tools to accelerate the research process by offering starting points that require further elaboration, critical analysis, and human refinement to reach a publishable standard. The final authority to complete and submit research lies solely with human authors, ensuring that full responsibility and ownership remain with them. Since the AI-generated content is not considered complete or officially authored, it does not raise issues of authorship or attribution.

## C. Artifact

We list all licenses for the data and models used in our paper in this section.

### C.1. Data license

All papers in RESEARCHBENCH come from top-tier machine learning conferences (ICLR 2024 and NeurIPS 2024). These papers are publicly available and under the license of CC-BY 4.0, allowing for redistribution and sharing. For the evaluation results of RESEARCHBENCH, all inputs and outputs are logged and open for access. Additionally, we keep an accessible record of all supplementary papers referenced during RESEARCHTOWN's inference process. All outputs from RESEARCHTOWN are released under the licenses of the papers used for generation.

### C.2. Model license

Our work relies on multiple foundation models, including GPT-4o-mini, Qwen-2.5-7B-Instruct, Deepseek-v3, text-embedding-3-large, and voyage-3. Specifically, we use `gpt-4o-mini-2024-07-18` accessed via the OpenAI API. We use `Qwen-2.5-7B-Instruct-Turbo` and `Deepseek-v3-0324` via the together.ai [1] inference API. We utilize the official inference API provided by OpenAI and VoyageAI to use text-embedding-3-large and voyage-3 separately.

The GPT-4o-mini, text-embedding-3-large, and voyage-3 models are closed-source and operate under proprietary licenses. We use them only for academic, and non-commercial purposes and ensure all inputs come from publicly available data, complying with their usage restrictions. By contrast, Qwen-2.5-7B-Instruct is released under the permissive Apache 2.0 license, and Deepseek-v3-0324 is available under the MIT License, allowing for broad academic and research use. We make no modifications to these models and use them as-is via their public APIs.

## D. RESEARCHTOWN Details

In this section, we provide more explanation and implementation details for RESEARCHTOWN simulation algorithm. To achieve better performance and efficiency, we design different prompts for each agent function and make the aggregation process run in parallel.

### D.1. RESEARCHTOWN aggregation setting implementation

We provide more information about the 4 aggregation experimental settings mentioned in Section §7 (*i.e.* self-agg, agent-agg, data-agg, global-agg). The main difference lies in the neighborhood nodes that participated during the message-passing process.

**AGG-self**. We do not rely on any neighborhood information.

▷ Paper writing: The LLM agent without profiles brainstorms independently without referencing any external data or other

---

[1] https://www.together.ai/inference

agents' ideas. This setting extends Equation 7 to

$$\mathbf{h}_v = \text{AGG}(\emptyset, \emptyset, \emptyset) = f_g(\emptyset) \tag{11}$$

▷ Review writing: The LLM agent without profiles writes the review based solely on the paper itself, without considering additional references or other agents. This setting extends Equation 8 to

$$\mathbf{r}_v = \text{AGG}(\mathbf{h}_v, \emptyset, \emptyset) = f_g(\mathbf{h}_v) \tag{12}$$

**AGG-agent**. We rely only on agent nodes and exclude data nodes.

▷ Paper writing: Multiple LLM agents collaborate by sharing their content and insights to produce the final paper's content. This setting extends Equation 7 to

$$\begin{aligned}
\mathbf{h}_v &= \text{AGG}\big(\mathbf{h}_v, \{f_a(\cdot), \mathbf{h}_a \mid (v, a) \in \mathcal{E}_{ad}\}, \emptyset\big) \\
&= f_g\big([f_a(\mathbf{h}_a) \mid (v, a) \in \mathcal{E}_{ad}]\big)
\end{aligned} \tag{13}$$

▷ Review writing: Multiple LLM agents collectively review the paper, sharing their input and critiques to form the final review. This setting extends Equation 8 to

$$\begin{aligned}
\mathbf{r}_v &= \text{AGG}\big(\mathbf{h}_v, \{f_a(\cdot), \mathbf{h}_a \mid (v, a) \in \mathcal{E}_{ad}\}, \emptyset\big) \\
&= f_g\big([\mathbf{h}_v, f_a([\mathbf{h}_a, \mathbf{h}_v]) \mid (v, a) \in \mathcal{E}_{ad}]\big)
\end{aligned} \tag{14}$$

**AGG-data**. We rely only on data nodes and exclude agent nodes.

▷ Paper writing: A single LLM agent without profiles reads and synthesizes information from related data sources to write a paper. This setting extends Equation 7 to

$$\begin{aligned}
\mathbf{h}_v &= \text{AGG}\big(\mathbf{h}_v, \emptyset, \{\mathbf{h}_d \mid (v, d) \in \mathcal{E}_{dd}\}\big) \\
&= f_g\big(\{\mathbf{h}_d \mid (v, d) \in \mathcal{E}_{dd}\}\big)
\end{aligned} \tag{15}$$

▷ Review writing: A single LLM agent without profiles produces a review by reading both the paper and its related data sources, integrating the information to form a comprehensive critique. This setting extends Equation 8 to

$$\begin{aligned}
\mathbf{r}_v &= \text{AGG}\big(\mathbf{h}_v, \emptyset, \{\mathbf{h}_d \mid (v, d) \in \mathcal{E}_{dd}\}\big) \\
&= f_g\big([\mathbf{h}_v, \{\mathbf{h}_d \mid (v, d) \in \mathcal{E}_{dd}\}]\big)
\end{aligned} \tag{16}$$

**AGG-global**. We include all neighborhood nodes (both agent and data) during aggregation.

▷ Paper writing: Multiple LLM agents produce content in parallel while referencing various data sources. Their aggregated outputs, which incorporate insights from both other agents and data, form the final paper. This setting extends Equation 7 to

$$\begin{aligned}
\mathbf{h}_v &= \text{AGG}\big(\mathbf{h}_v, \{f_a(\cdot), \mathbf{h}_a \mid (v, a) \in \mathcal{E}_{ad}\}, \{\mathbf{h}_d \mid (v, d) \in \mathcal{E}_{dd}\}\big) \\
&= f_g\big([\{f_a([\mathbf{h}_a, \mathbf{h}_d]) \mid (v, a) \in \mathcal{E}_{ad}, (v, d) \in \mathcal{E}_{dd}\}]\big)
\end{aligned} \tag{17}$$

▷ Review writing: Multiple LLM agents each consider the paper and its related works to write their review. The final review is a combination of these integrated perspectives. This setting extends Equation 8 to

$$\begin{aligned}
\mathbf{r}_v &= \text{AGG}\big(\mathbf{h}_v, \{f_a(\cdot), \mathbf{h}_a \mid (v, a) \in \mathcal{E}_{ad}\}, \{\mathbf{h}_d \mid (v, d) \in \mathcal{E}_{dd}\}\big) \\
&= f_g\big([\mathbf{h}_v, \{f_a([\mathbf{h}_a, \mathbf{h}_v, \mathbf{h}_d]) \mid (v, a) \in \mathcal{E}_{ad}, (v, d) \in \mathcal{E}_{dd}\}]\big)
\end{aligned} \tag{18}$$

### D.2. RESEARCHTOWN simulation algorithm implementation

One practical issue with Equation 4 and Equation 5 is that $f_a(\cdot)$ must be computed for every combination of agent and data nodes, leading to a significant computational burden if implemented directly as defined. We introduce the definition of $f_a(\cdot)$ to maintain scalability and alignment with traditional Graph Neural Network definitions. In practice, we can easily

parallelize $f_a(\cdot)$ over all data nodes, which means we can prompt once and put all data nodes' information in one prompt to get the results instead of repeating the prompting process on each data node. Thus, we provide a modified version of the original aggregation process below.

While the paper reading process remains the same for implementation, the paper writing process can be alternatively calculated as: This setting extends Equation 7 to

$$
\begin{aligned}
\mathbf{h}_v &= \text{AGG}\big(\emptyset, \{f_a(\cdot), \mathbf{h}_a \mid (v,a) \in \mathcal{E}_{ad}\}, \{\mathbf{h}_d \mid (v,d) \in \mathcal{E}_{dd}\}\big) \\
&= f_g\left(\left[\left\{f_a\big([\mathbf{h}_a, \{\mathbf{h}_d \mid (v,a) \in \mathcal{E}_{dd}\}]\big) \mid (v,a) \in \mathcal{E}_{ad}\right\}\right]\right)
\end{aligned}
\tag{19}
$$

Similarly, the review writing process can be calculated as: This setting extends Equation 8 to

$$
\begin{aligned}
\mathbf{r}_v &= \text{AGG}\big(\mathbf{h}_v, \{f_a(\cdot), \mathbf{h}_a \mid (v,a) \in \mathcal{E}_{ad}\}, \{\mathbf{h}_d \mid (v,d) \in \mathcal{E}_{dd}\}\big) \\
&= f_g\left(\left[\mathbf{h}_v, \left\{f_a\big([\mathbf{h}_a, \mathbf{h}_v, \{\mathbf{h}_d \mid (v,d) \in \mathcal{E}_{dd}\}]\big) \mid (v,a) \in \mathcal{E}_{ad}\right\}\right]\right)
\end{aligned}
\tag{20}
$$

Therefore, we reduce the calling of $f_a(\cdot)$ from $N \times M$ to $M$ where $M$ represents the number of agent nodes in the neighborhoods and $N$ represents the number of data nodes in the neighborhoods.

### D.3. RESEARCHTOWN agent function implementation

For each $f_a(\cdot)$ and $f_g(\cdot)$ in Algorithm 1, these functions represent LLMs equipped with task-specific prompt templates. In the *global-agg* setting, we provide examples of the prompt templates for each agent function. Other settings follow a similar style but use fewer details.

For the paper writing stage, Table 7 presents the $f_u(\cdot)$ prompt template used in it. During the paper writing stage, Table 8 and Table 9 show the prompt templates for $f_a(\cdot)$ and $f_g(\cdot)$ respectively.

For the review writing stage, since we need to separately generate strengths, weaknesses, and scores, $f_a(\cdot)$ combines the prompt templates from Table 10, Table 11, and Table 12. Similarly, $f_g(\cdot)$ is formed by combining the prompt templates from Table 13 and Table 14.

The aggregation function for classical GNN in Eq 1 and Eq 2, which is often a pooling or mean operation, is used to condense all neighborhood information into one embedding with the same size as the input. Similarly, our TextGNN layers in Eq 4 and Eq 5, act as an aggregation function similar to classical GNN, producing outputs with controlled textual formats and similar lengths with updated information in the neighborhood nodes by summarizing with LLMs. Therefore, the output length of multiple layers of TextGNN would not increase but would remain approximately the same. We achieve such length control in TextGNN via format control in prompting. We specifically designed prompts to ensure each output adheres to pre-defined constraints. These prompt-controlled constraints ensure stable output lengths at every TextGNN layer, avoiding text length inflation with increasing depth. Each aggregation step condenses and prioritizes critical information, effectively filtering less relevant details.

### D.4. RESEARCHTOWN future application

Any research-related content—*e.g.*, images, codebases, models, or social media posts—can be represented as nodes in the agent-data graph, with edge types like "cite the paper," "release model," or "comment on X post" (examples in Figure 1) defining interactions. By specifying appropriate edge types and agent functions, the framework can be extended to simulate tasks such as code writing, model release, panel discussions, or lectures. While we focus on paper and review writing due to their importance, available real-world data, and simplicity, the framework supports broader applications.

Additionally, RESEARCHTOWN can be extended to model social dynamics such as peer pressure, collaborations, and institutional roles via agent-agent relationship edges. Our current implementation already includes role-based dynamics (*e.g.*, leader vs. participant), and we plan to support richer simulations of institutional and reputational factors in future work.

## E. RESEARCHBENCH Details

In this section, we provide the technical details included in the construction process of RESEARCHBENCH. We describe the methodologies used for data collection across its three main components, and we name them as: (1) PAPERBENCH, (2)

HIGHIMPACTPAPERBENCH, and (3) REVIEWBENCH. Statistically, PAPERBENCH and HIGHIMPACTPAPERBENCH focus on a paper writing simulation, which contains 1,000 and 100 tasks, respectively. REVIEWBENCH focuses on review writing simulation and includes 200 tasks.

### E.1. Data collection details

We first include technical details related to how we collect paper, author, and review data from publicly available platforms as a source to build RESEARCHBENCH.

**Paper data collection**. We begin by recording the titles of all papers that we plan to crawl. Then, using the `arxiv` Python package[1], we query the arXiv API to check for any papers with identical titles. If a match is found, we note the corresponding arXiv ID and use the API to retrieve the paper's metadata, including its title, arXiv ID, author list, abstract, and citation information.

**Author data collection**. A primary challenge in collecting author data is that there might be multiple human researchers with the same name, and some human researchers may not have any publicly available publication records on public platforms, including arXiv, Google Scholar, or Semantic Scholar. As a result, at the paper collection stage, we only have each author's name. We use the `semanticscholar` Python package[2] to search for the author by name, verify that they have contributed to the specific target paper, and obtain a unique author ID from Semantic Scholar. This ID then allows us to retrieve their available publication information. To prevent information leakage when simulating paper writing and review scenarios, we exclude any of the author's publications released after the target paper's publication year. For example, if we aim to simulate a paper published in 2022, we ignore all of the author's publications appearing after 2022. We also exclude the target paper itself to avoid leaking information. Generally, we limit the maximum number of collected publications to around 20, focusing on those most relevant to the target time frame. Additionally, we gather each author's co-author network and their top publications to enrich the dataset with useful relational information.

**Review data collection**. In addition to paper and author data, we also leverage OpenReview to extract public review information. Since fully public review data is predominantly available for ICLR, we focus on collecting reviews from ICLR2024. Using the `openreview` Python package[3], we first verify the arXiv ID to ensure that we are retrieving the correct paper and its corresponding reviews. The collected review data aligns with ICLR's criteria, including detailed feedback on soundness, presentation, contributions, reviewer scores, and commentary on strengths and weaknesses. We adopt this review structure when generating our reviews, incorporating strengths, weaknesses, and ratings for the paper.

### E.2. PAPERBENCH details

PAPERBENCH is designed to evaluate the effectiveness of paper-writing simulations by gathering high-quality paper metadata from top-tier ML conferences, such as NeurIPS 2024 and ICLR 2024. Both NeurIPS 2024 and ICLR 2024 post-date beyond GPT-4o-mini's October 2023 knowledge cutoff. Thus, data leakage is not a concern. We also mask the full text during the simulation to avoid accidental exposure. Based on the collected author and paper data, we perform the following two post-processing steps:

First, we address cases where authors have no accessible publications beyond the current paper or where citation data extraction fails due to API issues. In such cases, we exclude these papers. We only retain those with full author publication information, as well as complete metadata including introduction, abstract, title, and citations. After this filtering step, we end up with approximately 1,200 papers, and then randomly sample 1,000 from them.

Second, to allow more fine-grained analysis, we split these 1,000 paper-writing tasks into three subgroups based on their difficulty level. We use the *data-agg* settings described in Section §7 to obtain results and compute similarity scores for our simulations. We then divide the dataset into three equal subsets: the worst 333 data points (hard), the middle 334 data points (medium), and the top 333 data points (easy). This results in a more granular categorization of the dataset's difficulty.

Intuitively, papers in the hard sub-part tend to be more theoretical and math-focused, while those in the easy sub-part are more application-oriented. Examples for hard sub-parts of the dataset include "*Stochastic Optimal Control Matching*" (Domingo-Enrich et al., 2023), "*Mixed Dynamics In Linear Networks: Unifying the Lazy and Active Regimes*" (Tu et al., 2024), and

---

[1] https://pypi.org/project/arxiv/
[2] https://github.com/danielnsilva/semanticscholar
[3] https://openreview-py.readthedocs.io/en/latest/

"*Multistable Shape from Shading Emerges from Patch Diffusion*" (Han et al., 2024). Examples for easy sub-parts of the dataset include "*4Real: Towards Photorealistic 4D Scene Generation via Video Diffusion Models*" (Yu et al., 2024), "*Skill Machines: Temporal Logic Skill Composition in Reinforcement Learning*" (Tasse et al., 2024), and "*On the Worst Prompt Performance of Large Language Models*" (Cao et al., 2024).

### E.3. REVIEWBENCH details

Since public review data is only fully accessible from ICLR, we focus on collecting review data for the ICLR 2024 papers included in PAPERBENCH. All reviews are anonymous, so no direct reviewer information is available. To address this, we identify suitable reviewers by first summarizing each researcher's publications. We then use the abstract of the target paper as a query and the researcher profiles as documents for a ranking task with the voyage-3 model. All authors included in RESEARCHBENCH serve as the corpus for retrieval. The top 20 most relevant authors, excluding the paper's authors, become the suitable reviewer candidates.

After obtaining the reviewer, paper, and author data, we filter out any papers lacking valid reviews during crawling. From the remaining set, we randomly select 200 reviews, each corresponding to one paper as REVIEWBENCH.

### E.4. HIGHIMPACTPAPERBENCH details

HIGHIMPACTPAPERBENCH serves as an extreme benchmark for RESEARCHTOWN, focused on simulating impactful research. We begin by collecting the 20 most-cited papers from each of 10 leading AI-related conferences—CVPR, ECCV, NeurIPS, ICLR, ICML, AAAI, IJCAI, ACL, EMNLP, and NAACL—based on Google Scholar citation rankings.[1] Additionally, we include classic machine learning algorithm papers such as those introducing VAE (Kingma, 2013), GAN (Goodfellow et al., 2014), and Adam (Kingma, 2014), each with over 1,000 citations, even if they are no longer listed in the current Google Scholar citation rankings.

For these impactful papers, it is crucial to prevent the inclusion of publications released after their publication year when gathering authors' publication data. Later works such as these could significantly alter the trajectory of the researcher, misrepresent the historical context of these influential contributions, and leak information for simulation. After collecting paper and author data, we remove any papers with incomplete information due to crawling errors. From the remaining set, we randomly sample 100 papers to form the final benchmark. These selected papers have averaged over 100 citations in the past five years, ensuring that HIGHIMPACTPAPERBENCH represents a collection of influential and well-established research.

The motivation for using impactful papers as evaluation is to use them as an extreme-case test for idea simulation. While some may exist in the LLM's training data, this benchmark is separate from our main results and serves to explore how LLMs handle well-known concepts. Our similarity analysis shows that 55% of generated papers score between 0.65–0.75, and 18% exceed 0.75, indicating moderate to high alignment. Only 1% scored below 0.45. These scores are comparable to PAPERBENCH, suggesting no abnormal inflation. Even famous papers like VAE, GAN, and LayerNorm do not receive notably high scores, implying that semantic similarity, not memorization based on citation relationships, drives the results, especially for tool/benchmark papers, which naturally resemble their references more.

## F. Embedding-based Evaluation Details

In this section, we first explain the motivation for our designed multi-component embedding-based evaluation, then we provide a more formal definition and implementation details related to our evaluation process.

**Decompositionality**. A single idea or a review can manifest through diverse descriptions or implementation strategies. Therefore, directly applying a cosine similarity-based metric is inadequate for capturing conceptual equivalence. To solve this, we design point-wise evaluation metrics to paraphrase the paper and review it into aligned key points with the same LLM-based prompting. This structure enables alignment between papers that differ methodologically but share similar motivations and problem framings. For instance, in Chen et al. (2025) and Jin et al. (2025), despite distinct methods and settings, experts would find strong alignment on the motivations and core concepts in these papers.

**Scalability**. To address the challenge that a single idea can take many concrete forms, we complement decomposition with scalability. LLMs can generate hundreds of semantically distinct research questions from a single prompt, but evaluating

---

[1]https://scholar.google.es/citations?view_op=top_venues&hl=en&vq=eng

these outputs traditionally requires domain experts—a process that is costly, slow, unscalable, and hard to reproduce. For example, Si et al. (2024) spent thousands hiring top-tier researchers solely for annotation and review, which is infeasible for evaluating large-scale, automated research generation. Our approach replaces this bottleneck with semantic similarity over 5Q-decomposed representations. We can select the best among the samples and make the score the final result.

**Extensibility**. While we acknowledge the importance of elements like mathematical formulations or algorithmic workflows, our framework is inherently extensible—the original format can be expanded with more key points by adding domain-specific dimensions such as algorithmic structure or key theoretical results. This is especially valuable in systems and theory papers, enabling more fine-grained and domain-aware similarity analysis. As demonstrated in [Fine-Grained Evaluation with LLM and Human], our approach also supports the integration of non-semantic metrics like logical consistency and factual accuracy, making it extensible from an evaluation metric perspective.

**Reliability**. Our embedding-based / LLM-based similarity metric builds on state-of-the-art models optimized for knowledge-intensive tasks. Voyage AI embeddings, widely adopted in real-world RAG systems, are designed to reduce hallucination and excel in high-precision semantic retrieval, making them ideal for evaluating research content. Additionally, state-of-the-art LLMs are highly effective at semantic comparison.

**Baselines for evaluation**. To check whether RESEARCHTOWN provides a realistic simulation, we benchmark similarity in real-world research activity. For paper writing, we reference two concurrent papers (Chen et al., 2025; Jin et al., 2025) recognized for presenting nearly identical ideas, yet with different writing styles and experiments, which yield a VoyageAI similarity of 0.8244. This suggests that scores above 0.82 can potentially indicate strong idea overlap. For review writing, we analyze the data of reviewers evaluating the same paper. The average inter-reviewer similarity is 0.5900 (strengths) and 0.5904 (weaknesses), reflecting natural variance in human judgment. These inter-similarity scores in the real world confirm that similarity scores represent realistic simulation.

**More details on paper evaluation**. To evaluate the paper writing stage, we define a distance function $d_p(\cdot, \cdot)$ to measure the similarity between the generated paper $\mathbf{h}_v$ and the ground-truth paper $\mathbf{h}_v^*$. Since directly comparing full papers in different formats can be challenging and inaccurate, we align $\mathbf{h}_v$ and $\mathbf{h}_v^*$ into a unified format using a well-recognized framework [1] that summarizes the core components of a paper through five questions: (1) What is the problem? (2) Why is it interesting and important? (3) Why is it hard? (4) Why hasn't it been solved before? (5) What are the key components of my approach and results? We mark these questions as Q1-Q5 for short. By using an LLM-based summarization function $f_{\text{sum}}(\cdot)$, we convert the input papers into an aligned text-based list $\mathbf{a}_v = f_{\text{sum}}(\mathbf{h}_v)$ and $\mathbf{a}_v^* = f_{\text{sum}}(\mathbf{h}_v^*)$, where each element in $\mathbf{a}_v$ and $\mathbf{a}_v^*$ corresponds to the answer of one question mentioned above. The distance function for paper writing is formally defined as:

$$d_p(\mathbf{h}_v, \mathbf{h}_v^*) = \frac{1}{5} \sum_{i=1}^{5} \text{SIM}(\mathbf{a}_{v,i}, \mathbf{a}_{v,i}^*) \tag{21}$$

where $\text{SIM}(\cdot, \cdot)$ represents an embedding-based similarity metric, such as voyage-3 [2] and text-embedding-large-3 [3]. By leveraging the LLM to generate structured embeddings for each question, this approach ensures a meaningful and consistent comparison of the generated and ground-truth papers.

**More details on review evaluation**. Another research activity we aim to evaluate is review writing. Similar to paper writing evaluation, we project both real-world and generated reviews into a unified format for comparison. For this purpose, we adopt a bullet point-based format to represent weaknesses and advantages in the review, as it effectively captures the key aspects of a review. Using an LLM-based summarization function $f_{\text{sum}}(\cdot)$, we convert the input reviews $\mathbf{r}_v$ and $\mathbf{r}_v^*$ into a bullet point list $\mathbf{b}_v = f_{\text{sum}}(\mathbf{r}_v)$ and $\mathbf{b}_v^* = f_{\text{sum}}(\mathbf{r}_v^*)$, where each element of $\mathbf{b}_v$ and $\mathbf{b}_v^*$ corresponds to a bullet point of the review. Formally, the distance function for review writing is computed as:

$$d_r(\mathbf{r}_v, \mathbf{r}_v^*) = \frac{1}{n} \sum_{j=1}^{n} \max_{i} \text{SIM}(\mathbf{b}_{v,i}, \mathbf{b}_{v,j}^*) \tag{22}$$

where $\text{SIM}(\cdot, \cdot)$ refers to the same similarity metric in paper writing evaluation. This metric emphasizes the *recall* rate of the generated review by measuring whether each point in the real-world review is potentially included in the generated

---

[1] https://cs.stanford.edu/people/widom/paper-writing.html
[2] https://blog.voyageai.com/2024/09/18/voyage-3/
[3] https://openai.com/index/new-embedding-models-and-api-updates/

review. Since each review consists of both strengths and weaknesses, we compute separate similarity scores for strengths and weaknesses. Additionally, since both $\mathbf{r}_v$ and $\mathbf{r}_v^*$ include a final score $\mathbf{S}_v$ and $\mathbf{S}_v^*$ as attributes, we calculate $\Delta\mathbf{S}_v = |\mathbf{S}_v - \mathbf{S}_v^*|$ to quantify the difference between the generated and real-world review scores.

**Prompt**. Table 15 presents the prompt used to convert any existing paper into responses to the five critical research questions. Similarly, Table 16 shows the prompt used to transform any existing review into a bullet-point format. Both prompts ensure that the transformed papers and reviews are aligned with the generated ones, facilitating consistent evaluation in the same format. The transformed format for the paper is considered as $\mathbf{a}_v^*$ and the concatenation of all ground-truth reviews is considered as $\mathbf{b}_v^*$, as mentioned in Section §6.

**Metric**. For our embedding-based similarity calculations, we use the *text-embedding-large-3* model via the `litellm` Python package by calling `litellm.embedding()`. For the *voyage-3* model, we rely on the `voyageai` Python package by calling `voyageai.Client().embed()`. We then compute the cosine similarity between the resulting embeddings to measure their similarity.

## G. LLM-based Evaluation Details

In this section, we provide more technical details about using LLM prompting for evaluation.

**Prompting for similarity**. For prompting-based evaluation, we decompose overall similarity into six fine-grained dimensions: (1) topic consistency, (2) method consistency, (3) factual consistency, (4) claim consistency, (5) application context consistency, and (6) overall semantic similarity. These dimensions are designed to capture distinct yet complementary aspects of alignment between the generated and reference proposals, ranging from high-level research focus (such as topic and application context) to specific technical content (such as methods, facts, and claims). Importantly, they are intended to capture nuances that may not be easily detected by embedding-based models, enabling a more comprehensive and interpretable assessment than relying on a single similarity score. Each dimension is rated on a scale from 0 to 10.

**Prompting for novelty and feasibility**. In addition to measuring similarity, we prompt LLMs to assess two intrinsic quality dimensions: (1) novelty and (2) feasibility, which we consider essential characteristics of a strong research proposal. While similarity captures how well the generated content aligns with a reference, it does not fully reflect the proposal's originality or practicality. These intrinsic dimensions address that gap by evaluating the creativity of the proposed idea and its potential for real-world implementation. Each dimension is scored on a scale from 0 to 10, complementing similarity-based metrics for a more holistic evaluation.

**Prompt**. To enable efficient evaluation, we adopt parallel prompting, where both the reference and generated proposals are input to the LLM in a single prompt, along with all evaluation criteria. This allows the model to produce scores for all dimensions simultaneously in one forward pass. The detailed descriptions of these evaluation criteria and the full prompts are provided in Table 17.

## H. Human Evaluation Details

**Annotator Information**. We recruit two graduate-level students with backgrounds in computer science and artificial intelligence. Both annotators have prior experience publishing in top-tier machine-learning conferences.

**Annotated Data**. We randomly sample 40 reference proposals and their corresponding generated proposals from PAPER-BENCH for human evaluation. We ask annotators to annotate on overall similarity, novelty, and feasibility.

**Annotation Process**. The annotation process consists of three stages: (1) preliminary annotation, (2) discussion, and (3) final annotation. In the preliminary stage, each annotator independently labels 10 examples. They then meet to discuss discrepancies, align their understanding, and refine the annotation criteria. Based on this discussion, they proceed to annotate the official 40 examples using the agreed-upon guidelines as the final results.

**Annotation Instructions**. At the start, annotators receive the same input information as used in the LLM-based prompting setup. During the discussion phase, they collaboratively develop more detailed and consistent annotation guidelines to ensure alignment in their final evaluations.

# I. Ablation Study Details

Due to the experimental setting, the ablation study on paper writing simulation tasks does not include all the 1,000 tasks that existed in RESEARCHBENCH. Therefore, we provide detailed explanations and technical details for this.

**Data for paper-writing researcher number ablation.**. Not all papers in RESEARCHBENCH have more than five authors. To ablate the effect of the number of researchers (1 to 5), we select a subset from the hard part of PAPERBENCH within RESEARCHBENCH, including 333 paper writing tasks, ensuring each paper has more than five authors. This filtering results in a subset of 172 paper-writing tasks. We focus on the hard subset because we believe that involving multiple research agents in more challenging scenarios yields a more significant difference in performance.

**Data for paper-writing paper number ablation.**. In this ablation, we vary the number of cited papers included in different sections of the target paper. Specifically, we examine citations in the related work, introduction, and other sections. To do this, we retrieve the raw LaTeX source from arXiv and extract references at the section level. Due to varying data availability, we finalize a subset of RESEARCHBENCH that includes 296 paper-writing tasks for this study.

**Data for review-writing researcher number ablation.**. Since the reviewer construction does not depend on any complex data preprocessing, we do not encounter data issues for the review-writing ablation. Consequently, the ablation results are based on all 200 review-writing tasks in RESEARCHBENCH.

# J. Additional Experimental Results

We provide more comprehensive experimental results on each sub-part of RESEARCHBENCH (PAPERBENCH, REVIEW-BENCH, HIGHIMPACTPAPERBENCH) in this section.

**Additional Results on PAPERBENCH**. Table 5 shows that all models—Qwen-2.5-7B-Instruct, GPT-4o-mini, and Deepseek-v3—consistently achieve better performance with richer reference contexts (AGG-data and AGG-global) compared to narrower ones (AGG-self and AGG-agent), highlighting the importance of contextual information in similarity evaluation.

**Additional Results on REVIEWBENCH**. As shown in Table 6, voyage-3 embeddings yield higher strength scores and larger $\Delta\mathbf{S}$ values than text-embedding-3, indicating greater discriminative power. While Qwen-2.5-7B-Instruct maintains strong similarity scores across all aggregation types, it exhibits larger deviations from human scores, suggesting potential scoring bias or overconfidence in its own outputs.

**Additional results on HIGHIMPACTPAPERBENCH**. Besides the full results on PAPERBENCH, we also evaluate RE-SEARCHTOWN under extreme conditions by attempting to simulate 100 of the most-cited machine learning papers from the past decade. RESEARCHTOWN achieves low similarity scores for papers introducing groundbreaking methods, such as "*Layer Normalization*" (Ba, 2016), or novel topics, such as "*Energy and Policy Considerations for Deep Learning in NLP*" (Strubell et al., 2019). However, the framework performs notably better on impactful papers focused on analysis or tool development. For instance, it achieves a similarity score exceeding 0.8 for papers like "*Is BERT Really Robust? A Strong Baseline for Natural Language Attack on Text Classification and Entailment*" (Jin et al., 2020), which provides adversarial analysis, and "*Stanza: A Python Natural Language Processing Toolkit for Many Human Languages*" (Qi et al., 2020), which offers a practical toolkit. These results suggest that high-impact research ideas may be more feasible than commonly perceived, and RESEARCHTOWN could potentially serve as a tool to inspire future impactful research.

# K. Additional Case Study

Beyond the examples included in Case Study Section §11, we provide additional examples to show the generation results of our work and provide further insights about the strengths and weaknesses of RESEARCHTOWN.

**Additional case study for in-distribution evaluation**. Tables 18, 19, 20, 21, and 22 present examples of tasks and their corresponding outputs for the in-distribution evaluation of RESEARCHTOWN. These examples illustrate the evaluation process defined in this work.

**Additional case study for out-of-distribution application**. In Table 23, 24, 25, 26, 27, 28, 29, 30, 31, and 32, we show examples of the inputs and outputs of the out-of-distribution application of RESEARCHTOWN. Additionally, each table caption includes a brief comment on the quality of the generated papers for reference.

Table 5: **Evaluation results on embedding-based similarity score of PAPERBENCH.** We include comprehensive results on three different models and include sub-scores from Q1-Q5.

| Model | Component | AGG-self | | | AGG-agent | | | AGG-data | | | AGG-global | | |
|---|---|---|---|---|---|---|---|---|---|---|---|---|---|
| | | easy | mid | hard | easy | mid | hard | easy | mid | hard | easy | mid | hard |
| DeepSeek-v3 | Q1 | 43.84 | 44.08 | 41.74 | 52.46 | 49.93 | 47.33 | 68.18 | 61.70 | 54.74 | 66.10 | 61.20 | 55.66 |
| | Q2 | 51.52 | 51.71 | 50.02 | 61.74 | 59.75 | 58.29 | 75.53 | 69.68 | 63.92 | 74.29 | 69.81 | 65.76 |
| | Q3 | 50.84 | 51.44 | 49.58 | 59.74 | 58.57 | 56.32 | 71.02 | 67.05 | 61.03 | 70.32 | 67.18 | 62.33 |
| | Q4 | 48.51 | 48.87 | 47.80 | 56.21 | 54.09 | 53.22 | 67.55 | 63.11 | 59.86 | 66.31 | 62.72 | 60.01 |
| | Q5 | 50.29 | 50.02 | 49.03 | 60.37 | 58.44 | 56.43 | 68.04 | 64.38 | 59.96 | 69.51 | 65.82 | 62.90 |
| | Overall | 49.00 | 49.22 | 47.64 | 58.10 | 56.16 | 54.32 | 70.06 | 65.18 | 59.90 | 69.31 | 65.35 | 61.33 |
| gpt-4o-mini | Q1 | 36.93 | 35.88 | 35.80 | 53.51 | 51.70 | 48.62 | 70.98 | 61.79 | 49.26 | 72.78 | 65.22 | 56.62 |
| | Q2 | 52.82 | 52.44 | 52.22 | 61.74 | 60.30 | 58.24 | 81.14 | 73.34 | 62.54 | 79.16 | 73.55 | 66.20 |
| | Q3 | 50.10 | 50.16 | 49.85 | 59.09 | 58.65 | 55.69 | 76.58 | 69.12 | 58.13 | 74.40 | 69.20 | 61.47 |
| | Q4 | 45.71 | 44.82 | 44.87 | 54.97 | 53.23 | 51.73 | 71.37 | 63.03 | 54.33 | 71.13 | 64.92 | 59.22 |
| | Q5 | 46.54 | 46.32 | 46.76 | 55.18 | 53.85 | 52.03 | 71.71 | 64.84 | 55.84 | 71.46 | 66.35 | 60.93 |
| | Overall | 46.42 | 45.92 | 45.90 | 56.90 | 55.55 | 53.26 | 74.36 | 66.42 | 56.02 | 73.79 | 67.85 | 60.89 |
| Qwen-2.5-7B-Instruct | Q1 | 40.41 | 39.90 | 40.05 | 49.58 | 48.91 | 45.11 | 70.34 | 63.38 | 57.73 | 71.51 | 64.55 | 58.52 |
| | Q2 | 52.67 | 52.43 | 53.00 | 61.74 | 61.40 | 58.10 | 75.17 | 71.08 | 66.74 | 75.16 | 71.22 | 66.81 |
| | Q3 | 50.34 | 50.39 | 50.66 | 58.55 | 57.82 | 54.56 | 73.60 | 67.96 | 63.50 | 73.40 | 68.09 | 63.62 |
| | Q4 | 42.58 | 42.24 | 42.65 | 50.97 | 49.99 | 47.35 | 63.66 | 58.18 | 54.98 | 62.56 | 57.62 | 53.80 |
| | Q5 | 46.46 | 46.11 | 46.86 | 56.66 | 55.26 | 52.64 | 66.78 | 63.35 | 58.97 | 68.13 | 64.54 | 60.07 |
| | Overall | 46.49 | 46.21 | 46.65 | 55.50 | 54.68 | 51.55 | 69.91 | 64.79 | 60.38 | 70.15 | 65.20 | 60.56 |

Table 6: **Evaluation results on embedding-based similarity score of REVIEWBENCH.** We include comprehensive results on three different models and include strengths, weaknesses, and $\Delta \mathbf{S}$.

| Setting | Model | text-embedding-3 | | | voyage-3 | | |
|---|---|---|---|---|---|---|---|
| | | Weakness | Strength | $\Delta S$ | Weakness | Strength | $\Delta S$ |
| AGG-self | Qwen-2.5-7B-Instruct | 49.79 | 52.08 | 1.36 | 65.24 | 64.82 | 1.36 |
| | Deepseek-v3 | 48.92 | 51.98 | 1.11 | 62.41 | 65.18 | 1.11 |
| | gpt-4o-mini | 47.16 | 51.23 | 1.27 | 61.24 | 65.18 | 1.27 |
| AGG-agent | Qwen-2.5-7B-Instruct | 50.12 | 51.38 | 1.41 | 66.24 | 65.49 | 1.41 |
| | Deepseek-v3 | 48.56 | 51.68 | 1.05 | 62.80 | 65.38 | 1.05 |
| | gpt-4o-mini | 46.75 | 51.66 | 1.19 | 61.29 | 66.03 | 1.19 |
| AGG-data | Qwen-2.5-7B-Instruct | 50.26 | 51.66 | 1.28 | 66.09 | 65.05 | 1.28 |
| | Deepseek-v3 | 49.21 | 51.34 | 1.07 | 63.11 | 65.19 | 1.07 |
| | gpt-4o-mini | 47.62 | 51.45 | 1.26 | 61.74 | 65.57 | 1.26 |
| AGG-global (k=5) | Qwen-2.5-7B-Instruct | 50.21 | 50.78 | 0.79 | 65.58 | 63.72 | 0.79 |
| | Deepseek-v3 | 48.95 | 50.57 | 0.81 | 62.56 | 64.21 | 0.81 |
| | gpt-4o-mini | 51.51 | 47.17 | 1.55 | 66.01 | 61.39 | 1.55 |

| Role | Content |
|------|---------|
| **System** | You are an autonomous intelligent agent tasked with writing the first-person persona of a researcher based on his publications.
You will be provided with the following information:
Publications - A list of paper abstracts written by the researcher that you will be writing about.
You should provide the following information:
Persona - A comprehensive first-person persona.
You should focus more on recent publications, which reflect the researcher's recent persona.
You should be concise and clear. The persona should range from 100 to 300 words. |
| **User** | Here is the publication history of one researcher:
Publication1:
Learning node embeddings that capture a node's position within the broader graph structure is crucial for many prediction tasks on graphs. However, existing Graph Neural Network (GNN) architectures have limited power in capturing the position/location of a given node concerning all other nodes of the graph. Here we propose Position-aware Graph Neural Networks (P-GNNs), a new class of GNNs for computing position-aware node embeddings. P-GNN first samples sets of anchor nodes, computes the distance of a given target node to each anchor set, and then learns a non-linear distance-weighted aggregation scheme over the anchor sets. This way, P-GNNs can capture the positions/locations of nodes concerning the anchor nodes. P-GNNs have several advantages: they are inductive, scalable, and can incorporate node feature information. We apply P-GNNs to multiple prediction tasks, including link prediction and community detection. We show that P-GNNs consistently outperform state-of-the-art GNNs, with up to 66% improvement in terms of the ROC AUC score.
...
Publication8:
AutoML has demonstrated remarkable success in finding an effective neural architecture for a given machine learning task defined by a specific dataset and an evaluation metric. However, most present AutoML techniques consider each task independently from scratch, which requires exploring many architectures, leading to high computational costs. Here we propose AutoTransfer, an AutoML solution that improves search efficiency by transferring the prior architectural design knowledge to the novel task of interest. Our key innovation includes a task-model bank that captures the model performance over a diverse set of GNN architectures and tasks, and a computationally efficient task embedding that can accurately measure the similarity among different tasks. Based on the task-model bank and the task embeddings, we estimate the design priors of desirable models of the novel task by aggregating a similarity-weighted sum of the top-K design distributions on tasks that are similar to the task of interest. The computed design priors can be used with any AutoML search algorithm. We evaluate AutoTransfer on six datasets in the graph machine learning domain. Experiments demonstrate that (i) our proposed task embedding can be computed efficiently and that tasks with similar embeddings have similar best-performing architectures; (ii) AutoTransfer significantly improves search efficiency with the transferred design priors, reducing the number of explored architectures by an order of magnitude. Finally, we release GNN-Bank-101, a large-scale dataset of detailed GNN training information of 120,000 task-model combinations to facilitate and inspire future research.
Please begin writing the first-person persona that covers the previous research experience. |
| **Assistant** | I am a researcher focused on advancing graph neural networks (GNNs) and machine learning. My key contributions include Position-aware GNNs (P-GNNs) for capturing node positions, Identity-aware GNNs (ID-GNNs) with greater expressive power, and the ROLAND framework for dynamic graph learning. I also study the relationship between neural network structures and their performance using relational graphs and have explored a vast GNN design space to enhance predictive capabilities across diverse tasks. |
| **User** | Here is the publication history of one researcher:
Publication 1: {abstract for paper 1}
...
Publication $n$: {abstract for paper $n$}
Please begin writing the first-person persona. |

Table 7: Paper reading message prompt template for $f_u(\cdot)$.

| Role | Content |
|------|---------|
| **User** | Who are you? |
| **Assistant** | {profile} |
| **User** | Here is the content collected from related papers:
Paper 1: {abstract for cited paper 1}
Paper 2: {abstract for cited paper 2}
...
Paper $n$: {abstract for cited paper $n$}
You need to write a research proposal for a paper in the field of Machine Learning based on these related papers.
The research proposal should rely more on the cited papers rather than your own research experience.
Your research experience should be utilized to select the most useful and valuable papers from the related papers for proposal writing.
Here is a high-level, summarized insight into the Machine Learning research field.
Here are the five core questions:
[Question 1] - What is the problem?
Formulate the specific research question you aim to address.
Only output one question and do not include any more information.
[Question 2] - Why is it interesting and important?
Explain the broader implications of solving this problem for the research community.
Discuss how the paper will affect future research.
Discuss how addressing this question could advance knowledge or lead to practical applications.
[Question 3] - Why is it hard?
Discuss the challenges and complexities involved in solving this problem.
Explain why naive or straightforward approaches may fail.
Identify any technical, theoretical, or practical obstacles that need to be overcome.
MAKE IT CLEAR.
[Question 4] - Why hasn't it been solved before?
Identify gaps or limitations in previous research or existing solutions.
Discuss any barriers that have prevented this problem from being solved until now.
Explain how your approach differs from or improves upon prior work.
MAKE IT CLEAR.
[Question 5] - What are the key components of my approach and results?
Outline your proposed methodology in detail, including the method, dataset, and metric that you plan to use. But you must include these in one paragraph and not use subtitles.
Describe the expected outcomes.
MAKE IT CLEAR.
Please brainstorm the following proposal with the given format. |

Table 8: Paper writing message prompt template for $f_a(\cdot)$.

| Role | Content |
|------|---------|
| **User** | Who are you? |
| **Assistant** | {profile} |
| **User** | Here is a high-level, summarized insight of a research field: Machine Learning.
Here are the five core questions to consider:
[Question 1] - What is the problem?
[Question 2] - Why is it interesting and important?
[Question 3] - Why is it hard?
[Question 4] - Why hasn't it been solved before?
[Question 5] - What are the key components of my approach and results?
Multiple papers have been generated for the above questions:
Paper 1: {agent written paper 1}
Paper 2: {agent written paper 2}
...
Paper $n$: {agent written paper $n$}
Your task is to summarize and select the key insights that are suitable from these proposals.
1. Identify shared themes and common points among the proposals.
2. Highlight and select any valuable perspectives or contrasting elements and combine them into one proposal.
3. Provide a concise proposal for each question based on the proposal candidates.
Output the result in the provided five-question format.
Ensure the generated paper is clear, concise, and avoids repeating full proposals verbatim. |

Table 9: Paper writing aggregation prompt template for $f_g(\cdot)$.

| Role | Content |
|------|---------|
| **System** | You are an autonomous intelligent agent tasked to review a submission to an academic conference.
You should write the strength of this paper.
You will be provided with the following information:
Submission - Full content of the submitted paper.
You should provide the following information:
Strength - Advantages and strengths of the submission that can improve its chances of being accepted. |
| **User** | Here is your profile:
{profile}
Here is the submission:
{full content for paper}
Here are the abstracts of the cited papers:
Paper 1: {abstract for cited paper 1}
Paper 2: {abstract for cited paper 2}
...
Paper $n$: {abstract for cited paper $n$}
Please evaluate the submission based on the following criteria:
Clarity: Is the writing clear, structured, and do terms defined?
Baselines: Are baseline comparisons relevant, sufficient, and not excessive?
Novelty: Is the approach innovative or distinct from prior work?
Results: Are improvements significant, well-supported, and statistically robust?
Limitations: Are weaknesses acknowledged and future work discussed?
Related Work: Are key references cited and connections made?
Technical: Are methods detailed enough for replication?
Please combine both the ideas and the experiments in the submission when evaluating it.
When commenting on the experiments, refer to the exact numbers from the experiments.
Please begin writing the strength of the submission.
It should be 200 words long.
Please write in bullet points.
Do not limit yourself to the aforementioned criteria, like clarity, baselines, novelty, results, limitations, related work, and technical.
You should also use your previous experience in your profile when analyzing the submission. |

Table 10: Review writing (strength) message prompt template for $f_u(\cdot)$.

| Role | Content |
|------|---------|
| **System** | You are an autonomous intelligent agent tasked to review a submission to an academic conference.
You should write the weaknesses of this paper.
You will be provided with the following information:
Submission - Full content of the submitted paper.
You should provide the following information:
Weakness - Disadvantages and drawbacks of the submission that must be improved before it can be accepted.
You should notice that the abstract might not cover every detail, so you shouldn't be overly strict. |
| **User** | Here is your profile:
{profile}
Here is the submission:
{full content for paper}
Here are the abstracts of the cited papers:
Paper 1: {abstract for cited paper 1}
Paper 2: {abstract for cited paper 2}
...
Paper $n$: {abstract for cited paper $n$}
Please evaluate the submission based on the following criteria:
Clarity: Is the writing clear, structured, and terms defined?
Baselines: Are baseline comparisons relevant, sufficient, and not excessive?
Novelty: Is the approach innovative or distinct from prior work?
Results: Are improvements significant, well-supported, and statistically robust?
Limitations: Are weaknesses acknowledged and future work discussed?
Related Work: Are key references cited and connections made?
Technical: Are methods detailed enough for replication?
Please combine both the ideas and the experiments in the submission when evaluating it.
When commenting on the experiments, refer to the exact numbers from the experiments.
Please begin writing the strength of the submission.
It should be 200 words long.
Please write in bullet points.
Do not limit yourself to the aforementioned criteria, like clarity, baselines, novelty, results, limitations, related work, and technical.
You should also use your previous experience in your profile when analyzing the submission. |

Table 11: Review writing (weakness) message prompt template for $f_u(\cdot)$.

| Role | Content |
|------|---------|
| **System** | You are an autonomous intelligent agent tasked to score the following submission. You should act as a professional and fair member of that conference to score. The score should be between 1 and 10, where 1 is the lowest and 10 is the highest. You will be provided with the following information: Paper - Full content of a submission to an academic conference. Strengths - Strengths of the submission. Weakness - Weakness of the submission. You should provide the following information: Score - A score between 1 to 10 to evaluate the overall quality of the submission to an academic journal. It should be one of 1, 2, ..., 10. 1 is the lowest score, while 10 is the highest score. You should just provide one number as the score and nothing else. Please evaluate the submission based on the summarized strengths and weaknesses provided. The score should be more related to weakness. If there is a critical weakness in the submission, you should give a lower score. If the submission has a minor weakness, you can give a higher score. If the submission has no weaknesses, you should give a high score. But the strengths should also be considered in the evaluation. |
| **User** | Here is your profile: {profile} Here is the strength of the paper: {strength} Here is the weakness of the paper: {weakness} Please refer to the rubrics below to evaluate the submission: 10/10: The submission is in the top 2% of all papers. It changed my thinking on its topic, being one of the most thorough, convincing, and well-written papers I have ever read. I will fight for this paper to be accepted. 8/10: The submission is among the top 10% of all the papers. It provides sufficient justification for all its arguments and claims. Some extra experimentation is needed, but they are not essential. The proposed method is very original and can be generalized to various fields. This submission deepens the understanding of some phenomena or lowers the bar for future research on an existing problem. 6/10: The submission gives sufficient support for its major arguments or claims. However, some minor points are not well justified and need extra support or details. The proposed method is moderately original, and it is generalizable to various fields. The submission itself is not particularly innovative, so it would not be a significant loss if it were not accepted. 5/10: Some of the major arguments or claims are not sufficiently justified. There exist major weaknesses in technical or methodological aspects. The proposed method is somewhat original, and it is generalizable to various fields. I am more on the side of rejection, but I can be convinced otherwise. 3/10: The submission makes only marginal contributions to the field. 1/10: The submission is not sufficiently thorough for publication or is not relevant to the conference. Your score is: [score] |

Table 12: Review writing (score) message prompt template for $f_u(\cdot)$.

| Role | Content |
|------|---------|
| **System** | You are an autonomous intelligent agent tasked to write the strength of the submission for the following submission you have made to an academic conference. Your summary of strength should summarize the reviews to help the reviewers make a decision.
You will be provided with the following information:
Submission - Full content of the paper submitted to this conference.
Reviews - It typically contains the score, strength, and weakness of the submission, each by a different reviewer.
You should provide the following information:
Strength - The strength of the submission based on the reviews. |
| **User** | Here is the paper:
{full content of paper}
Here are the reviews:
Review 1: {review 1}
Review 2: {review 2}
...
Review $n$: {review $n$}
Please summarize the important points from the 'strength' section of the reviews.
Please write in bullet points. It should be 200 words long. |

Table 13: Review writing (strength) aggregation prompt template for $f_g(\cdot)$.

| Role | Content |
|------|---------|
| **System** | You are an autonomous intelligent agent tasked to write the weakness of the submission for the following submission you have made to an academic conference. Your summary of weakness should summarize the reviews to help the reviewers make a decision.
You will be provided with the following information:
Submission - Full content of the paper submitted to this conference.
Reviews - It typically contains the score, weakness, and strength of the submission, each by a different reviewer.
You should provide the following information:
Weakness - The weakness of the submission based on the reviews. |
| **User** | Here is the paper:
{full content of paper}
Here are the reviews:
Review 1: {review 1}
Review 2: {review 2}
...
Review $n$: {review $n$}
Please summarize the important points from the 'weakness' section of the reviews.
Please write in bullet points. It should be 200 words long. |

Table 14: Review writing (weakness) aggregation prompt template for $f_g(\cdot)$.

| Role | Content |
|------|---------|
| **User** | Here is a high-level, summarized insight into a research field of machine learning.
Here are the five core questions:
[Question 1] - What is the problem?
Formulate the specific research question you aim to address.
Only output one question and do not include any more information.
[Question 2] - Why is it interesting and important?
Explain the broader implications of solving this problem for the research community.
Discuss how such a paper will affect future research.
Discuss how addressing this question could advance knowledge or lead to practical applications.
[Question 3] - Why is it hard?
Discuss the challenges and complexities involved in solving this problem.
Explain why naive or straightforward approaches may fail.
Identify any technical, theoretical, or practical obstacles that need to be overcome.
MAKE IT CLEAR.
[Question 4] - Why hasn't it been solved before?
Identify gaps or limitations in previous research or existing solutions.
Discuss any barriers that have prevented this problem from being solved until now.
Explain how your approach differs from or improves upon prior work.
MAKE IT CLEAR.
[Question 5] - What are the key components of my approach and results?
Outline your proposed methodology in detail, including the method, dataset, and metric that you plan to use. Describe the expected outcomes.
MAKE IT CLEAR.
The introduction of paper:
{introduction section of paper}
Please provide the five core questions based on the above content. |

Table 15: Format transformative prompt for real-world papers.

| Role | Content |
|------|---------|
| **System** | You are a helpful agent. |
| **User** | {strength/weakness}
Please rewrite the following strength in bullet points.
Do not include anything else.
Start from '-' for each bullet point. |

Table 16: Format transformative prompt for real-world reviews.

| Role | Content |
|------|---------|
| **User** | You are an expert evaluator of research proposals. Given a reference and a generated proposal, rate their similarity from 1 (completely different) to 10 (nearly identical) for each of the following dimensions:

- Topic Consistency: Do both proposals address the same research topic or problem area?
- Method Consistency: Are the research methods and approaches used in both proposals similar?
- Factual Consistency: Are the datasets, metrics, and models mentioned consistent between the two proposals?
- Claim Consistency: Do both proposals present similar conclusions or findings?
- Application Context Consistency: Are the application domains or use-cases targeted by both proposals the same?
- Overall Semantic Similarity: Do the two proposals convey the same overall idea or message?

Additionally, assess the following for each proposal individually:
- Novelty of Reference Proposal: Does the reference proposal introduce new ideas or perspectives?
- Feasibility of Reference Proposal: How realistic and implementable is the reference proposal based on its described methods and objectives?
- Novelty of Generated Proposal: Does the generated proposal introduce new ideas or perspectives?
- Feasibility of Generated Proposal: How realistic and implementable is the generated proposal based on its described methods and objectives?

Respond strictly in JSON format like:
{
"Topic Consistency": X,
"Method Consistency": X,
"Factual Consistency": X,
"Claim Consistency": X,
"Application Context Consistency": X,
"Overall Semantic Similarity": X,
"Novelty of Reference Proposal": X,
"Feasibility of Reference Proposal": X,
"Novelty of Generated Proposal": X,
"Feasibility of Generated Proposal": X
} |

Table 17: Prompt for LLM-based evaluation of fine-grained similarity and quality assessment of research proposals.

| | PAPERBENCH-easy **Task** | |
|---|---|---|
| | The Marginal Value of Momentum for Small Learning Rate SGD | |
| | PAPERBENCH-easy **Output** | |
| **Question** | **Reference Paper** | **Generated Paper** |
| What is the problem? | How can we theoretically establish the benefits of momentum in stochastic gradient descent methods for deep learning, particularly in the presence of stochastic gradient noise? | How can we effectively leverage momentum-based optimization methods in stochastic gradient descent (SGD) to enhance the generalization performance of deep learning models, particularly when trained on large-scale datasets with noisy labels and large batch sizes? |
| Why is it interesting and important? | Understanding the role of momentum in stochastic optimization is crucial for the research community as it can lead to more stable and efficient training of deep learning models. If we can theoretically prove the advantages of momentum in the presence of noise, it could reshape optimization strategies, leading to faster convergence rates and improved performance in various applications. This advancement could also inspire new methodologies that leverage momentum more effectively, potentially influencing future research directions in optimization techniques. | This research is significant as it addresses the growing trend of using large batch sizes in deep learning, which can lead to poorer generalization performance. By improving our understanding of momentum-based methods, we can enhance model robustness and efficiency in real-world applications across various domains, including healthcare, finance, and autonomous systems. Insights gained could inform future optimization strategies, balancing training efficiency with generalization capabilities. |
| Why is it hard? | The challenge lies in the inherent stochasticity introduced by mini-batch sampling, which can obscure the true gradient and complicate the analysis of momentum's effects. Naive approaches may fail because they do not account for the noise's impact on convergence rates, leading to misleading conclusions. Additionally, the theoretical frameworks established for noiseless scenarios do not directly translate to the stochastic case, creating a significant gap in understanding. Overcoming these complexities requires rigorous mathematical analysis and potentially new theoretical tools. | The complexity arises from the intricate dynamics between momentum parameters, learning rates, and the stochastic nature of gradient updates, especially in the presence of noisy labels and large batch sizes. Naive implementations may lead to suboptimal convergence and generalization due to the interaction of these factors. Additionally, the theoretical understanding of momentum's effects in non-convex optimization landscapes is still limited, complicating the design of effective algorithms. |
| Why hasn't it been solved before? | Previous research has primarily focused on the noiseless case or has not rigorously analyzed the stochastic setting, leading to gaps in understanding momentum's role in noisy environments. Existing studies often conclude that momentum does not provide a significant speedup compared to vanilla SGD, but they lack a comprehensive theoretical framework that addresses the stochastic nature of deep learning. Our approach aims to fill this gap by providing a more nuanced analysis that considers the effects of stochastic gradient noise on momentum's performance. | Previous research has often focused on either the empirical performance of momentum methods or their theoretical foundations in isolation, neglecting the combined effects of noise and batch size on optimization dynamics. Many studies have not adequately explored how momentum can be optimally tuned in noisy environments or how it interacts with varying batch sizes. This gap has hindered a comprehensive understanding of momentum's role in SGD. |
| What are the key components of my approach and result? | Our proposed methodology involves a rigorous theoretical analysis of momentum in stochastic gradient descent, utilizing a combination of mathematical modeling and empirical validation. We will analyze various datasets to evaluate the performance of standard SGDM against modified versions that account for stochastic noise. The key metrics for evaluation will include convergence rates and stability of training loss. We expect to demonstrate that momentum can indeed stabilize the optimization process and lead to faster convergence in the presence of noise, thereby providing a solid theoretical foundation for its use in deep learning. | I propose to develop a novel momentum-based optimization algorithm that dynamically adjusts momentum parameters based on the noise level in the training data and the batch size. This will involve conducting experiments on benchmark datasets such as CIFAR-10 and ImageNet, evaluating performance against standard SGD and existing momentum methods using metrics like accuracy and generalization error. The expected outcome is a robust optimization algorithm that demonstrates improved generalization performance and provides practical guidelines for effectively applying momentum in SGD, particularly in challenging training scenarios characterized by noise and large-scale data. |

Table 18: Case study on paper writing results of PAPERBENCH-easy.

| PAPERBENCH-medium **Task** | | |
|---|---|---|
| L4GM: Large 4D Gaussian Reconstruction Model | | |
| PAPERBENCH-medium **Output** | | |
| **Question** | **Reference Paper** | **Generated Paper** |
| What is the problem? | How can we efficiently generate high-quality animated 3D assets from monocular videos or text inputs? | How can we effectively generate high-fidelity 4D dynamic scenes from monocular video inputs while ensuring spatial-temporal consistency and realistic motion representation? |
| Why is it interesting and important? | Solving this problem has significant implications for the research community as it addresses the growing demand for automated tools in 3D content creation, which is currently a labor-intensive process. By enabling the generation of animated 3D assets from easily accessible data sources, this research could democratize access to 3D modeling, fostering innovation in fields such as gaming, virtual reality, and film. Furthermore, it could lead to advancements in related areas like computer vision and generative modeling, paving the way for future research that explores more complex 4D content editing and real-time applications. | This problem is critical for advancing computer vision and graphics, particularly in applications such as virtual reality, gaming, and film production. By enabling the generation of dynamic 3D scenes from single-view inputs, we can democratize access to high-quality content creation tools, allowing artists and developers to produce immersive experiences without extensive resources. This research could lead to breakthroughs in automated content generation, enhancing user experiences and paving the way for innovations in interactive media and AI-driven storytelling. |
| Why is it hard? | The challenges in solving this problem stem from the need for high-quality 4D reconstruction from limited input data, such as monocular videos. Naive approaches may fail due to the inherent complexity of accurately capturing temporal dynamics and spatial details from a single viewpoint. Additionally, existing methods often rely on extensive multiview data, which is costly and time-consuming to collect. The fragility of score distillation techniques and the computational intensity of current models further complicate the task, necessitating innovative solutions to achieve both speed and quality in 4D reconstruction. | Generating 4D dynamic scenes from monocular videos is challenging due to the inherent ambiguity of single-view data, which limits the ability to accurately infer depth and motion dynamics. The lack of comprehensive datasets and the complexities of ensuring temporal coherence and spatial consistency add further difficulty. Existing methods often struggle with maintaining high visual fidelity while capturing the intricate relationships between appearance and motion, leading to artifacts and inconsistencies in the generated output. |
| Why hasn't it been solved before? | Previous research has been limited by the reliance on multi-view data, which restricts applicability due to the high costs associated with data collection. Additionally, existing methods, such as video score distillation, are often slow and sensitive to input variations, leading to inconsistent results. The lack of a large-scale dataset specifically designed for training models on 4D reconstruction has also been a barrier. Our approach differs by leveraging a new dataset of 12 million multiview videos and introducing a feed-forward model that incorporates temporal self-attention, allowing for faster and more reliable 4D reconstruction. | Previous research has primarily focused on static scene reconstruction or required multi-view inputs, which are not always available in practical scenarios. Techniques like Neural Radiance Fields (NeRF) have shown promise but often rely on extensive optimization and multi-view data, limiting their applicability. Additionally, many existing methods do not effectively disentangle motion from appearance, leading to challenges in generating realistic animations. The lack of a unified framework that integrates both 3D and 2D diffusion models has hindered progress in this area. |
| What are the key components of my approach and result? | Our proposed methodology, L4GM, utilizes a large-scale dataset of 12 million multiview videos to train a 4D Large Reconstruction Model that reconstructs sequences of 3D Gaussians from monocular video inputs. The model employs temporal self-attention layers to ensure consistency across frames and includes an interpolation model to enhance output frame rates. | We propose a novel two-stage framework that first utilizes a 3D-aware image diffusion model to generate a high-quality static 3D asset from monocular video input. The second stage will involve optimizing a deformable neural radiance field to learn motion dynamics, ensuring temporal coherence and spatial consistency. Our methodology will be evaluated using diverse datasets of monocular videos, employing metrics such as visual fidelity, temporal consistency, and user preference assessments. The expected outcomes include the generation of realistic 4D scenes that maintain high-quality visual appearance and coherence across frames, significantly advancing the state-of-the-art in dynamic scene generation. |

Table 19: Case study on paper writing results of PAPERBENCH-medium.

| | PAPERBENCH-hard **Task** | |
|---|---|---|
| | Provably Robust Conformal Prediction with Improved Efficiency | |
| | PAPERBENCH-hard **Output** | |
| **Question** | **Reference Paper** | **Generated Paper** |
| What is the problem? | How can we enhance the robustness of conformal prediction methods against adversarial examples while maintaining computational efficiency? | How can we develop a robust and efficient framework for conformal prediction that maintains valid coverage guarantees in the presence of adversarial perturbations and label noise in machine learning classification tasks? |
| Why is it interesting and important? | Solving this problem is crucial for the research community as it addresses the growing concern of adversarial attacks on machine learning models, particularly in safety-critical applications. By improving the robustness of conformal prediction, we can ensure more reliable uncertainty quantification, which is essential for decision-making processes in fields such as healthcare, finance, and autonomous systems. This research could pave the way for future studies that explore more resilient predictive models and lead to practical applications where trustworthiness and safety are paramount. | This problem is critical for enhancing the reliability of machine learning models, especially in high-stakes applications such as medical diagnosis and autonomous systems, where incorrect predictions can have severe consequences. By improving conformal prediction methods to effectively handle adversarial conditions and label noise, we can provide more trustworthy uncertainty quantification. This advancement is essential for the practical deployment of AI systems, fostering greater confidence in their predictions and enabling their use in diverse domains like finance, healthcare, and security. |
| Why is it hard? | The challenges in solving this problem stem from the inherent complexity of adversarial attacks, which can manipulate model predictions in subtle ways. Naive approaches may fail because they do not account for the diverse nature of adversarial perturbations, leading to inadequate coverage guarantees. Additionally, the computational overhead associated with randomized smoothing techniques complicates the implementation of robust conformal prediction, as it requires extensive sampling and can significantly increase training time. Overcoming these technical and practical obstacles is essential to developing an effective solution. | The challenge arises from the complexities associated with label noise and adversarial perturbations, which can distort data distributions and violate the assumptions of traditional conformal prediction methods. Existing approaches often fail to account for the adversarial nature of noise or the distribution shifts that occur during inference. Additionally, ensuring valid coverage guarantees while maintaining model performance requires sophisticated techniques that balance robustness and accuracy, complicating the design of effective algorithms. |
| Why hasn't it been solved before? | Previous research has primarily focused on either conformal prediction or adversarial robustness, often treating them as separate domains. Limitations in existing solutions include a lack of comprehensive methods that integrate robust conformal prediction with adversarial noise handling. Barriers such as insufficient understanding of the interaction between conformal prediction and adversarial examples have hindered progress. Our approach differs by providing a robust conformal training method that does not introduce additional computational costs at test time, thus addressing both robustness and efficiency. | Previous research has largely focused on either conformal prediction under ideal conditions or on adversarial robustness without integrating these two aspects. Many existing methods lack a unified framework that effectively combines conformal prediction with robust techniques against label noise and adversarial attacks. The absence of formal guarantees for coverage in the presence of such perturbations has hindered practical applicability, leaving a gap that our approach aims to fill. |
| What are the key components of my approach and result? | Our proposed methodology involves developing a robust conformal prediction (RSCP) framework that utilizes randomized smoothing to enhance adversarial robustness. We will employ datasets such as CIFAR10 for evaluation and measure performance using metrics like coverage probability and computational efficiency. The expected outcomes include demonstrating that our RSCP method maintains robust coverage guarantees against adversarial perturbations while minimizing computational overhead during both training and testing phases, thus enabling the use of larger base models without increased costs. | We propose a novel framework that integrates probabilistically robust conformal prediction with adversarial training techniques. Our methodology will involve developing an adaptive conformal prediction algorithm that utilizes a quantile-of-quantile approach to establish thresholds for both clean and perturbed data. We will evaluate our approach on benchmark datasets such as CIFAR-10, CIFAR-100, and ImageNet, using metrics like coverage probability and prediction set size to assess performance. The expected outcomes include improved coverage guarantees under adversarial conditions, reduced prediction set sizes, and enhanced computational efficiency, ultimately leading to a more reliable framework for uncertainty quantification in machine learning. |

Table 20: Case study on paper writing results of PAPERBENCH-hard.

| | HIGHIMPACTPAPERBENCH **Task** | |
|---|---|---|
| | GLIDE: Towards Photorealistic Image Generation and Editing with Text-Guided Diffusion Models | |
| | HIGHIMPACTPAPERBENCH **Output** | |
| **Question** | **Reference Paper** | **Generated Paper** |
| What is the problem? | How can we develop a model that generates photorealistic images from natural language prompts while maintaining fidelity to the specified content? | How can we effectively generate high-resolution, semantically coherent, and diverse images from textual descriptions while ensuring fairness and reducing computational costs in text-to-image synthesis models? |
| Why is it interesting and important? | Solving this problem has significant implications for the research community as it bridges the gap between natural language processing and computer vision, enabling more intuitive and accessible content creation. This advancement could lead to practical applications in various fields, such as digital art, advertising, and education, where users can generate tailored visual content effortlessly. Furthermore, it could inspire future research into more sophisticated generative models that integrate multimodal data, enhancing our understanding of how language and visual perception interact. | This problem is significant as it bridges the gap between natural language processing and computer vision, enhancing applications in creative industries such as content creation, advertising, and virtual reality. Improving the quality and diversity of generated images can lead to more accurate visual storytelling and better user experiences. Additionally, addressing fairness in AI-generated content is crucial for ethical deployment, ensuring that diverse demographic groups are accurately represented and not perpetuated by biases. |
| Why is it hard? | The challenges in solving this problem include the complexity of accurately interpreting natural language prompts and translating them into detailed visual representations. Naive approaches may fail due to the inherent ambiguity in language and the difficulty of capturing intricate details like lighting, shadows, and textures in generated images. Additionally, achieving a balance between photorealism and adherence to the prompt requires overcoming technical obstacles related to model training, data representation, and the integration of different generative techniques. | The challenges arise from the complexity of aligning nuanced textual descriptions with visual representations, which often leads to issues of semantic drift and loss of detail. Existing models may struggle with maintaining high fidelity and diversity in generated images, particularly when trained on biased datasets that lack representation. Furthermore, the computational demands of current methods can hinder accessibility and scalability, making it difficult to experiment with more efficient training paradigms. |
| Why hasn't it been solved before? | Previous research has primarily focused on either generating images from text or achieving photorealism, but not both simultaneously. Limitations in existing models often stem from their inability to effectively combine the strengths of text-conditional and unconditional image generation techniques. Barriers such as insufficient training data, lack of robust evaluation metrics, and the complexity of integrating multiple generative approaches have hindered progress. Our approach aims to address these gaps by leveraging diffusion models, classifier guidance, and CLIP to enhance both the quality and relevance of generated images. | Previous research has often focused on improving either the quality of generated images or the alignment between text and images, but few have successfully integrated these aspects into a unified framework. Many existing models rely on complex architectures that require extensive labeled datasets and computational resources, limiting their practical application. Additionally, insufficient emphasis on fairness and representation in training datasets has hindered progress in creating inclusive generative models. |
| What are the key components of my approach and result? | Our proposed methodology involves using diffusion models augmented with classifier-free guidance and CLIP for image generation. We will train our model on a diverse dataset of images and corresponding text prompts, employing metrics such as Inception Score and Fréchet Inception Distance to evaluate the quality of generated images. The expected outcomes include the generation of high-fidelity images that accurately reflect the content of the prompts, along with improved performance in detecting and filtering out undesired content, such as images containing people, thereby enhancing the model's applicability in real-world scenarios. | We propose a novel framework that combines a Denoising Diffusion Probabilistic Model (DDPM) with contrastive learning techniques to enhance text-to-image synthesis. This approach will utilize a balanced dataset that includes diverse demographic representations, focusing on effective text-image feature alignment. We will evaluate our model using metrics such as Inception Score (IS) and Fréchet Inception Distance (FID) to assess both image quality and fairness. The expected outcomes include high-quality, diverse images that accurately reflect the input text while demonstrating equitable performance across various demographic categories, ultimately setting a new standard for fairness and efficiency in generative models. |

Table 21: Case study on paper writing results of HIGHIMPACTPAPERBENCH.

| REVIEWBENCH **Task** | | |
|---|---|---|
| T-MARS: Improving Visual Representations by Circumventing Text Feature Learning | | |
| REVIEWBENCH **Output** | | |
| **Attribute** | **Reference Review** | **Generated Review** |
| **Strength** | - Understanding how to better design datasets is an important and timely problem, with many open challenges remaining.
- This paper presents a significant step forward in that direction, making it of interest to many in the community.
- The research is likely to have a substantial impact on practitioners looking to build better multimodal models.
- The overall idea is straightforward and easy to understand.
- The paper presents good empirical results demonstrating increased accuracy when the proposed method is used to filter data.
- The filtering method was applied to the LAION dataset, leading to the development of a newly curated dataset.
- Trained models on the curated dataset were tested across a decent number of downstream tasks.
- The paper's findings align with other works emphasizing the importance of data quality, such as [1].
- The motivation of this paper is reasonable.
- Some experimental results are impressive.
- Emphasizes the importance of data cleaning in deep learning, specifically addressing noise in image datasets.
- Identifies a significant issue in LAION's dataset, revealing that nearly 40% of images have text overlapping their captions.
- Employs a creative method to mask and restore text regions by using the average color of surrounding pixels, enhancing data quality.
- Introduces a similarity score calculation between images and captions as a novel filtering technique.
- Demonstrates robustness through extensive evaluation on multiple baselines, ranging from 2 million to 128 million images. | - Innovative Approach: The submission presents the T-MARS method, a novel data filtering technique that enhances visual representation learning by addressing text dominance in image-caption pairs, filling a critical gap in multimodal training.
- Significant Results: T-MARS demonstrates substantial accuracy improvements, outperforming competitive baselines by up to 6.5% on the DataComp benchmark, showcasing its effectiveness in real-world applications.
- Comprehensive Evaluation: The authors conduct extensive evaluations across multiple datasets, including LAION and DataComp, providing a thorough analysis of filtering metrics and validating the effectiveness of T-MARS.
- Acknowledgment of Limitations: The authors thoughtfully discuss potential biases introduced by data filtering and suggest future work to refine their approach, indicating a mature understanding of the research landscape.
- Strong Related Work Context: The paper effectively situates its contributions within existing literature, referencing key studies and demonstrating how T-MARS builds upon and diverges from prior methodologies.
- Technical Rigor: The methodology is well-detailed, allowing for replication, which is crucial for advancing research in multimodal language models.
- Clear Structure and Clarity: The paper is generally well-structured, with a logical flow that aids comprehension, although some sections could benefit from improved clarity. |
| **Weakness** | - The paper lacks large-scale experiments.
- Running large-scale CLIP pre-training experiments can be prohibitively expensive for many institutions.
- The authors present clear scaling trends that indicate their approach has great promise for larger scales.
- The motivation for the work is weak and lacks a theoretical analysis of why text-only images degrade visual learning compared to mislabeled data.
- Chapter 3 performs a manual analysis of 500 sample images from the LAION dataset to categorize them based on the correlation between image features and captions, but lacks metrics to quantify the representativeness of this sample within the entire dataset.
- Additional details provided in the appendix are appreciated; however, the work would benefit from more experiments, details, and analytics.
- A larger random sample with statistical estimates of error bars on proportions is recommended.
- Chapter 6 is difficult to follow; a rewrite to better present the experiments would be beneficial.
- The methodology relies on the CLIP score for filtering, which can be noisy and introduce additional biases, and the current version of the paper does not address this concern.
- The writing of this paper is somewhat obscure, making it difficult to follow.
- Is it possible to directly remove all the text in the images? This may help reduce distractions.
- It would be better to conduct experiments on more datasets, in addition to LAION.
- The proposed method has only been evaluated using accuracy as a metric, which may not provide a comprehensive understanding and could introduce bias for other important metrics.
- The overlap of text with the image caption may hinder the learning of visual features. A dedicated subsection discussing various solutions to this issue could provide valuable insights for researchers, rather than relying solely on the masking technique. | - Clarity Issues: The writing lacks clarity in several sections, especially in the methodology, making it difficult for readers to understand the filtering algorithms (C-SSFT, C-RHO, T-MARS) and their operational mechanics.
- Baseline Relevance: The relevance of some baseline comparisons is questionable, with insufficient justification for selecting specific baselines like C-SSFT and C-RHO, which may mislead readers regarding the significance of T-MARS's performance.
- Novelty Concerns: The novelty of the T-MARS method is not convincingly articulated, as it does not sufficiently differentiate itself from existing methods like SemDeDup and DataComp, raising questions about its overall contribution to the field.
- Statistical Robustness: The reported accuracy gains (3.3% and 6.5%) lack comprehensive statistical validation, and the small pilot study sample size of 500 raises concerns about the generalizability of the findings.
- Superficial Limitations Discussion: The discussion of limitations is superficial, failing to explore the implications of potential biases introduced by data filtering on model performance.
- Insufficient Technical Details: The methods section lacks sufficient detail for replication, particularly regarding hyperparameter settings and specific configurations used in experiments, hindering reproducibility. |

Table 22: Case study on review writing results of REVIEWBENCH.

| RESEARCHTOWN **Input** | |
|---|---|
| **Researcher** | 4 × NLP researchers + 2 × astronomy researchers |
| **Paper** | MagicPIG: LSH Sampling for Efficient LLM Generation
Do LLMs exhibit human-like response biases? A Case study in survey design
A Survey of Large Language Models
WebArena: A Realistic Web Environment for Building Autonomous Agents
Gaia Data Release 3: Summary of the content and survey properties
Validation of SDSS-derived kinematic models for the Milky Way's disk and halo stars
The Simons Observatory: Science goals and forecasts |
| RESEARCHTOWN **Output** | |
| **Question** | **Answer** |
| What is the problem? | The specific research question addressed in this proposal is: How can kinematic modeling techniques from astrophysics be applied to analyze and understand the evolution of writing styles in large-scale linguistic datasets? |
| Why is it interesting and important? | This research bridges the gap between astrophysics and linguistics, providing a novel interdisciplinary approach to studying language evolution. By applying kinematic modeling, we can uncover dynamic shifts in writing styles that may reflect broader cultural and environmental influences. The implications for the research community are profound; this study could lead to new methodologies in linguistic analysis and enhance our understanding of language development across different families. Furthermore, the insights gained could inform practical applications in fields such as computational linguistics, sociolinguistics, and artificial intelligence, advancing the understanding of how language adapts and evolves in response to societal changes. |
| Why is it hard? | Solving this problem presents several challenges. Firstly, the complexity of language is inherently multifaceted, with numerous variables influencing writing styles, including cultural context, authorial intent, and historical events. Traditional linguistic analysis may not adequately capture these dynamic shifts, as it often relies on static models. Moreover, applying astrophysical kinematic models requires a deep understanding of both fields, necessitating interdisciplinary expertise. Naive approaches that merely replicate astrophysical models without adapting for linguistic nuances are likely to fail, as they may overlook critical contextual elements. Additionally, the sheer volume of linguistic data presents technical obstacles in terms of data processing and analysis, requiring robust machine learning algorithms to identify and classify transient phenomena accurately. |
| Why hasn't it been solved before? | Previous research has largely focused on static analyses of linguistic datasets, often missing the dynamic aspects of language evolution. Existing solutions have been limited by the lack of interdisciplinary approaches that integrate methodologies from other fields, such as astrophysics. Additionally, there has been a scarcity of comprehensive datasets that span significant temporal and cultural shifts in language use. Barriers such as insufficient computational resources and the complexity of developing suitable machine learning models have also hindered progress. This proposal distinguishes itself by employing advanced kinematic modeling techniques, which have not been previously applied to linguistic datasets, thus offering a fresh perspective and a more nuanced understanding of language evolution. |
| What are the key components of my approach and result? | The proposed methodology involves developing a framework that integrates kinematic modeling techniques with machine learning algorithms to analyze linguistic datasets. We will utilize large-scale corpora from diverse language families, focusing on historical texts to capture changes in syntax, vocabulary, and stylistic elements over time. Specifically, we will adapt kinematic models by defining linguistic "velocity" metrics that quantify shifts in writing style, analogous to how velocities are analyzed in astrophysics. Key metrics will include the frequency and distribution of linguistic features over time, allowing for the identification of transient phenomena in language use. Expected outcomes include the classification of dynamic shifts in writing styles and the elucidation of relationships between linguistic evolution and cultural or environmental influences. This research aims to produce a robust, interdisciplinary framework that enhances our understanding of language dynamics, paving the way for future studies in both linguistics and adjacent fields. Furthermore, we will address ethical considerations by ensuring respectful treatment of historical texts and acknowledging the limitations of drawing parallels between astrophysical phenomena and linguistic evolution. |

Table 23: Case study on using RESEARCHTOWN to write interdisciplinary research papers combining LLM and Astronomy. The idea creatively applies modeling techniques from astrophysics to explore how language styles evolve over time.

| RESEARCHTOWN Input | |
|---|---|
| **Researchers** | 2 × NLP researcher + 3 × Criminology researcher |
| **Paper** | MagicPIG: LSH Sampling for Efficient LLM Generation
Do LLMs exhibit human-like response biases? A Case study in survey design
A Survey of Large Language Models
WebArena: A Realistic Web Environment for Building Autonomous Agents
Collateral Consequences of Punishment: A Critical Review and Path Forward
Criminal organizations exhibit hysteresis, resilience, and robustness by balancing security and efficiency |
| RESEARCHTOWN Output | |
| **Question** | **Answer** |
| What is the problem? | The specific research question I aim to address is: How can a multimodal large language model that integrates qualitative narrative analysis with real-time speech translation effectively address the communication needs of communities affected by mass incarceration? |
| Why is it interesting and important? | Solving this problem holds significant implications for both the research community and society at large. The integration of qualitative narrative analysis with real-time speech translation can enhance our understanding of the complex social dynamics and personal experiences of individuals affected by mass incarceration. By capturing and translating haunting narratives that reflect the legacies of state violence, this research can contribute to the broader discourse on restorative justice and community resilience. Furthermore, the outcomes of this study could inform emergency response strategies, ensuring that vulnerable populations receive appropriate support during crises. This advancement in knowledge not only has the potential to improve communication practices but also to influence policy and practical applications in social justice and community support initiatives. |
| Why is it hard? | The challenges in addressing this problem are multifaceted. First, the integration of qualitative narrative analysis with real-time speech translation requires sophisticated algorithms that can accurately interpret and convey nuanced meanings, particularly in emotionally charged narratives. Naive approaches may fail to capture the socio-cultural context essential for effective communication, leading to misinterpretations and potentially harmful consequences. Additionally, there are technical hurdles in processing multimodal data—combining text, audio, and contextual cues—while ensuring the model remains sensitive to the lived experiences of marginalized communities. The theoretical complexities of understanding and representing narratives of trauma and resilience further complicate the development of a robust model. |
| Why hasn't it been solved before? | Previous research has often focused on either qualitative narrative analysis or speech translation in isolation, overlooking the critical intersection of these fields. Existing solutions have been limited by their inability to adapt translations based on socio-cultural contexts, which is vital for accurately conveying personal stories from affected communities. Barriers such as a lack of interdisciplinary collaboration and insufficient datasets that reflect the experiences of those impacted by mass incarceration have also hindered progress. My approach differs by explicitly incorporating narrative analysis into the translation process and prioritizing socio-cultural contextualization, thus addressing the gaps in prior work and providing a more holistic solution. |
| What are the key components of my approach and result? | My proposed methodology involves developing a multimodal large language model that utilizes advanced natural language processing (NLP) techniques for narrative analysis and real-time speech translation. I will employ a mixed-methods approach, combining qualitative data from interviews with impacted individuals and quantitative data from existing linguistic resources. The model will be trained on a diverse dataset that captures a wide range of narratives related to mass incarceration, ensuring representation of various socio-cultural contexts. Metrics for evaluation will include translation accuracy, contextual appropriateness, and user satisfaction among community members, as well as measures of community resilience and restorative justice outcomes through longitudinal studies. The expected outcomes include a functional model that enhances communication in emergency scenarios, informs restorative justice practices, and ultimately contributes to community resilience by empowering voices that have historically been marginalized. |

Table 24: Case study on using RESEARCHTOWN to write interdisciplinary research papers combining LLM and Criminology. The idea creatively utilizes a multimodal LLM to integrate qualitative narrative analysis with real-time speech translation, aiming to enhance communication for communities impacted by mass incarceration.

| RESEARCHTOWN Input | |
| --- | --- |
| **Researchers** | 4 × NLP researcher + 2 × Biology researcher |
| **Paper** | MagicPIG: LSH Sampling for Efficient LLM Generation
Do LLMs exhibit human-like response biases? A Case study in survey design
A Survey of Large Language Models
WebArena: A Realistic Web Environment for Building Autonomous Agents
Estimating Waiting Distances Between Genealogy Changes under a Multi-Species Extension of the Sequentially Markov Coalescent
The interplay between microbial communities and soil properties efficiency
Ancient orogenic and monsoon-driven assembly of the world's richest temperate alpine flora |

| RESEARCHTOWN Output | |
| --- | --- |
| **Question** | **Answer** |
| What is the problem? | The specific research question we aim to address is: How can a novel framework that integrates genealogical variation principles with generative retrieval techniques be developed to analyze and enhance the evolution of language model performance? |
| Why is it interesting and important? | Solving this problem is significant because it could provide the research community with a deeper understanding of the interplay between training dataset composition and language model capabilities. By elucidating the structural and thematic relationships among documents, this framework can lead to advancements in both the performance and fairness of language models. Furthermore, the insights gained could inform future research in natural language processing (NLP) by providing methodologies to mitigate biases inherent in language models, thus enhancing their applicability in real-world scenarios. This research has the potential to influence how language models are trained and evaluated, ultimately improving their effectiveness in diverse applications, including automated translation, content generation, and conversational agents. |
| Why is it hard? | The challenges in solving this problem stem from the complexities involved in modeling the genealogical relationships among training datasets and their temporal evolution. Naive approaches may overlook the intricate interactions between data composition and model performance, leading to a superficial understanding of the underlying mechanisms. Technical obstacles include the need for advanced statistical methods to capture coalescent-like relationships, as well as the computational demands of analyzing large and diverse datasets. Theoretically, the lack of established frameworks that unify genealogical principles with generative retrieval techniques poses a significant barrier. Additionally, practical challenges in collecting, organizing, and analyzing the vast amounts of data required for comprehensive insights further complicate the endeavor. |
| Why hasn't it been solved before? | Previous research has largely focused on either the performance of language models or the biases present in training datasets, but rarely have these aspects been integrated in a cohesive framework. Limitations in existing studies often include a narrow focus on individual datasets or specific model architectures without considering the broader genealogical context. Barriers such as the absence of interdisciplinary approaches that combine computational linguistics, evolutionary theory, and data science have prevented this problem from being effectively addressed until now. Our approach differs by explicitly modeling the relationships between datasets and their impact on language model evolution, thereby bridging these critical gaps and offering a more holistic understanding of language model performance. |
| What are the key components of my approach and result? | Our proposed methodology involves developing a framework that applies genealogical variation principles through a coalescent-like model to analyze the training datasets of language models. We will utilize a diverse dataset encompassing various domains and document types to capture shifts in token frequency and thematic representation. The methodology will incorporate generative retrieval techniques to enhance the analysis of data relationships. The primary metrics for evaluation will include model performance indicators such as perplexity, accuracy, and bias detection scores. Expected outcomes include a comprehensive understanding of how data composition influences language model capabilities, along with practical guidelines for optimizing training datasets to improve model performance and mitigate biases in real-world applications. Additionally, we will address potential data collection challenges by leveraging existing datasets and collaborating with institutions to ensure a representative sample. We will also outline a clear timeline and roadmap for the research process in the revised proposal, ensuring feasibility and accountability in our approach. |

Table 25: Case study on using RESEARCHTOWN to write interdisciplinary research papers combining LLM and Biology. The idea integrates patterns of inherited traits with generated retrieval methods to study and improve how language models grow and perform over time.

| RESEARCHTOWN Input | |
|---|---|
| **Researchers** | 3 × System researcher + 2 × Biology researcher |
| **Paper** | Zeno++: Robust Fully Asynchronous SGD
Comprehensive Exploration of Synthetic Data Generation: A Survey
AdaScale SGD: A User-Friendly Algorithm for Distributed Training
Efficient Training of Large Language Models on Distributed Infrastructures: A Survey
Estimating Waiting Distances Between Genealogy Changes under a Multi-Species Extension of the Sequentially Markov Coalescent
The interplay between microbial communities and soil properties efficiency
Ancient orogenic and monsoon-driven assembly of the world's richest temperate alpine flora |

| RESEARCHTOWN Output | |
|---|---|
| **Question** | **Answer** |
| What is the problem? | The specific research question I aim to address is: How can a hybrid framework that integrates genealogical variation models optimize multi-objective breeding programs for climate-resilient crops while enhancing distributed communication protocols among IoT devices in smart agricultural environments? |
| Why is it interesting and important? | Solving this problem holds significant implications for the research community, particularly in agricultural science, genetics, and IoT technology. The development of a framework that optimizes breeding programs for climate-resilient crops directly addresses the pressing global challenge of food security and agricultural sustainability in the face of climate change. This research can influence future studies by providing a comprehensive model that combines genetic insights with technological advancements, promoting interdisciplinary approaches. Furthermore, integrating genealogical variation models with IoT communication protocols can lead to practical applications such as more efficient resource management in agriculture, improved crop yields, and enhanced adaptability to environmental changes, thus contributing to the broader goals of sustainable development. |
| Why is it hard? | The complexity of this problem arises from several intertwined challenges. First, integrating genealogical variation models into breeding programs requires a nuanced understanding of genetic relationships and their influence on crop resilience. Traditional breeding methods often lack the adaptability needed to respond to rapid environmental changes, and naive approaches may overlook critical genetic diversity, which is essential for resilience. Additionally, optimizing IoT communication protocols in agricultural settings involves overcoming technical obstacles such as ensuring network reliability, achieving fault tolerance, and maintaining load balancing—all of which are complicated by the dynamic nature of agricultural environments. The hybrid framework must effectively address these challenges, ensuring that both genetic strategies and IoT protocols work synergistically without compromising either aspect. |
| Why hasn't it been solved before? | Previous research has largely focused on either genetic optimization for crop resilience or improving communication protocols in IoT systems, with few studies attempting to integrate these two domains. Existing solutions often operate in silos, failing to leverage potential synergies between genetic models and IoT frameworks. Barriers to integration include a lack of interdisciplinary collaboration and insufficient data on the interaction between genetic diversity and real-time IoT communications. My approach differs from prior work by explicitly combining genealogical insights with soft-hard functions (SHFs) to create a unified framework that addresses both breeding optimization and communication efficiency, thus filling a critical gap in current research. |
| What are the key components of my approach and result? | I propose a methodology that involves developing a hybrid framework incorporating genealogical variation models and soft-hard functions (SHFs). This framework will utilize comprehensive datasets, including genetic information from diverse crop species (sourced from databases like the USDA and global gene banks) and real-time data from IoT devices deployed in agricultural settings (such as environmental sensors and drones). The performance metric for the breeding programs will focus on genetic diversity, adaptability, and crop yield, while the IoT communication efficiency will be assessed based on fault tolerance and data exchange rates. The expected outcomes include a robust model that optimizes breeding strategies for climate resilience and enhances the effectiveness of distributed IoT communications, ultimately contributing to sustainable agricultural practices and improved resilience to climate change. Additionally, I will discuss potential limitations and challenges in implementing the hybrid framework in real-world settings, such as technical constraints and scalability of the solutions, to ensure a comprehensive understanding of the proposed approach. |

Table 26: Case study on using RESEARCHTOWN to write interdisciplinary research papers combining System and Biology. The idea attempts to build a hybrid system combining genetic variation models and IoT protocols for resilient crop breeding, but it risks being overshadowed by excessive terminologies.

| RESEARCHTOWN **Input** | |
|---|---|
| **Researchers** | 4 × Math researcher + 3 × Criminology researcher |
| **Paper** | Singular Hodge theory for combinatorial geometries
Adam: A Method for Stochastic Optimization
On the monotonicity of the Fisher information for the Boltzmann equation
Varieties of general type with many vanishing plurigenera, and optimal sine and sawtooth inequalities
Collateral Consequences of Punishment: A Critical Review and Path Forward
Criminal organizations exhibit hysteresis, resilience, and robustness by balancing security and efficiency |

| RESEARCHTOWN **Output** | |
|---|---|
| **Question** | **Answer** |
| What is the problem? | How can we develop an adaptive stochastic optimization framework that effectively solves stochastic partial differential equations (SPDEs) by integrating concepts from the Adam algorithm, while accommodating time-varying boundary conditions and optimizing numerical methods in real-time? |
| Why is it interesting and important? | Solving SPDEs is crucial in various fields such as physics, finance, and engineering, where systems are influenced by randomness and uncertainty. An effective framework that adapts to time-varying boundary conditions can significantly enhance the accuracy and efficiency of numerical solutions, leading to deeper insights into the behavior of dynamic systems. This research has the potential to advance the understanding of critical phenomena, such as phase transitions and turbulence, which are vital for modeling real-world scenarios. Moreover, the integration of adaptive stochastic optimization methods and neural architecture search can pave the way for future research by providing a versatile toolset that can be applied to a wide range of complex problems, ultimately leading to practical applications in predictive modeling and risk assessment. |
| Why is it hard? | The challenge in addressing SPDEs lies in their inherent complexity and the stochastic nature of the equations, which often makes traditional numerical methods insufficient. Naive approaches may fail due to their inability to adapt to changing conditions or to handle high-dimensional spaces effectively. Additionally, the presence of singularities and complex boundary conditions complicates the mathematical landscape, requiring sophisticated techniques for accurate approximation. The integration of deep learning and adaptive optimization introduces further challenges, such as the need for extensive computational resources and robust training methodologies, which must be carefully designed to ensure convergence and reliability. Specifically, overfitting poses a significant concern, as the model's performance may degrade when exposed to unseen data or extreme conditions. |
| Why hasn't it been solved before? | Previous research on SPDEs has primarily focused on either deterministic methods or has not adequately incorporated adaptive techniques that account for real-time data. Many existing solutions lack the flexibility needed to address the dynamic nature of boundary conditions, leading to limitations in their applicability. Additionally, the integration of deep learning with traditional numerical methods is still an emerging area, with few studies exploring the optimization of neural architectures specifically for SPDEs. My approach distinguishes itself by leveraging the Adam optimization algorithm alongside neural architecture search to create a framework that automatically identifies optimal strategies. This integration is crucial as it allows for the real-time adjustment of numerical methods, thus filling the gaps left by prior work and overcoming barriers related to adaptability and efficiency. |
| What are the key components of my approach and result? | My proposed methodology involves developing an adaptive stochastic optimization framework that employs the Adam algorithm's moment estimation to optimize numerical methods for SPDEs in real-time. The framework will utilize a comprehensive dataset of simulated SPDE solutions, incorporating various boundary conditions and singularity scenarios. Key metrics for evaluation will include convergence rates, computational efficiency, and accuracy of the simulations. Expected outcomes include enhanced numerical strategies that automatically adapt to dynamic conditions, improved understanding of critical phenomena in SPDEs, and a significant reduction in computational time compared to traditional methods, ultimately leading to more effective simulations in high-dimensional contexts. Additionally, I plan to provide a clear outline of the theoretical foundations supporting my methodologies and address potential limitations such as data availability and noise, ensuring a comprehensive understanding of the framework's applicability to real-world problems. |

Table 27: Case study on using RESEARCHTOWN to write interdisciplinary research papers combining Math and Criminology. Due to the two fields being too far apart conceptually, the generated idea primarily focuses on mathematical methods, with minimal incorporation of criminology insights.

| RESEARCHTOWN Input | |
|---|---|
| **Researchers** | 4 × NLP researcher + 3 × Criminology researcher + 4 × Math researcher |
| **Paper** | MagicPIG: LSH Sampling for Efficient LLM Generation
Do LLMs exhibit human-like response biases? A Case study in survey design
A Survey of Large Language Models
WebArena: A Realistic Web Environment for Building Autonomous Agents
Singular Hodge theory for combinatorial geometries
On the monotonicity of the Fisher information for the Boltzmann equation
Varieties of general type with many vanishing plurigenera, and optimal sine and sawtooth inequalities
Collateral Consequences of Punishment: A Critical Review and Path Forward
Criminal organizations exhibit hysteresis, resilience, and robustness by balancing security and efficiency |

| RESEARCHTOWN Output | |
|---|---|
| **Question** | **Answer** |
| What is the problem? | The specific research question I aim to address is: How can Fisher information be effectively utilized within the framework of stochastic partial differential equations (SPDEs) to model the dynamics of social networks in child welfare interventions, with a focus on ensuring transparency and accountability in AI-generated recommendations? |
| Why is it interesting and important? | Solving this problem is crucial as it has significant implications for the research community, particularly in the fields of social dynamics, child welfare, and artificial intelligence ethics. By developing a theoretical framework that integrates Fisher information and SPDEs, we can enhance our understanding of how social networks influence child welfare interventions and the subsequent outcomes for families. This research could lead to improved predictive models that not only inform interventions but also ensure ethical considerations are embedded in decision-making processes. Advancing knowledge in this area can facilitate the development of decision support tools that promote transparency and accountability, ultimately safeguarding the quality of care provided to vulnerable populations. |
| Why is it hard? | The challenges in addressing this problem stem from the inherent complexities of modeling social networks using SPDEs, particularly in the context of child welfare interventions. Traditional approaches may oversimplify the dynamics at play, failing to capture the nuanced relationships and interactions within these networks. Additionally, integrating Fisher information requires sophisticated mathematical formulations that can accurately reflect the stability and regularity of solutions in complex systems. Technical obstacles include the need for robust statistical methods to analyze the interplay between network structures and ethical considerations, as well as the difficulty in ensuring that AI-generated recommendations are interpretable and traceable by stakeholders. Furthermore, operationalizing Fisher information within SPDEs necessitates clear methodologies for parameter estimation and model validation, which are non-trivial tasks. |
| Why hasn't it been solved before? | Previous research has typically focused on either the mathematical modeling of social networks or the ethical implications of AI in child welfare, but rarely have these domains been integrated. Existing solutions often lack a comprehensive framework that combines statistical mechanics with SPDEs, leading to a limited understanding of the dynamics involved. Barriers to progress include a fragmented approach to research, where interdisciplinary collaboration has been minimal. My approach differs by explicitly linking Fisher information to SPDEs while emphasizing the ethical dimensions of AI in sensitive domains, thereby filling a critical gap in the literature. Additionally, the lack of systematic integration of stakeholder perspectives in existing models has hindered the development of practical decision support tools. |
| What are the key components of my approach and result? | My proposed methodology involves developing a theoretical framework that employs Fisher information to derive SPDEs modeling the dynamics of social networks in child welfare contexts. This will include specific steps to operationalize Fisher information, such as defining appropriate metrics to quantify uncertainty and variability in the network dynamics. The analysis will utilize real-world datasets from child welfare agencies to validate the model, with metrics focusing on stability, regularity, and predictive accuracy of AI-generated recommendations. I will also address potential limitations in data collection by outlining strategies for ethical data access and stakeholder collaboration. Expected outcomes include a robust model that characterizes the dynamics of social networks, insights into ethical implications of AI use, and a decision support tool that enhances transparency and accountability in interventions. This framework aims to provide stakeholders with the ability to trace decision provenance while ensuring high-quality care for families, thereby addressing concerns raised in the reviews. |

Table 28: Case study on using RESEARCHTOWN to write interdisciplinary research papers combining LLM, Math, and Criminology. The idea focuses on modeling social network dynamics in child welfare interventions by integrating a series of mathematical concepts. The practicability of the method remains questioned due to its heavy reliance on complex mathematical frameworks.

| RESEARCHTOWN **Input** | |
|---|---|
| **Researchers** | 3 × System researcher + 2 × Criminology researcher + 2 × Biology researcher |
| **Paper** | Zeno++: Robust Fully Asynchronous SGD
Comprehensive Exploration of Synthetic Data Generation: A Survey
AdaScale SGD: A User-Friendly Algorithm for Distributed Training
Efficient Training of Large Language Models on Distributed Infrastructures: A Survey
Estimating Waiting Distances Between Genealogy Changes under a Multi-Species Extension of the Sequentially Markov Coalescent
The interplay between microbial communities and soil properties efficiency
Ancient orogenic and monsoon-driven assembly of the world's richest temperate alpine flora
Collateral Consequences of Punishment: A Critical Review and Path Forward
Criminal organizations exhibit hysteresis, resilience, and robustness by balancing security and efficiency |

| RESEARCHTOWN **Output** | |
|---|---|
| **Question** | **Answer** |
| What is the problem? | How do "ghost networks" within marginalized communities, resulting from systemic disruptions and mass incarceration, influence perceptions of safety, justice, and social cohesion, and what role do policing technologies play in this dynamic? |
| Why is it interesting and important? | This research is significant as it seeks to illuminate the often-overlooked social dynamics that arise in marginalized communities due to mass incarceration. By exploring the concept of g̈host networks,ẗhis study will provide insights into how invisible social ties and resources can affect community resilience and perceptions of justice. The findings will have broader implications for the research community by advancing knowledge on the interplay between social networks and policing technologies, and their collective impact on community well-being. Moreover, this research could lead to practical applications in criminal justice reform, guiding the development of AI frameworks that prioritize ethical considerations and promote equitable outcomes, thereby influencing future policy decisions and community rehabilitation efforts. |
| Why is it hard? | Addressing this problem is challenging due to the complexities of social dynamics in marginalized communities, where the effects of mass incarceration and systemic violence create intricate, often hidden networks. Naive approaches may fail to capture the nuanced interactions within these communities, as traditional quantitative methods may overlook the qualitative dimensions of social ties and community resilience. Additionally, there are technical obstacles in accurately mapping these "ghost networks", as well as theoretical challenges in integrating social impact metrics with AI applications. The deployment of policing technologies further complicates the landscape, as these tools can exacerbate existing inequalities, making it difficult to disentangle their effects from those of community dynamics. |
| Why hasn't it been solved before? | Previous research has often focused on the direct consequences of mass incarceration, neglecting the subtler implications of social networks and the role of policing technologies. Limitations in existing studies include a lack of mixed-methods approaches that combine quantitative data with qualitative insights, resulting in an incomplete understanding of community dynamics. Barriers such as insufficient community engagement and a lack of interdisciplinary collaboration have also hindered progress. My approach differs by integrating participatory mapping and qualitative interviews to capture the richness of community experiences, thus providing a more comprehensive analysis of the interplay between social networks, resilience, and policing technologies. Additionally, I will operationalize "ghost networks" by defining specific indicators such as social ties, resource accessibility, and community engagement, allowing for a more precise identification and measurement. |
| What are the key components of my approach and result? | My proposed methodology will utilize a mixed-methods approach that combines participatory mapping to visualize the "ghost networks" and qualitative interviews to gather in-depth insights from community members. The dataset will consist of both spatial data from community mapping exercises and qualitative data from interviews with residents and local stakeholders. Metrics will include social cohesion indices, perceptions of safety and justice, and indicators of community resilience, analyzed through natural language processing techniques to assess public sentiment. I will implement a stratified sampling strategy for qualitative interviews to ensure representation across different demographics and experiences. The expected outcomes include a nuanced understanding of how these hidden networks affect community dynamics and the development of a multi-objective optimization framework that integrates ethical considerations and social impact metrics into AI applications, ultimately guiding responsible algorithm design in criminal justice contexts. Furthermore, I will establish clear criteria for measuring success in terms of practical applications, including policy recommendations and community engagement strategies, while addressing ethical considerations throughout the research process to enhance feasibility and acceptance within marginalized communities. |

Table 29: Case study on using RESEARCHTOWN to write interdisciplinary research papers combining System, Biology, and Criminology. The idea investigates "ghost networks" in marginalized communities, exploring how systemic disruptions and mass incarceration affect perceptions of safety, justice, and social cohesion while incorporating the role of policing technologies in this dynamic.

| RESEARCHTOWN **Input** | |
|---|---|
| **Researchers** | 4 × NLP researcher + 2 × Criminology researcher + 2 × Biology researcher |
| **Paper** | MagicPIG: LSH Sampling for Efficient LLM Generation
Do LLMs exhibit human-like response biases? A Case study in survey design
A Survey of Large Language Models
WebArena: A Realistic Web Environment for Building Autonomous Agents
Estimating Waiting Distances Between Genealogy Changes under a Multi-Species Extension of the Sequentially Markov Coalescent
The interplay between microbial communities and soil properties efficiency
Ancient orogenic and monsoon-driven assembly of the world's richest temperate alpine flora
Collateral Consequences of Punishment: A Critical Review and Path Forward
Criminal organizations exhibit hysteresis, resilience, and robustness by balancing security and efficiency |

| RESEARCHTOWN **Output** | |
|---|---|
| **Question** | **Answer** |
| What is the problem? | The specific research question I aim to address is: How can an interdisciplinary online platform effectively mitigate the collateral consequences of online toxicity, specifically focusing on its impacts on the mental health and community well-being of marginalized populations affected by systemic violence and discrimination? |
| Why is it interesting and important? | This research is crucial because online toxicity, particularly hate speech and harassment, adversely affects marginalized communities, compounding existing societal inequalities. By developing a platform that not only detects and mitigates hate speech in real-time but also integrates user-reported impacts, we can significantly advance the understanding of how digital discourse influences mental health and community dynamics. This study will provide valuable insights for researchers and practitioners, promoting future investigations into the psychological effects of online interactions and informing interventions that foster inclusivity. The anticipated outcomes include a comprehensive dataset that captures the multifaceted impacts of online toxicity, which could lead to innovative machine learning models and strategies for creating safer online environments. |
| Why is it hard? | Solving this problem involves several complexities. First, accurately detecting and categorizing online toxicity is challenging due to the nuanced nature of language, context, and cultural differences. Naive approaches that rely solely on keyword filtering may fail to capture the subtleties of hate speech, leading to false negatives or positives. Additionally, understanding the psychological and societal impacts requires robust qualitative data from affected individuals, which is difficult to obtain and analyze systematically. There are also technical hurdles in integrating diverse datasets, ensuring user privacy, and developing machine learning models that can effectively contextualize and respond to the unique experiences of marginalized groups. |
| Why hasn't it been solved before? | Previous research has primarily focused on either automated hate speech detection or the psychological impacts of online toxicity, often in isolation. There is a notable gap in interdisciplinary approaches that combine these perspectives while specifically addressing marginalized communities. Existing solutions have been limited by a lack of comprehensive datasets that reflect both the historical narratives of systemic violence and contemporary online interactions. Barriers such as insufficient collaboration between tech developers and social scientists, as well as the challenges of gathering user-reported data, have prevented a holistic approach to this issue. My approach differs by integrating qualitative insights with quantitative data, which will provide a richer understanding of the problem and inform more effective interventions. |
| What are the key components of my approach and result? | My proposed methodology involves developing an online platform that employs natural language processing (NLP) algorithms to detect hate speech in real-time while incorporating user-reported data on mental health impacts and community well-being. The dataset will be built through surveys and feedback mechanisms targeting marginalized communities, ensuring diverse representation. Key metrics will include the frequency and severity of reported incidents, psychological distress levels, and community cohesion indicators. To address the concerns around user privacy, the platform will implement robust encryption and anonymization techniques during data collection and storage, ensuring sensitive data is protected. Additionally, we will establish clear protocols for data usage and inform users about how their data will contribute to research while maintaining confidentiality. The anticipated outcomes include a validated dataset that captures the interplay between online toxicity and its collateral consequences, contributing to the development of machine learning models that can provide contextual analysis and tailored intervention strategies. This platform aims to serve as a resource for researchers, mental health professionals, and community advocates in their efforts to create a safer and more inclusive online environment. |

Table 30: Case study on using RESEARCHTOWN to write interdisciplinary research papers combining LLM, Biology, and Criminology. The idea offers a novel interdisciplinary approach that developing an online platform that detects online toxicity in real-time while addressing its societal impacts on marginalized communities.

| RESEARCHTOWN Input | |
|---|---|
| **Researchers** | 2 × Astronomy researcher + 3 × Criminology researcher + 2 × Biology researcher |
| **Paper** | Gaia Data Release 3: Summary of the content and survey properties
Validation of SDSS-derived kinematic models for the Milky Way's disk and halo stars
The Simons Observatory: Science goals and forecasts
Estimating Waiting Distances Between Genealogy Changes under a Multi-Species Extension of the Sequentially Markov Coalescent
The interplay between microbial communities and soil properties efficiency
Ancient orogenic and monsoon-driven assembly of the world's richest temperate alpine flora
Collateral Consequences of Punishment: A Critical Review and Path Forward
Criminal organizations exhibit hysteresis, resilience, and robustness by balancing security and efficiency |

| RESEARCHTOWN Output | |
|---|---|
| **Question** | **Answer** |
| What is the problem? | The specific research question I aim to address is: How can machine learning techniques be integrated with genomic approaches to enhance astrometric calibration of celestial measurements by identifying genetic markers associated with adaptive traits in response to environmental stressors within the Viburnum clade?" |
| Why is it interesting and important? | Solving this problem holds significant implications for both the research community and practical applications in ecology and astronomy. By developing a framework that correlates genetic adaptations in plant species with celestial mapping precision, we can advance our understanding of how environmental factors influence biological and astronomical systems. This research could lead to improved methodologies for astrometric calibration, which is crucial for accurate celestial measurements that inform various fields, including astrophysics and navigation. Furthermore, this study encourages future interdisciplinary collaborations between genomic research and astronomical sciences, contributing to a more nuanced understanding of ecological and evolutionary dynamics. |
| Why is it hard? | Addressing this problem is complex due to several interrelated challenges. First, integrating machine learning with genomic data requires sophisticated algorithms capable of handling high-dimensional datasets while minimizing overfitting, especially given the unique characteristics of genomic data such as sparsity and noise. Second, the environmental stressors affecting hybridization dynamics in the Viburnum clade are multifaceted, making it difficult to isolate specific genetic markers linked to adaptive traits. Naive approaches may fail because they often overlook the intricate relationships between genetic, ecological, and astronomical factors. Additionally, ensuring that the genomic data accurately reflects the phenotypic adaptations observed in response to celestial measurements demands robust validation methods that can bridge both domains, which is a significant technical challenge. |
| Why hasn't it been solved before? | Previous research has largely focused on either genomic studies of plant species or the calibration of celestial measurements, with limited interdisciplinary efforts to merge these areas. A significant gap exists in understanding how ecological pressures influence genetic adaptations and how these adaptations can be quantitatively linked to systematic errors in astrometric measurements. Barriers such as a lack of integrated datasets and the absence of frameworks that facilitate cross-disciplinary analysis have hindered progress. My approach differs from prior work by explicitly connecting genetic markers and environmental stressors to astrometric calibration, utilizing machine learning to uncover patterns that have previously gone unexamined. The absence of a clear framework for quantitative integration of these domains has also contributed to the lack of progress in this area. |
| What are the key components of my approach and result? | My proposed methodology involves a multi-step framework that includes: (1) collecting genomic data from various Viburnum species, focusing on environmental stressors that influence hybridization dynamics, with a stratified sampling strategy to ensure representation across different habitats; (2) employing advanced machine learning algorithms such as Random Forests and Gradient Boosting Machines, tailored to handle the high dimensionality and sparsity of genomic data; (3) correlating identified genetic markers with systematic errors in astrometric measurements using celestial mapping datasets, while addressing potential challenges related to data quality and availability; and (4) validating the findings through robust statistical methods, including cross-validation and permutation tests, to ensure that the identified genetic markers are indeed linked to adaptive traits. The expected outcomes include the identification of key genetic markers that can predict adaptive responses, improved calibration techniques for celestial measurements, and a comprehensive model that enhances our understanding of the interplay between ecological factors and astronomical phenomena. This research aims to contribute significantly to both ecological and astronomical fields, providing a novel perspective on the integration of biological and celestial systems. |

Table 31: Case study on using RESEARCHTOWN to write interdisciplinary research papers combining Astronomy, Biology, and Criminology. Due to the significant conceptual gap between the three fields, the generated idea heavily leans on terminology accumulation.

| RESEARCHTOWN **Input** | |
|---|---|
| **Researchers** | 4 × NLP researcher + 2 × Astronomy researcher + 3 × Criminology researcher + 2 × Biology researcher |
| **Paper** | MagicPIG: LSH Sampling for Efficient LLM Generation
Do LLMs exhibit human-like response biases? A Case study in survey design
A Survey of Large Language Models
WebArena: A Realistic Web Environment for Building Autonomous Agents
Gaia Data Release 3: Summary of the content and survey properties
Validation of SDSS-derived kinematic models for the Milky Way's disk and halo stars
The Simons Observatory: Science goals and forecasts
Estimating Waiting Distances Between Genealogy Changes under a Multi-Species Extension of the Sequentially Markov Coalescent
The interplay between microbial communities and soil properties efficiency
Ancient orogenic and monsoon-driven assembly of the world's richest temperate alpine flora
Collateral Consequences of Punishment: A Critical Review and Path Forward
Criminal organizations exhibit hysteresis, resilience, and robustness by balancing security and efficiency |

| RESEARCHTOWN **Output** | |
|---|---|
| **Question** | **Answer** |
| What is the problem? | The specific research question I aim to address is: How does coded language in political discourse influence societal biases, and how can a Bayesian hierarchical model be employed to analyze this effect while simultaneously addressing observational biases in white dwarf population studies? |
| Why is it interesting and important? | Addressing this question is crucial for understanding the dynamics of political communication and its impact on public sentiment. By elucidating how specific linguistic patterns contribute to societal biases, this research can inform ethical communication practices in digital platforms, fostering a more informed public discourse. Moreover, integrating this analysis with astronomical data enhances the relevance of linguistic studies in a broader scientific context, potentially leading to advancements in both social sciences and astronomy. The findings could stimulate further interdisciplinary research, encouraging scholars to explore connections between language, bias, and observational methodologies, ultimately contributing to a more comprehensive understanding of both social and celestial phenomena. |
| Why is it hard? | The complexity of this problem arises from the multifaceted nature of language and its impact on societal biases, which is often influenced by context, culture, and individual interpretation. A naive approach might oversimplify the relationships between linguistic cues and public sentiment, neglecting the nuances of communication ethics. Additionally, integrating photometric and astrometric data from white dwarf populations introduces significant technical challenges, including the need to harmonize disparate datasets and account for observational biases that may skew results. The theoretical framework required to simultaneously analyze linguistic patterns and astronomical data is intricate, necessitating advanced statistical methods to ensure robust conclusions. |
| Why hasn't it been solved before? | Previous research has largely focused on either linguistic analysis of political discourse or astronomical studies of celestial objects, often in isolation. The gap lies in the lack of a unified framework that encompasses both domains, limiting the understanding of how language affects societal perceptions while also failing to address biases in astronomical observations. Barriers to solving this problem include the absence of interdisciplinary collaboration and the complexity of developing models that integrate diverse data types. My approach differs from prior work by proposing a Bayesian hierarchical model that simultaneously tackles these issues, offering a more holistic perspective on the interplay between language and bias, as well as improving the characterization of white dwarf properties. |
| What are the key components of my approach and result? | My proposed methodology involves developing a Bayesian hierarchical model that utilizes the Silent Signals dataset, which comprises a diverse range of political speeches and discourse across various platforms, including social media and traditional media. This dataset has been curated to include over 10,000 instances of coded language, with a focus on linguistic patterns that are statistically significant in shaping public sentiment. The model will be designed to quantify the effects of coded language on societal biases while integrating photometric and astrometric data from white dwarf populations to address observational biases in astronomical surveys. The metrics for evaluation will include the correlation between identified linguistic patterns and changes in public sentiment, alongside the accuracy of intrinsic property characterizations of white dwarfs. Expected outcomes include a clearer understanding of the relationship between language and bias, practical guidelines for ethical communication, and improved methodologies for astronomical data analysis, ultimately bridging the gap between social sciences and astronomy. Additionally, I will establish measurable benchmarks, such as a timeline of six months for initial findings and a year for comprehensive analysis, ensuring the project's rigor and accountability. |

Table 32: Case study on using RESEARCHTOWN to write interdisciplinary research papers combining LLM, Astronomy, Biology, and Criminology. Due to combining researchers and papers from too many diverse domains, the generated idea becomes an incoherent mix of terms without a clear focus or practical direction.

