# OpenReview forum: "ResearchTown: Simulator of Human Research Community"
_ICML.cc/2025/Conference — ICML 2025 poster_

### Official Review · Reviewer_t7zy · 2025-03-11

**Overall Recommendation:** 3

**Summary:**

The paper proposes ResearchTown, a multi-agent simulation framework for research community simulation. The research community is simplified as an agent-data graph, where researchers are modeled as agent nodes and research outputs (such as papers and reviews) as data nodes. The interactions, including paper reading, paper writing, and review writing, are modeled through a unified text-based message-passing framework named TextGNN. The main contributions claimed are: (1) a realistic simulation of collaborative research activities, (2) robustness in simulating complex multi-researcher and multi-paper interactions, and (3) the ability to inspire interdisciplinary research ideas. The authors validate the framework on ResearchBench, a benchmark evaluating ResearchTown via masked-node prediction tasks.

**Claims And Evidence:**

1. Claim 1: The paper claims that ResearchTown “provides a realistic simulation of collaborative research activities, including paper writing and review writing.” This claim is partially supported by evidence from their node-masking experiments. For a large set of existing papers, the system attempts to regenerate each paper’s content given its authors and references; the similarity between the generated text and the actual paper is reasonably high. These results suggest that the simulated agents can often reproduce or predict key elements of actual papers and reviews. However, the evidence for “realism” relies entirely on embedding-based similarity metrics – no human evaluation. The similarity measurement cannot capture other important aspects in paper rewriting and reviewing, such as logical consistency.
2. Claim 2: Robustness with Multiple Researchers and Diverse Papers. This one is supported by their ablation studies (Figure 4, 5).
3. Claim 3 – Generating Interdisciplinary Research Ideas. The author provides some qualitative evidence that ResearchTown can brainstorm non-obvious research questions by bridging fields, which aligns with the claim. However, the support is still limited: no systematic evaluation of “idea novelty” or quality is done. To better support this claim, future work should include a more rigorous assessment of the novelty and usefulness of generated ideas – perhaps by soliciting evaluations from domain experts on a sample of cross-domain proposals, etc.

**Essential References Not Discussed:**

--

**Experimental Designs Or Analyses:**

See the section of Methods and Evaluation Criteria.

**Methods And Evaluation Criteria:**

1. One concern to me is about data leakage. The author found that the ResearchTown framework performs notably better on impactful papers. This can be attributed to data leakage, i.e., the LLM may have seen the paper before during pretraining. The author tried to address the problem of information leakage by excluding any of the author's publications released after the target paper's publication. However, this is not enough, as the LLM can still recall details for released papers and reviewers, especially for those highly impactful papers.
2.  The paper only utilizes the similarity as evaluation metrics. However, the similarity measurement is not sufficient and cannot capture other important aspects in paper rewriting and reviewing, such as logical consistency, originality, etc.
3. Another concern lies in the absence of baseline comparisons beyond the ablations of their own approach.
4. Lastly, this is essentially a prompting engineering paper without any training or optimization involved. This is fine to me, but the author framed their method in an overly complex way. For example, equation (4), (5) is over complicated and hard to follow.

**Other Comments Or Suggestions:**

.

**Other Strengths And Weaknesses:**

Strengths:
1. Novel framework combining multi-agent LLMs and graphs for research community simulation.
2. RESEARCHTOWN can maintain robust simulation with multiple researchers and diverse papers.

Weakness:
1. The evaluation metric used, semantic similarity, is insufficient and may not fully capture novelty or logical consistency. Adding additional metrics or incorporating human expert evaluation could strengthen validation.
2. The authors state: "To prevent information leakage when simulating paper writing and review scenarios, we exclude any of the author’s publications released after the target paper’s publication year." However, I believe this measure is insufficient to prevent data leakage, as the LLM may have already been pretrained on these papers, including the target paper itself—especially for high-impact papers. The authors also observe that ResearchTown tends to achieve higher performance on high-impact papers, which could potentially be attributed to data leverage and LLM memorization, as these papers are more frequently included in the pretraining corpus.
3. A related concern is that the trends in Figures (4) and (5) could also be attributed to data leakage. Moreover, increasing the number of agents may further increase the likelihood that LLMs recall information from the pretraining corpus related to the target paper.
4. Given that this is essentially a prompting-based paper, the mathematical notation used may be somewhat confusing, making the paper more complex than necessary. In particular, Equations (4) and (5) are difficult to understand. I suggest simplifying the mathematical annotations and formulations to enhance clarity.

**Questions For Authors:**

Questions:
1. Why can paper reading be described as ‘inserting a new agent node’?
2. The framework performs significantly better on impactful papers that focus on analysis or tool development. Could this indicate data leakage? The LLM may have encountered the paper during pretraining.
3. 'Combining agent and data aggregation leads to a decrease in score differences, possibly because the presence of related papers causes reviewers to apply stricter novelty judgments' -> Are there any qualitative examples to support this?

**Relation To Broader Scientific Literature:**

The paper builds on prior work in multi-agent LLM frameworks, graph-based modeling of research communities, and text-attributed graphs (TAGs).

**Theoretical Claims:**

The paper does not present explicit theoretical proof.

---

> ### Author Rebuttal · Authors · 2025-04-01
>
> Thank you very much for your insightful and constructive comments. We ddress each of your comments in detail.
>
> **[Fine-grained evaluation with LLM and human]** Please check the same tag under **Reviewer YZ7h** for more human and LLM eval.
>
> **[Novelty+feasibility evaluation with LLM and human]** Please check the same tag under **Reviewer YZ7h** for more human and LLM eval.
>
> **[More baseline comparison]** Please check the same tag under **Reviewer 5ZWh** for baseline comparison.
>
> **[Data leakage concern]** For our main results (Table 1–2) and ablation studies (Figures 3–5), as noted in *Appendix C.2 (Line 813–815)*, we use NeurIPS 2024 and ICLR 2024 papers, which are *post-dated beyond GPT-4o-mini's* October 2023 knowledge cutoff. Thus, *data leakage is not a concern*. We also mask the full text during the simulation to avoid accidental exposure.
>
> For *HighImpactPaperBench* (Appendix C.3, Line 840–843), we use high-impact papers from the past decade as an *extreme-case test* for idea simulation. While some may exist in the LLM’s training data, this benchmark is separate from our main results and serves to explore how LLMs handle well-known concepts.
>
> Our similarity analysis shows that 55% of generated papers score between 0.65–0.75, and 18% exceed 0.75, indicating moderate to high alignment. Only 1% score below 0.45. These scores are *comparable to PaperBench* (Table 1), suggesting no abnormal inflation. Even famous papers like *VAE, GAN, LayerNorm* do not receive notably high scores, implying that *semantic similarity—not memorization based on citation relationships—drives the results*, especially for tool/benchmark papers, which naturally resemble their references more.
>
> **[Review writing performance analysis]** For the behavior behind Global-Agg of review writing results, we conduct fine-grained analysis on the difference between predicted score $S$ and real-world score $S^*$ on a subset of review writing tasks. The results show that *GPT-4o-mini consistently assigns lower scores than the real-world reviewers*, especially in the *Global-agg* setting, where the mean of (S - S*) is -1.47. In contrast, *Deepseek-v3* does not show this consistent bias, indicating a better performance on review writing.
>
> | Experimental Setting     | Mean of \|S - S*\| | Mean of (S - S*) | Std of S |
> | :----------------------- | ------------------ | ---------------- | -------- |
> | Global-agg (GPT-4o-mini) | 1.49               | **-1.47**        | 0.80     |
> | Global-agg (deepseek-v3) | 0.74               | **-0.02**        | 0.91     |
>
> Qualitatively, Global-agg reviews with GPT-4o-mini tend to provide *more specific and critical assessments*, particularly highlighting weaknesses related to *novelty* and *experiment* that are often missed in Self-agg settings, resulting in lower scores.
>
> Example 1: Global-agg provides a more detailed description of novelty concern and make the score lower.
>
> - *Global-agg*: *"The proposed method does not present a sufficiently innovative approach compared to existing frameworks. Many cited works, such as P2B and PTTR, already address similar challenges, and the submission fails to articulate clearly how it advances the state of the art.”* → Score: *4*
> - *Self-agg*: *“The novelty of the proposed methods is not clearly articulated. The paper does not convincingly demonstrate how the approach differs from existing methods or why it is a significant advancement in the field.”* → Score: *5*
>
> Example 2: Global-agg notices more weaknesses compared with self-agg.
>
> - *Global-agg*: *“Experimental Support: The experimental results presented are not robust. For instance, the claim of maintaining output quality is not backed by sufficient statistical analysis or metrics, making it difficult to assess the validity of the findings.”* --> Score: 4
> - *Self-agg*: Not mention this weakness. --> Score: 6
>
> **[Paper Reading as inserting new node]** As described in *Algorithm 1 (Line 220–237)*, the input to the simulation is the paper content itself. To initialize the agent for simulation, we first perform a *“paper reading”* step, which sets up the agent profile. This is implemented in *Line 6* of Algorithm 1. We interpret this as a *form of agent node insertion*—specifically, initializing the *text attributes* of an agent node based on an external paper. Thus, although it may not be an insertion in the structural sense, it serves the role of *initializing a new agent node* in the simulation graph.
>
> **[Math notation]** Thank you for pointing this out. Our intention was to provide a formal definition analogous to message-passing GNNs, such as in *Equation 3 (Line 145–148)*. However, due to the *heterogeneous nature* of our agent–data graph, we define *different aggregation functions* depending on the node types. We acknowledge that this may introduce complexity in notation. We will *simplify and clarify* the mathematical presentation in the revised version of the paper to improve readability.

---

### Official Review · Reviewer_5ZWh · 2025-03-12

**Overall Recommendation:** 3

**Summary:**

The paper introduces RESEARCHTOWN, a multi-agent framework for simulating human research communities using Large Language Models (LLMs). The key idea is to model the research community as an agent-data graph, where researchers (agent nodes) and papers (data nodes) interact through edges representing authoring, citations, and reviews. The authors propose TextGNN, a text-based message-passing mechanism that borrows concepts from Graph Neural Networks (GNNs), treating LLM-powered functions as GNN blocks to unify research activities (e.g., paper writing, reviewing) as operations on this graph. In addition, the paper proposes the RESEARCHBENCH benchmark, which evaluates the performance of RESEARCHTOWN by comparing the similarity between the generated papers and real papers. Experiments on the RESEARCHBENCH benchmark show that RESEARCHTOWN can provide a realistic simulation of collaborative research activities, with multi-agent setups outperforming single-agent configurations.

**Claims And Evidence:**

The claims are generally supported by evidence, but some require deeper scrutiny:
1. Realistic simulation of collaborative research: Supported by node-masking prediction results (similarity scores), but it's unclear what level of similarity score sufficiently demonstrates alignment with the claim of realistic simulation.
2. Inspiring interdisciplinary idea generation: While the paper provides examples (e.g., NLP + criminology), no quantitative evidence or human evaluation validates their novelty or feasibility.

**Essential References Not Discussed:**

I have no concerns regarding essential references that were not discussed.

**Experimental Designs Or Analyses:**

- Strengths: The experiments evaluate the framework's performance using well-justified tasks (masked-node reconstruction). Furthermore, the authors conduct systematic ablation studies to compare different framework configurations.

- Weaknesses:
  - No comparison with existing LLM frameworks.
  - No ablation studies to isolate the contribution of the graph structure vs. LLM capabilities.
  - Similarity scores (via text embeddings) may not capture practical utility.

**Methods And Evaluation Criteria:**

- Agent-data graph + TextGNN: The framework is innovative, simple and well-suited for modeling dynamic research communities. The use of LLMs as agent functions for text-based message passing is a creative adaptation of GNNs.
- RESEARCHBENCH: The node-masking task is a reasonable proxy for evaluating reconstruction fidelity, but it focuses on similarity rather than research quality (e.g., novelty, feasibility). Human evaluation or downstream task validation (e.g., citation impact simulation) could strengthen the assessment.

**Other Comments Or Suggestions:**

- Include a running example of a simulated research workflow to improve readability.

**Other Strengths And Weaknesses:**

- Other Strengths:
  - The modular design allows for flexible expansion with new node types (e.g., code repositories, blogs) and edge types (e.g., commits, participations), offering the possibility to simulate complex academic ecosystems.
  - The maintenance of hidden states in the text space (TextGNN) effectively preserves the semantic coherence of research outputs.

- Other Weaknesses:
  - The social dynamics in simulated academic communities are not considered.
  - The computational cost of text generation may limit the practicality of large-scale community simulations.
  - Not tested the dynamic community evolution capabilities.

**Questions For Authors:**

1. Could you provide quantitative evidence (e.g., human evaluation scores) for the interdisciplinary idea generation claims? This would help validate if the generated ideas are truly novel and feasible beyond surface-level text similarity metrics.

2. How does RESEARCHTOWN compare against existing LLM frameworks for idea generation? Comparative results would help position the framework's unique contributions.

3. The similarity scores (0.67 paper/0.49 review) are presented as evidence of realistic simulation - what is the baseline/expected score range for human-written content? Contextualizing these metrics would strengthen their interpretation.

4. Have you considered modeling social dynamics (e.g., senior-junior researcher interactions, institutional affiliations) in the community graph?

**Relation To Broader Scientific Literature:**

RESEARCHTOWN builds on:
1. LLM-driven research automation (e.g., The AI Scientist) but emphasizes multi-agent collaboration in simulated research communities.
2. Multi-agent LLM systems for social simulation (e.g., Generative Agents, S^3, SOTOPIA) but extends them to research communities framework.
3. Graph-based research modeling (citation networks, academic social networks) but shifts focus from analysis to dynamic simulation.

**Theoretical Claims:**

N/A. The paper does not present formal theoretical proofs.

---

> ### Author Rebuttal · Authors · 2025-04-01
>
> Thank you very much for your insightful and constructive comments. We address each of your comments in detail.
>
> **[Novelty+feasibility evaluation with LLM and human]** Please check the same tag under **Reviewer YZ7h** fo novelty and feasibility evaluation.
>
> **[Fine-grained evaluation with LLM and human]** Please check the same tag under **Reviewer YZ7h** for fine-grained consistency evaluation.
>
> **[Cost and scalability for ResearchTown]** Please check the same tag under **Reviewer Cny3** for analyzing the computational cost.
>
> **[Future Application of ResearchTown]** Please check the same tag under **Reviewer Cny3** for analyzing social dynamics simulation.
>
> **[Baseline score of realistic simulation]** To check whether ResearchTown provides realistic simulation, we benchmark similarity in real-world research activity. For paper writing, we reference two concurrent papers [1,2] recognized for presenting nearly identical ideas—yet with different writing styles and experiments—which yield a VoyageAI similarity of 0.8244. This suggests that scores above 0.82 can potentially indicate strong idea overlap. For review writing, we analyze data of reviewers evaluating the same paper. The average inter-reviewer similarity is 0.5900 (strengths) and 0.5904 (weaknesses), reflecting natural variance in human judgment. These inter-similarity score in the real world confirm that ResearchTown’s similarity scores represent realistic simulation.
>
> **[Ablation study on LLMs and graphs]** For graph structure ablation, *Table 1-2* already demonstrates the effect of different types of neighboring nodes during aggregation with different sub-parts of the neighborhood. It can be considered as ablation on graph structures.
>
> Additionally, we provide results on **LLM ability variation** using *Deepseek-V3* (potentially larger than GPT-4o-mini) and *Qwen2.5-7B-Instruct* (potentially smaller than GPT-4o-mini) on a sample of 100 examples each from PaperBench and ReviewBench.
>
> For paper writing tasks, we find when given the same aggregation setting, the performance improves when the models are larger: Qwen2.5-7B-Instruct < GPT-4o-mini < Deepseek-V3 (evaluated by openai embedding).
>
> | Aggregation Setting | Qwen2.5-7B-Instruct | GPT-4o-mini | Deepseek-V3 |
> | ------------------- | ------------------- | ----------- | ----------- |
> | Global-agg          | 71.94               | 73.93       | **74.09**   |
>
> For review writing tasks, we test the performance of different models under different aggregation settings. We observe that Deepseek-V3 benefits more from multi-agent + multi-paper settings; GPT-4o-mini tends to be stricter than human reviewers, especially when more context is available.
>
> | Aggregation Setting | Model       | Strength | Weakness | Avg Δs (Abs) |
> | ------------------- | ----------- | -------- | -------- | ------------ |
> | Data-agg            | GPT-4o-mini | 71.08    | 68.22    | 1.17         |
> | Global-agg          | GPT-4o-mini | 62.40    | 56.79    | 1.49         |
> | Data-agg            | Deepseek-V3 | 68.63    | 67.76    | 1.08         |
> | Global-agg          | Deepseek-V3 | 68.04    | 67.99    | **0.74**     |
>
> **[More baseline comparison]** The TextGNN framework in our work is a general-purpose multi-agent simulation tool where existing LLM-based frameworks can be used to define the message-passing mechanism. Beyond our default setup, we conducted a small-scale experiment extending the AGG-agent setting [*Line297–298*] into a **multi-turn conversation** using the SWARM framework. This mimics multiple iterations within a single GNN layer and improves similarity scores from 52.32 to 57.68, evaluated using `text-embedding-large-3`. We also compare ResearchTown with the Sakana AI scientist framework, which uses five rounds of single-agent reflection. On the same subset of paper writing tasks, Sakana **AI Scientist** achieves a score of 0.63, while ResearchTown reaches 0.66 using multi-agent simulation.
>
> **[Running example]** *Table14-28 (Line1430-2254)* includes multiple running examples from ResearchTown: (1) Table14–17: Paper writing examples; (2) Table18: Review writing examples; (3) Table19–28: Interdisciplinary research examples. We will add more *end-to-end running examples*, beginning from citations and ending with paper + review outputs, in the revised version.
>
> **[Downstream task validation of ResearchTown]** We would include more downstream tasks like citation prediction in the modified version of our paper. Based on our HighImpactPaperBench (details in Appendix C.3, Line837-853), we observe that highly cited novel papers like GAN and VAE are harder to simulate the thinking process based on citation and multi-agent, indicating some kind of correlation between citation impact and simulation tasks.
>
> [1] Chen et al. ReSearch: Learning to Reason with Search for LLMs via Reinforcement Learning
>
> [2] Jin et al. Search-R1: Training LLMs to Reason and Leverage Search Engines with Reinforcement Learning

---

### Official Review · Reviewer_Cny3 · 2025-03-14

**Overall Recommendation:** 4

**Summary:**

This paper aims to simulate the human research community (called ResearchTown), which is modeled as a graph structure, where researchers and papers are represented as nodes and they are connected based on their relationships. Also, each researcher over the graph structure is powered by Large Language Models (LLMs), making the simulation of the research community under the framework of multi-agent (or multi-LLM) collaboration. For evaluation, the authors propose two tasks: paper writing and review writing based on the process of paper reading, and handle them with text-based message passing over the graph structure. The authors then show that the proposed ResearchTown can not only provide a realistic simulation of the research activity (e.g., paper writing and review writing) but also facilitate interdisciplinary research.

---

### Update after rebuttal:

Thank you so much for your response, which addresses my last concern on the scalability of the TextGNN. In my view, I still believe that the design of TextGNN has a clear latency issue compared to the typical GNN that plays over the embedding space (which I hope the authors would discuss in the updated version); however, I also see some benefits of propagating and aggregating information of nodes via natural language texts (in terms of their effectiveness when working with LLMs and their interpretability). I will raise my score from 2 to 4 (accept). Good luck!

**Claims And Evidence:**

The claims made in the submission are supported by clear and convincing evidence.

**Essential References Not Discussed:**

There are some papers [A, B] that aim to automate the research process with multi-agent collaboration frameworks (similar to the concept of the ResearchTown that aims to simulate the research process with multi-agents), and it may be worth discussing them.

[A] ResearchAgent: Iterative Research Idea Generation over Scientific Literature with Large Language Models

[B] Chain of Ideas: Revolutionizing Research Via Novel Idea Development with LLM Agents

**Experimental Designs Or Analyses:**

Most of the experiments are sound and valid. However, I view validating the proposed ResearchTown with one single (proprietary) LLM as a clear weakness of the current experimental setting. Specifically, the proprietary LLM is usually not reproducible even if we control the temperature value, and also it is questionable whether the proposed framework can work with other LLMs (such as larger or smaller than GPT-4o-mini).

**Methods And Evaluation Criteria:**

While the proposed (graph-structure-based) approach to simulate the human research community is reasonable, I have an important concern about the evaluation criteria. First of all, as an evaluation metric, the authors use the embedding-level similarity between the generated item (such as the paper or review) and the target item; however, I believe one single embedding-level similarity may be suboptimal to measure the alignment between them. In other words, there are many aspects that should be considered when validating whether the generated paper (for example) is similar to the target paper (such as factual consistency, logical coherence, methodological relevance, or novelty), and embedding-based metrics often fail to account for finer-grained structural and conceptual differences.

**Other Comments Or Suggestions:**

Some sentences are not clear and it would be worth clarifying them:
* Could you clarify the sentence in Lines 273 - 277 (starting with "More specifically")?
* In Lines 359 - 361, it is not clear why the inclusion of both the agent and data nodes would degrade the performance of the review writing simulation task.
* In Lines 317 - 318, how to select the top 5 researchers most related to the paper?
* Overall, I feel it would be worth including the inputs and outputs for each task. For example, in paper reading and subsequent paper writing, do you use the full text of the neighboring papers? For the review writing task, what are the targets that should be generated (i.e., they are the strengths and weaknesses of the paper)?

**Other Strengths And Weaknesses:**

While the authors claim that the proposed ResearchTown is designed to simulate the human research communities with LLMs, the current set of benchmark evaluations is limited to paper writing and review writing, done based on the paper reading among researchers. I believe there are more activities (that can be measured) within ResearchTown with more types of nodes and extra interactions between them, and it would be valuable if the authors would discuss them. Also, the proposed TextGNN (i.e., the message passing framework between nodes with text) seems not scalable when the number of layers becomes moderate (e.g., three or four), due to the nature of text-based communication in contrast to embedding-based communication where the information across nodes can be aggregated more efficiently over the vector space. I view this as another limitation of the proposed approach, and perhaps showing some experiments on the scalability of the proposed approach according to the number of layers may be beneficial.

**Questions For Authors:**

Please see my previous comments. I feel the core idea of this paper is interesting, I would like to increase my rating if the authors would address them.

**Relation To Broader Scientific Literature:**

The key contribution of this paper is related to the recent effort to simulate or automate AI research, which is a very important and timely topic.

**Theoretical Claims:**

N/A

---

> ### Author Rebuttal · Authors · 2025-04-01
>
> Thank you very much for your insightful and constructive comments. We address each of your comments in detail.
>
> **[Novelty+feasibility evaluation with LLM and human]** Please check the same tag under **Reviewer YZ7h** for novelty and feasibility evaluation conducted by both LLMs and humans.
>
> **[Fine-grained evaluation with LLM and human]** Please check the same tag under **Reviewer YZ7h** for factual consistency and other fine-grained metrics.
>
> **[Ablation study on LLMs and graphs]** Please check the same tag under **Reviewer 5ZWh** for experimental results with three different LLMs.
>
> **[Review writing performance analysis]** For clarification of [*Line359-361*], please check the same tag under **Reviewer t7zy** for more analysis about why the performance drop.
>
> **[Input and output of ResearchTown]** ResearchTown processes either full papers or abstracts, depending on the task, and outputs standardized formats to support consistent and scalable evaluation. For paper writing, it generates a condensed 5Q format [*Line893–894*]; for review writing, it produces bullet-point strengths and weaknesses [*Line910*]. Prompt templates are shown in Table12–13 [*Line1379–1425*], with examples in Table14–28 [*Line1432–2253*]. As described in Appendix C.3 [*Line837–853*], this alignment reduces evaluation complexity and enables sub-part similarity scoring [*Line898–902*]. Input sources vary: only the paper’s abstract is used during reading [*Line994–995*] and full papers are used for review writing [*Line1171–1172*]. Aggregation setting details are in [*Line688–726*].
>
> **[Future Application of ResearchTown]** In [*Line139–144*], any research-related content—e.g., images, codebases, models, or social media posts—can be represented as nodes in the agent-data graph, with edge types like “cite paper,” “release model,” or “comment on X post” (examples in Figure 1) defining interactions. By specifying appropriate edge types and agent functions (`f_u` in [*Line134–135*]), the framework can be extended to simulate tasks such as code writing, model release, panel discussions, or lectures. While we focus on paper and review writing due to their importance, available real-world data, and simplicity, the framework supports broader applications.
>
> Additionally, ResearchTown can be extended to model social dynamics such as peer pressure, collaborations, and institutional roles via agent-agent relationship edges [*Line133–135*]. Our current implementation already includes role-based dynamics (e.g., leader vs. participant), and we plan to support richer simulations of institutional and reputational factors in future work.
>
> **[Cost and scalability of ResearchTown]** TextGNN’s complexity scales linearly with the number of layers under standard GNN inference with full-batch inference. Our implementation uses GraphSAGE-style [1] to support per-paper evaluation, which is slightly less efficient but more practical. In line with findings from models like GraphSAGE, we observe that a 2-layer TextGNN is both robust and sufficient, making the cost affordable and controllable. Instead of deepening the GNN layer, we increase the number of transformation steps within each message-passing layer—interpreted as more agentic conversations—making it another scalable and effective approach. To demonstrate intra-layer scalability, we extend the AGG-agent setting [*Line297–298*] using the SWARM framework for multi-turn agent interactions, which boosts similarity scores from 52.32 to 57.68. These results show that agentic iteration within layers offers a practical and scalable alternative.
>
> **[Details on reviewer selection]** To simulate realistic reviewer assignment, we collect over 3,000 unique authors from the author list of PaperBench dataset and generate profiles by summarizing their recent publications. Using the `voyage-3` API, we embed each profile with the target paper’s abstract and select the top 5 most similar researchers, excluding the original authors. This method enables high-quality reviewer-paper matching—for example, a social learning paper was matched with reviewers experienced in social science and LLMs.
>
> **[Details on review writing inputs]** To clarify [*Line273–277*], in the review writing simulation (Algorithm 1 [*Line220–236*]), both the paper and review are outputs of the ResearchTown pipeline. While one option is to evaluate reviews based on generated papers, this introduces compounding errors if the paper is inaccurate. Instead, as noted in [*Line273*], we use the ground-truth paper as input to isolate and more reliably evaluate the review writing stage.
>
> **[Missing related work]** Thank you for highlighting valuable related works—we will include and discuss them in the revised version around [*Line120–125*].
>
> [1] Hamilton et al. Inductive Representation Learning on Large Graphs

---

> > ### Comment · Reviewer_Cny3 · 2025-04-04
> >
> > Thank you for your response, which addresses most of my concerns. One remaining concern that I have is regarding the scalability of the proposed approach. As described in my original review, the proposed text-based GNN framework propagates the information between nodes via natural language (instead of embeddings), which may be less scalable. For example, in the case of embeddings, the information propagated from 10 different nodes is merged into one single representation (typically); however, in the case of natural language, the proposed texts from 10 different nodes are 10 times longer than the text that each node creates. In this regard, I think the authors may provide some experimental results to clarify this, for example, is the proposed method scalable with more than only 2 layers?

---

> > > ### Author Response · Authors · 2025-04-08
> > >
> > > Thank you for your additional feedback. We're happy to answer any further questions and would appreciate it if you could consider raising the score.
> > >
> > > **[constrained output length for each layer of TextGNN]**
> > >
> > > The aggregation function for classical GNN (*Equation 3, Lines 145-148*), which is often a pooling or mean operation, is used to condense all neighborhood information into one embedding with the same size as input. Similarly, for our TextGNN layers (*Equations 4-5, Lines 182-203*), $f_u$ and $f_g$ are acted as aggregation function similar in classical GNN, they produce outputs with controlled textual formats and similar lengths with updated information in the neighborhood nodes by *summarizing* with LLMs. Therefore, *the output length of multiple layers of TextGNN would not increase but keep approximately the same*.
> > >
> > > We achieve such length control in TextGNN via *format control in prompting*. We specifically designed prompts ensure each output adheres to pre-defined constraints:
> > >
> > > - **Paper writing:** "5Q" format for paper (mentioned in *Lines 1068-1089*).
> > > - **Review writing:** ~200-word bullet points for review (mentioned *Lines 1165-1196*).
> > > - **Paper reading:** 100-300 words persona for researcher (mentioned *Lines 996-998*).
> > >
> > > These prompt-controlled constraints ensure stable output lengths at every TextGNN layer, avoiding text length inflation with increasing depth. Each aggregation step condenses and prioritizes critical information, effectively filtering less relevant details.
> > >
> > >
> > >
> > > **[multi-layer aggregation example]**
> > >
> > > We illustrate why controlled length can support multi-layer of TextGNN clearly with an example aggregation across multiple layers. As you can see, an example of 3-layer TextGNN does not result in longer and longer text but remains a highly informative condensed version of text:
> > >
> > > *Layer 1*: Paper1, Paper2, Paper3 (each 5Q format) → Researcher Profile1 (100-300 word persona).
> > >
> > > *Layer 2*: Researcher Profile1, Researcher Profile2, Paper4, Paper5 (each 5Q format) → Paper6 ( 5Q format).
> > >
> > > *Layer 3*: Researcher Profile3, Researcher Profile4, Paper6, Paper7 (each 5Q format) → Paper8 ( 5Q format).
> > >
> > > As demonstrated in experiments (*Table 23*), even when aggregating many paper and researcher inputs, TextGNN outputs consistently maintain controlled lengths. A more concrete example we show below is that more layers can provide more condensed version of description but not necessarily with longer length:
> > >
> > > Part of 3-layers of TextGNN paper writing results:
> > >
> > > *This framework will utilize structural causal models (SCMs) to identify causal relationships while incorporating machine learning methods to enhance predictive performance. Key metrics for evaluation will include causal identifiability, robustness to distribution shifts, and interpretability of the learned models. The expected outcomes include a comprehensive understanding of causal mechanisms in complex systems and improved performance of machine learning models in real-world applications.*
> > >
> > > Part of 2-layer of TextGNN paper writing results:
> > >
> > > *This framework will utilize causal influence diagrams to model dependencies among agents and their intentions, allowing for the computation of causal queries related to decision-making processes. The expected outcomes include a clearer understanding of how intentions influence actions in AI systems, improved algorithms for causal discovery in multi-agent settings, and enhanced safety analysis tools that can be applied to various AI applications.*
> > >
> > > More aligned information is simulated and discussed when more layers of TextGNN are considered and aggregated. However, the length remains approximately the same.
> > >
> > >
> > > **[empirical validation on more layers of TextGNN]**
> > >
> > > To empirically validate the scalability of TextGNN beyond two layers (paper reading and writing), we conducted additional experiments incorporating multi-hop information besides the current 2-hop. Previously, we initialized researcher personas by aggregating researchers' authored papers and leveraging immediate paper and researcher neighborhoods to generate new papers. In our extended experiments, we now include authors of those cited papers and papers cited by/related with those cited papers, effectively integrating deeper multi-hop connections. Due to the complexity of collecting extensive multi-hop data, our evaluation is limited to 42 samples, focusing specifically on the quality of the generated paper nodes. We can see that 3-layer provides more improvement under full-agg, agent-agg, and self-agg settings while drops slightly for data-agg. The drop in data-agg might be because too much noisy paper is involved.
> > >
> > > | Setting           | OpenAI Sim Avg |
> > > | ----------------- | ------- |
> > > | 2-layer self-egg  | 0.6182  |
> > > | 2-layer agent-agg | 0.7068  |
> > > | 2-layer data-agg  | 0.7508  |
> > > | 2-layer full-agg  | 0.7348  |
> > > | 3-layer self-agg  | 0.6225  |
> > > | 3-layer agent-agg | 0.7488  |
> > > | 3-layer data-agg  | 0.7271  |
> > > | 3-layer full-agg  | 0.7435  |

---

### Official Review · Reviewer_YZ7h · 2025-03-14

**Overall Recommendation:** 2

**Summary:**

This research work starts from the idea that we can leverage LLMs to simulate human research communities and proposes ResearchTown, a multi-agent framework designed to model human research societies and behaviors. This work also introduces TextGNN to model various research activities, including paper reading, paper writing, and review writing. In addition, it develops ResearchBench, which uses a node-masking task to evaluate whether ResearchTown can successfully simulate the masked paper node by masking a paper within a graph.

**Claims And Evidence:**

No. This paper claims that the simulation of automated research processes should be correlated with human research processes. However, under this assumption, it seems that discovering groundbreaking and insightful scientific ideas may become more challenging. The multi-agent system tends to produce known scientific knowledge while sacrificing the ability to learn, explore, and discover.

**Essential References Not Discussed:**

N/A

**Experimental Designs Or Analyses:**

The specific calculation methods for the metrics used in the experimental section are not very clear. For example, in Table 1, were text-embedding-large-3 and voyage-3 used to extract embeddings? Was cosine similarity used to compute the scores?

**Methods And Evaluation Criteria:**

Some specific implementation details are unclear:

- 1) How does ResearchTown generate a complete research paper including many parts of abstract, method, and experimental results?

- 2) Unlike standard GNNs, each node in TextGNN is based on the text space, but what exactly constitutes the text? Is it the entire paper or a condensed summary of the paper?

**Other Comments Or Suggestions:**

N/A

**Other Strengths And Weaknesses:**

[Strengths]

- 1) This work models human research communities by constructing TextGNN and attempts to use a masking-node approach for evaluation, which is insightful.

- 2) This work constructs ResearchBench, a benchmark consisting of 1,000 paper-writing tasks and 200 review comments.

[Weaknesses]

- 1) This paper claims that the simulation of automated research processes should be correlated with human research processes. However, under this assumption, it seems that discovering groundbreaking and insightful scientific ideas may become more challenging. The multi-agent system tends to produce known scientific knowledge while sacrificing the ability to learn, explore, and discover.

- 2) ResearchTown conducts experimental validation to demonstrate its alignment with human research communities. However, how can we further verify that the ideas generated by ResearchTown are valuable and can be followed? Moreover, this work has the potential to advance automatic scientific discovery, but it requires more robust evaluation and validation.

- 3) How does ResearchTown generate a complete research paper including many parts of abstract, method, and experimental results?

- 4) Unlike standard GNNs, each node in TextGNN is based on the text space, but what exactly constitutes the text? Is it the entire paper or a condensed summary of the paper?

- 5) The specific calculation methods for the metrics used in the experimental section are not very clear. For example, in Table 1, were text-embedding-large-3 and voyage-3 used to extract embeddings? Was cosine similarity used to compute the scores?

**Questions For Authors:**

Please refer to Part of [Other Strengths And Weaknesses]

**Relation To Broader Scientific Literature:**

ResearchTown conducts experimental validation to demonstrate its alignment with human research communities. However, how can we verify that the ideas generated by ResearchTown are valuable and can be followed? Moreover, this work has the potential to advance automatic scientific discovery, but it requires more robust evaluation and validation.

**Theoretical Claims:**

This work does not involve theoretical derivations.

---

> ### Author Rebuttal · Authors · 2025-04-01
>
> Thank you very much for your insightful and constructive comments. We address each of your comments in detail.
>
> **[Input and output of ResearchTown]**
> Please check the same tag under **Reviewer Cny3** for a detailed explanation.
>
> **[Creativity of ResearchTown's output]**
> LLMs have been shown capable of generating novel research ideas through large-scale human studies [1]. Multi-agent role-playing and discussion further enhance creativity [2] and originality [3]. Based on these studies, ResearchTown encourages exploration of interdisciplinary ideas via structured prompting and multi-agent role-play design [*Line414-426*]. Examples are shown on Page27–Page41. We design HighImpactPaperBench in Appendix C.3 [*Line837-853*] to show ResearchTown’s capacity to generate impactful research [*Line355-376*].
>
> **[Metric calculation in ResearchTown]**
> Our evaluation metrics for both paper and review writing are detailed in Appendix E [*Line886–930*]. To enable meaningful comparison, we standardize papers into a condensed 5Q format [*Line893–895*] and reviews into a bullet point format [*Line909–911*]. As described in Appendix C.3 [*Line837–853*], we also convert real-world papers and reviews into these aligned formats using strong LLMs. We then compute cosine similarities with `voyage-3`,`text-embedding-3-large` and `nv-embed-v2`, based on Equation 21 for papers [*Line898–901*] and Equation 22 for reviews [*Line912–915*]. Detailed sub-part similarity scores are provided in Table 3 [*Line935–953*].
>
> **[Fine-grained evaluation with LLM and human]**
> We extend beyond embedding-based metrics using prompting-based GPT-4o evaluations, covering factual consistency, logical/method alignment, motivation, and context. Each is scored from 1–10.
>
> |            | Semantic Similarity | Factual Consistency | Motivation Alignment | Method Alignment | Logical Consistency | Application Context Consistency |
> | ---------- | ------------------- | ------------------- | -------------------- | ---------------- | ------------------- | ------------------------------- |
> | Self-Agg   | 1.22                | 1.49                | 1.57                 | 1.23             | 1.22                | 1.41                            |
> | Agent-Agg  | 2.51                | 2.40                | 3.50                 | 2.41             | 2.22                | 3.08                            |
> | Data-Agg   | 3.94                | 3.48                | 5.05                 | 3.68             | 3.27                | 4.97                            |
> | Global-Agg | 4.43                | 3.94                | 5.56                 | 4.32             | 3.69                | 5.33                            |
>
> These results show clearer differences than embedding-based scores, with Global-Agg (paper nodes + agents) performing best on motivation/method alignment. A human study over 40 papers on similarity-based evaluation (20 in-domain, 20 cross-domain) yields Pearson’s r = 0.745 and Spearman’s ρ = 0.735, supporting the validity of embedding/LLM-based metrics.
>
> **[Novelty+feasibility evaluation with LLM and human]**
> We conducted small-scale human and LLM evaluations (20 interdisciplinary and 20 ML papers) on novelty and feasibility. Interdisciplinary examples include papers tagged with multiple fields on arXiv (e.g., CS+Economics). Each cell below includes *real-world* data evaluation vs *simulated* results evaluation.
>
> |                                   | LLM-Eval Novelty | LLM-Eval Feasibility | Human-Eval Novelty | Human-Eval Feasibility |
> | --------------------------------- | ---------------- | -------------------- | ------------------ | ---------------------- |
> | Interdisciplinary Research Papers | 7.5 vs. 7.35     | 6.45 vs. 6.9         | 7.4 vs. 7          | 7.4 vs. 7              |
> | ML Research Papers                | 7.85 vs. 7.65    | 6 vs. 6.9            | 6.6 vs. 6.95       | 5.65 vs. 7.15          |
>
> LLM vs human novelty/feasibility scores show a moderate Pearson correlation (~0.38), highlighting evaluation difficulty. ResearchTown’s outputs are generally comparable in novelty and more feasible than real papers in the ML domain.
>
> **[Text form defined in TextGNN]**
> Each node’s "hidden state" in TextGNN represents a condensed form of a paper or review. Initially, full paper contents serve as node states [*Line158-161*]. After iterative message passing, paper nodes adopt the standardized 5Q format [*Line893-894*], condensing information for easier evaluation. Review nodes similarly use bullet points [*Line910*] as condensed information.
>
> [1] Si, et al. Can LLMs Generate Novel Research Ideas? A Large-Scale Human Study with 100+ NLP Researchers.
>
> [2] Lu, et al. LLM Discussion: Enhancing the Creativity of Large Language Models via Discussion Framework and Role-Play.
>
> [3] Zhao, et al. Assessing and Understanding Creativity in Large Language Models.
>
> [4] Lu, et al. The AI Scientist: Towards Fully Automated Open-Ended Scientific Discovery.

---

> > ### Comment · Reviewer_YZ7h · 2025-04-02
> >
> > Thank you for the authors' response. After carefully reading the rebuttal, I still have the following concerns:
> >
> > - The evaluation and metric calculation method used by ResearchTown seems somewhat unreasonable. ResearchTown employs cosine similarities to indicate alignment with real-world research community, but ideas and research are more high-level concepts. A single idea can have multiple forms of concrete description or implementation method, making it difficult to accurately be captured using cosine similarities.
> >
> > - In TextGNN, each node represents a summary of a paper or review (generated using 5Q format), which may result in the loss of important knowledge from the papers (such as key mathematical equations or algorithm workflow), making it challenging to simulate the research process of human society.
> >
> > As a result, I choose to maintain my initial rating.

---

> > > ### Author Response · Authors · 2025-04-04
> > >
> > > Thank you for your additional feedback. We're happy to answer any further questions and would appreciate it if you could consider raising the score.
> > >
> > > **[Decompisitionality of our evaluation metric]**
> > >
> > > We agree that *“a single idea can manifest through diverse descriptions or implementation strategies, rendering surface-level metrics like cosine similarity inadequate for capturing conceptual equivalence.”* This is exactly what motivates us to propose **5Q-based evaluation framework**—a decompositional evaluation with 5 sub-parts:
> > >
> > > Q1: **What is the problem?**
> > >
> > > Q2: **Why is it interesting and important?**
> > >
> > > Q3: **Why is it hard?**
> > >
> > > Q4: **Why hasn’t it been solved before?**
> > >
> > > Q5: **What are the key components of the proposed approach and results?**
> > >
> > > This structure enables alignment between papers that differ methodologically (Q5) but share similar motivations and problem framings (Q1–Q4). For instance, in [1] and [2], despite distinct methods and settings, experts would find strong alignment on Q1–Q3.
> > >
> > > We validate this framework through per-question similarity analysis (Table 3, Lines 935–953). In *PaperBench-easy*, Q2 (motivation) shows the highest alignment (80.25), while Q4 and Q5 score lower (71.54, 70.60), indicating that motivation is easier to capture than novelty or method. In *PaperBench-hard*, alignment drops on Q1, Q4, and Q5 (55.35, 58.55, 57.84), showing that even problem formulation becomes challenging in complex domains, while Q2 and Q3 remain relatively stable.
> > >
> > > These results align with our intuition: understanding *why* a problem matters (Q2, Q3) is easier with domain knowledge, while identifying novel formulations (Q1) and implementation details (Q5) requires deeper expertise. The 5Q framework thus enables structured, fine-grained evaluation beyond surface-level similarity.
> > >
> > >
> > >
> > > **[Scalability of our evaluation metric]**
> > >
> > > To address the challenge that a *single idea can take many concrete forms*, we complement decomposition with **scalability**. LLMs can generate hundreds of semantically distinct research questions from a single prompt, but evaluating these outputs traditionally requires domain experts—a process that is **costly, slow, unscalable, and hard to reproduce**. For example, [3] spent thousands hiring top-tier researchers solely for annotation and review, which is infeasible for evaluating large-scale, automated research generation. Our approach replaces this bottleneck with **semantic similarity over 5Q-decomposed representations**. We can select the best among sampled and make the score the final result.
> > >
> > >
> > >
> > > **[Extensibility of our evaluation metric]**
> > >
> > > While we acknowledge the importance of elements like *mathematical formulations* or *algorithmic workflows*, our framework is **inherently extensible**—the 5Q format can be expanded into 6Q or 7Q by adding domain-specific dimensions such as *algorithmic structure* or *key theoretical results*. This is especially valuable in systems and theory papers, enabling **more fine-grained and domain-aware similarity analysis**. As demonstrated in **[Fine-Grained Evaluation with LLM and Human]**, our approach also supports integration of non-semantic metrics like **logical consistency** and **factual accuracy**, making it extensible from evaluation metric perspective.
> > >
> > >
> > >
> > > **[Reliability of our evaluation metric]**
> > >
> > > Our embedding-based / LLM-based similarity metric builds on state-of-the-art models optimized for knowledge-intensive tasks. **Voyage AI embeddings**, widely adopted in real-world RAG systems, are designed to reduce hallucination and excel in high-precision semantic retrieval—making them ideal for evaluating research content. Additionally, sota LLMs are highly effective at semantic comparison. As demonstrated in our **[Fine-Grained Evaluation with LLM and Human]** section, our method yields interpretable similarity scores, and human evaluations further validate its alignment with expert judgment.
> > >
> > >
> > >
> > > **[Main contribution of paper]**
> > >
> > > We emphasize the main contribution here. The main contribution is **ResearchTown**, a framework that simulates collaborative research activities by modeling paper writing and peer review as dynamic message-passing on a graph. It represents the research ecosystem as a graph of researchers, papers, and reviews, capturing complex temporal interactions in a structured and scalable way. This design supports both realistic simulation and graph-based evaluation through techniques like node masking. Inspired by Graph Neural Networks (GNNs), ResearchTown employs a **TextGNN** for inference, where nodes are iteratively generated and updated via textual message passing, enabling nuanced modeling of how research communities evolve over time.
> > >
> > >
> > >
> > > [1] Chen et al. ReSearch: Learning to Reason with Search for LLMs via Reinforcement Learning
> > >
> > > [2] Jin et al. Search-R1: Training LLMs to Reason and Leverage Search Engines with Reinforcement Learning
> > >
> > > [3] Si et al. Can LLMs Generate Novel Research Ideas?

---

### Decision · Program_Chairs · 2025-05-01

**Decision:**

Accept (poster)

**Comment:**

The authors presents ResearchTown, a multi-agent simulation framework for modeling research communities through an agent-data graph and a text-based message-passing mechanism. The authors introduce ResearchBench to evaluate the quality of simulated papers and reviews using masked node prediction and embedding-based similarity metrics. The system supports tasks such as writing papers, reviewing papers, and generating interdisciplinary ideas. The authors run experiments showing alignment between simulated and real-world outputs.

The core idea is interesting, and the overall design of ResearchTown is interesting nd well-motivated. The modeling of research activities as structured message-passing over text nodes seem unique and the proposed 5Q decomposition for paper evaluation provides a meaningful lens into simulation quality.

Reviewers generally appreciated the ambition of the work and the potential of the framework, though concerns remain around the limits of embedding-based evaluation, scalability of the TextGNN, and the robustness of the interdisciplinary generation claims. The authors provide detailed rebuttals addressing these points. While not all concerns are fully resolved the clarifications and empirical evidence strengthen the case for the paper.

On balance, I find the direction promising and the framework potentially impactful, though there is room for refinement in both evaluation and framing. I strongly recommend the authors take into account the points raised by the reviewers and by my notes above in their revisions.